# LOCAL CONVERGENCE OF SIMULTANEOUS MIN-MAX ALGORITHMS TO DIFFERENTIAL EQUILIBRIUM ON RIEMANNIAN MANIFOLD

**Sixin Zhang**
Université de Toulouse, INP, IRIT
2, rue Camichel, BP 7122, 31071 Toulouse Cedex 7, France
sixin.zhang@irit.fr

## ABSTRACT

We study min-max algorithms to solve zero-sum differential games on Riemannian manifold. Based on the notions of differential Stackelberg equilibrium and differential Nash equilibrium on Riemannian manifold, we analyze the local convergence of two representative deterministic simultaneous algorithms $\tau$-GDA and $\tau$-SGA to such equilibria. Sufficient conditions are obtained to establish the linear convergence rate of $\tau$-GDA based on the Ostrowski theorem on manifold and spectral analysis. To avoid strong rotational dynamics in $\tau$-GDA, $\tau$-SGA is extended from the symplectic gradient-adjustment method in Euclidean space. We analyze an asymptotic approximation of $\tau$-SGA when the learning rate ratio $\tau$ is big. In some cases, it can achieve a faster convergence rate to differential Stackelberg equilibrium compared to $\tau$-GDA. We show numerically how the insights obtained from the convergence analysis may improve the training of orthogonal Wasserstein GANs using stochastic $\tau$-GDA and $\tau$-SGA on simple benchmarks.

## 1 INTRODUCTION

Riemannian min-max problem has attracted a lot of research attention in recent years, with various machine learning applications including robust PCA (Jordan et al., 2022), robust neural network training (Huang & Gao, 2023) and generative adversarial network (GAN) (Han et al., 2023). This problem is formalized as a two-player zero-sum game, where the variables of each player are constrained on the Riemannian manifold $\mathcal{M}_1$ and $\mathcal{M}_2$,

$$\min_{x \in \mathcal{M}_1} \max_{y \in \mathcal{M}_2} f(x, y).$$

When $\mathcal{M}_1$ and $\mathcal{M}_2$ are Euclidean, gradient-based methods such as gradient-descent-ascent (GDA) (Daskalakis & Panageas, 2018; Lin et al., 2020), extra-gradient (Gidel et al., 2019; Mahdavinia et al., 2022), optimistic mirror descent (Mertikopoulos et al., 2019), Hamiltonian-gradient descent (Loizou et al., 2020) and symplectic gradient-adjustment (SGA) (Balduzzi et al., 2018) are often considered to solve this problem. When $\mathcal{M}_1$ and $\mathcal{M}_2$ are Riemannian, how to extend existing algorithms from Euclidean space to Riemannian manifold and how to analyze their convergence become an interesting topic in recent years. In Table 1, we summarize some related works on the Riemannian min-max problem. In particular, we observe that under suitable assumptions of the game, one can obtain the global convergence of a suitable algorithm towards Nash equilibrium. One major issue is that these assumptions typically do not hold in applications such as GAN (Razaviyayn et al., 2020). However, when a min-max problem is non-convex and non-concave, even in the Euclidean case, the global convergence of existing algorithms can be complicated (Hsieh et al., 2021). To achieve this, a suitable notion of solution set and novel algorithmic development and analysis are needed (Jin et al., 2020; Benaim & Miclo, 2024). In this article, we focus on the following solutions sets: differential Stackelberg equilibrium (DSE) and differential Nash equilibrium (DNE), which are known to be generic among local Stackelberg equilibrium (resp. local Nash equilibrium) in Euclidean smooth non-convex non-concave min-max problems (Fiez et al., 2020). We then develop local convergence analysis of min-max algorithms and show its relevance to the training of GAN.

Table 1: Related works of Riemannian min-max problem. These works study the global convergence of min-max algorithms to different solution sets. However, they make assumptions on the game $(\mathcal{M}_1, \mathcal{M}_2, f)$ which typically do not hold in the practice of GANs.

| Reference | Class of problem | Solution set / Algorithm |
|---|---|---|
| Zhang et al. (2023) | Geodesically convex (compact) $\mathcal{M}_1, \mathcal{M}_2$ $f$ is geodesically convex/concave (quasi), | Nash equilibrium / Extra-gradient semi-continuous (lower/upper) |
| Jordan et al. (2022) | +Bounded $\mathcal{M}_1, \mathcal{M}_2$ $f$ is geodesically convex/concave, smooth | Nash equilibrium / Extra-gradient |
| Huang & Gao (2023) | Euclidean $\mathcal{M}_2$ $f$ is strongly concave in $y$ | Stationary point of $\max_y f$ / GDA |
| Han et al. (2023) | Complete $\mathcal{M}_1, \mathcal{M}_2$ $(x, y) \mapsto \|\mathrm{grad}_x f\|_x^2 + \|\mathrm{grad}_y f\|_y^2$ is Polyak-Łojasiewicz | Stationary point of $f$ / Hamiltonian |
| This work | $f$ is twice continuously differentiable | differential equilibrium / GDA,SGA |

Section 2 reviews the definition of DSE on Riemannian manifold, which includes DNE as a special case. We then analyze the local convergence of simultaneous min-max algorithms to DSE and DNE, in which the variables $(x, y)$ are updated simultaneously at each iteration. In Section 3.1, we adopt the classical Ostrowski theorem on manifold to analyze the local convergence of deterministic simultaneous algorithms to a fixed point. The problem is reduced to analyze the eigenvalues of a Jacobian matrix in a Euclidean local coordinate, which is at the heart of analyzing various Euclidean min-max algorithms (Daskalakis & Panageas, 2018; Azizian et al., 2020; Fiez & Ratliff, 2021; Zhang et al., 2022; Li et al., 2022; de Montbrun & Renault, 2022). In Section 3.2, sufficient conditions on $\tau$ are given to ensure the local convergence of $\tau$-GDA to DSE or DNE, where the learning rate ratio $\tau$ is used to adjust $x$ and $y$ at two different learning rates.

One issue of $\tau$-GDA is its slow convergence rate when there are strong rotational dynamics (Li et al., 2022). This is a well-known phenomenon near a Nash equilibrium in bilinear games due to the competitive nature between two players. To improve the convergence, first-order methods such as extra-gradient and optimistic mirror descent are often used to correct the gradient direction of $\tau$-GDA. Recently, these methods have been extended to Riemannian manifold (Zhang et al., 2023; Hu et al., 2023; Wang et al., 2023). However, their computations rely on exponential map and parallel transport which can be costly (Absil et al., 2008). In Section 3.3, we develop an algorithm $\tau$-SGA to address the same rotational problem based on auto-differentiation. It naturally extends the SGA algorithm (Gemp & Mahadevan, 2018; Balduzzi et al., 2018; Letcher et al., 2019) to Riemannian manifold by using a learning rate ratio $\tau$ as in $\tau$-GDA. We then analyze the convergence of an asymptotic variant of the deterministic $\tau$-SGA which is an approximation of $\tau$-SGA when $\tau$ is large.

In Section 4, we apply $\tau$-GDA and $\tau$-SGA to train orthogonal Wasserstein GANs (Müller et al., 2019). The underlying min-max problem is Riemannian since we shall impose Stiefel manifold constraints on $y$ to construct Lipschitz-continuous discriminators. This allows one to compute a lower bound of Wasserstein distance which can generalize in high dimension with a polynomial number of training samples (Arora et al., 2017; Biau et al., 2021). Section 5 concludes with some discussions. In summary, our main contributions are:

• Based on the notions of DSE and DNE on Riemannian manifold, we derive sufficient conditions in terms of the range of $\tau$ and the learning rate of $x$ to obtain a linear convergence rate of deterministic $\tau$-GDA to DSE and DNE. They rely on intrinsic quantities of Riemannian manifold.

• We develop an algorithm $\tau$-SGA to improve the convergence of $\tau$-GDA. In some cases, the asymptotic variant of deterministic $\tau$-SGA allows for a broader range of $\tau$ to be chosen to ensure its convergence to DSE with a faster rate. This indicates that $\tau$-SGA can achieve a faster local convergence compared to $\tau$-GDA.

• We apply the insights from the local convergence analysis to improve the training of orthogonal Wasserstein GANs. We find numerically that an improved convergence of stochastic $\tau$-GDA and $\tau$-SGA may also improve the generator quality on simple benchmarks. Results can be reproduced from a pytorch software `https://gitlab.com/sixin-zh/riemannian_minmax_algo`.

## 2 Differential equilibrium on Riemannian manifold

The notion of DNE on manifold was given in Ratliff et al. (2013). This section reviews the notions of DSE and DNE through their intrinsic and local coordinate definitions on Riemannian manifold. We then provide some simple examples to illustrate their difference. Additional mathematical notations used in this article are summarized in Appendix A.

### 2.1 Differential Stackelberg equilibrium (DSE)

We write $f \in C^2$ if it is twice continuously differentiable on the product manifold $\mathcal{M}_1 \times \mathcal{M}_2$. When $\mathcal{M}_1$ and $\mathcal{M}_2$ are Euclidean, the notion of DSE is defined based on the first and second order derivatives of $f$ at this equilibrium (Fiez et al., 2020). The next definition of DSE is a natural extension of this concept to Riemannian manifold.

**Definition 2.1.** *We say that $(x^*, y^*)$ is a DSE of $f \in C^2$ if*

$$grad_x f(x^*, y^*) = 0, \quad grad_y f(x^*, y^*) = 0, \tag{1}$$

$$- Hess_y f(x^*, y^*) \quad \text{p.d (abbrev. of positive definite)}, \tag{2}$$

$$[Hess_x f - grad_{yx}^2 f \cdot (Hess_y f)^{-1} \cdot grad_{xy}^2 f](x^*, y^*) \quad \text{p.d.} \tag{3}$$

In this definition, we rely on the following intrinsic quantities in the literature of Riemannian optimization (see Absil et al. (2008, Section 3.6,Definition 5.5.1) and Han et al. (2023, Section 2.1)):

- Riemannian gradient: $grad_x f(x, y) \in T_x \mathcal{M}_1, grad_x f(x, y) \in T_y \mathcal{M}_2$,
- Riemannian Hessian: $Hess_x f(x, y) : T_x \mathcal{M}_1 \to T_x \mathcal{M}_1, Hess_y f(x, y) : T_y \mathcal{M}_2 \to T_y \mathcal{M}_2$,
- Riemannian cross-gradient: $grad_{xy}^2 f(x, y) : T_x \mathcal{M}_1 \to T_y \mathcal{M}_2, grad_{yx}^2 f(x, y) : T_y \mathcal{M}_2 \to T_x \mathcal{M}_1$.

Here we denote the tangent space of $\mathcal{M}_1$ at $x \in \mathcal{M}_1$ by $T_x \mathcal{M}_1$ (resp. $\mathcal{M}_2$ at $y \in \mathcal{M}_2$ by $T_y \mathcal{M}_2$). The condition (1) means that $(x^*, y^*)$ is a critical point of $f$. Note that the eigenvalues of $Hess_x f(x^*, y^*)$ and $Hess_y f(x^*, y^*)$ depend on the Riemannian metric on $\mathcal{M}_1 \times \mathcal{M}_2$. However, the notion of DSE does not depend on the choice of Riemannian metric, due to the following known fact.

**Fact:** The notion of DSE in Definition 2.1 can be equivalently defined in a local coordinate chart which does not depend on the choice of Riemannian metric.

To make this point clear, we take a local coordinate chart $(O_1 \times O_2, \varphi_1 \times \varphi_2)$ around $(x^*, y^*) \in \mathcal{M}_1 \times \mathcal{M}_2$ (see Absil et al. (2008, Section 3.1,3.2)). We the rewrite the function $f(x, y)$ on $O_1 \times O_2$ using this chart by

$$\bar{f}(u_1, u_2) = f(\varphi_1^{-1}(u_1), \varphi_2^{-1}(u_2)).$$

In Appendix B, we verify the equivalence between (1)-(3) and the following conditions (4)-(6):

$$\partial_{u_1} \bar{f}(u_1^*, u_2^*) = 0, \quad \partial_{u_2} \bar{f}(u_1^*, u_2^*) = 0, \tag{4}$$

$$- \partial_{u_2 u_2}^2 \bar{f}(u_1^*, u_2^*) \quad \text{p.d}, \tag{5}$$

$$\left[ \partial_{u_1 u_1}^2 \bar{f} - \partial_{u_2 u_1}^2 \bar{f} \cdot (\partial_{u_2 u_2}^2 \bar{f})^{-1} \cdot \partial_{u_1 u_2}^2 \bar{f} \right] (u_1^*, u_2^*) \quad \text{p.d.} \tag{6}$$

We see that the conditions (4)-(6) do not depend on the choice of Riemannian metric, therefore Definition 2.1 still holds if the Riemannian metric is changed on $\mathcal{M}_1 \times \mathcal{M}_2$.

It is known that a DSE is a local minimax point (Jin et al., 2020). The conditions (4)-(6) imply that $(u_1^*, u_2^*) = (\varphi_1(x^*), \varphi_2(y^*))$ is a local minimax point of $\bar{f}$. Furthermore, from the implicit function theorem on Riemannian manifold given in Appendix C, $(x^*, y^*)$ is a local minimax point of $f$ in the following sense: there exists an open subset $U_1 \times U_2$ of $\mathcal{M}_1 \times \mathcal{M}_2$, which includes $(x^*, y^*)$ and on which there is a unique function $h : U_1 \to U_2$, such that the following holds

$$f(x^*, y) \leq f(x^*, y^*) \leq \max_{y' \in U_2} f(x, y') = f(x, h(x)), \quad \forall (x, y) \in U_1 \times U_2.$$

The set $\{(x, h(x)) | x \in U_1\}$ is sometimes called the ridge near a DSE (Wang et al., 2020).

## 2.2 DIFFERENTIAL NASH EQUILIBRIUM (DNE) AND EXAMPLES

In game theory, one is often interested in finding a Nash equilibrium since it maintains a symmetry between the role of the players. When $\mathcal{M}_1$ and $\mathcal{M}_2$ are Euclidean, it is also called "strongly local min-max point" (Daskalakis & Panageas, 2018, Definition 1.6).

The notion of DNE was introduced in Ratliff et al. (2013)[Definition 3] through a local coordinate chart. This is equivalent to the following intrinsic definition:

**Definition 2.2.** *We say that $(x^*, y^*)$ is a DNE of $f \in C^2$ if*

$$grad_x f(x^*, y^*) = 0, \quad grad_y f(x^*, y^*) = 0, \tag{7}$$

$$- Hess_y f(x^*, y^*) \quad p.d., \quad Hess_x f(x^*, y^*) \quad p.d. \tag{8}$$

From the definition, it is clear that a DNE is a DSE. We remark that this concept is defined locally, and therefore it is different to the global Nash-type equilibria on manifold in Kristály (2014).

### 2.2.1 EXAMPLE 1: DSE

Consider $f(x, y) = \langle y, Ax - b \rangle$, with $x \in \mathcal{M}_1 = \mathbb{R}^{d_1}$ and $y \in \mathcal{M}_2 = S^{d_2}$. The manifold $S^{d_2}$ is the unit sphere embedded in $\mathbb{R}^{d_2+1}$, endowed with the Euclidean metric $\langle \cdot, \cdot \rangle$ on $\mathbb{R}^{d_2+1}$. Let $A^+$ denote the pseudo-inverse of $A \in \mathbb{R}^{(d_2+1) \times d_1}$. We next construct a scenario where DSE exists.

**Proposition 2.1.** *Assume $b \notin Range(A)$, $Ker(A) = \{0\}$. Let $x^* = A^+ b$, $y^* = \frac{Ax^* - b}{\|Ax^* - b\|}$, then $(x^*, y^*)$ is a DSE of the $f$ in Example 1.*

The proof is given in Appendix D. Since $\mathcal{M}_1$ is Euclidean, it is clear that $\partial^2_{xx} f(x^*, y^*) = 0$. This implies that the $(x^*, y^*)$ is not DNE. But each eigenvalue of $\mathbf{A} = -Hess_y f(x^*, y^*)$ equals to $\|Ax^* - b\| > 0$, since from the proof $\langle \mathbf{A}[\eta^*], \eta^* \rangle = \|Ax^* - b\| \|\eta^*\|^2$ for any $\eta^* \in T_{y^*} \mathcal{M}_2$.

### 2.2.2 EXAMPLE 2: DSE

Consider $f(x, y) = \langle y, Ax - b \rangle - \frac{\kappa}{2} \|Ax\|^2$, with $x \in \mathcal{M}_1 = \mathbb{R}^{d_1}$ and $y \in \mathcal{M}_2 = S^{d_2}$. Compared to the $f$ in Example 1, we add a quadratic function of $Ax$ with a curvature parameter $\kappa > 0$. We next show that if $\kappa$ is close to zero, we can still find a DSE near the DSE in Proposition 2.1.

**Proposition 2.2.** *Assume $b \notin Range(A)$, $Ker(A) = \{0\}$. There exists $\kappa_0 > 0$ such that for any $0 < \kappa < \kappa_0$, there is a number $c$ close to 1, $x^* = cA^+ b$ and $y^* = \frac{Ax^* - b}{\|Ax^* - b\|}$, so that $(x^*, y^*)$ is a DSE of the $f$ in Example 2.*

The proof is given in Appendix E. From the proof, we have $\mathbf{C} = \partial^2_{xx} f(x, y) = -\kappa A^{\mathsf{T}} A$. The spectral radius of $\mathbf{C}$ equals to its operator norm $\|\mathbf{C}\| = \kappa \|A^{\mathsf{T}} A\| > 0$. As in Example 1, all the eigenvalues of $\mathbf{A} = -Hess_y f(x^*, y^*)$ equal to $\|Ax^* - b\| = \|cAA^+ b - b\|$. These quantities will be used in Section 3 to analyze the local convergence of min-max algorithms.

### 2.2.3 EXAMPLE 3: DNE

Consider $f(x, y) = \frac{1}{2} \|Ax + y - b\|^2$ with $x \in \mathcal{M}_1 = \mathbb{R}^{d_1}$ and $y \in \mathcal{M}_2 = S^{d_2}$. $\mathcal{M}_2$ is the same embedded sub-manifold of $\mathbb{R}^{d_2+1}$ as above. We next provide a sufficient condition on the existence of DNE. The proof is given in Appendix F.

**Proposition 2.3.** *Assume $b \notin Range(A)$, $Ker(A) = \{0\}$. Let $x^* = A^+ b$, $y^* = \frac{Ax^* - b}{\|Ax^* - b\|}$, then $(x^*, y^*)$ is a DNE of the $f$ in Example 3.*

## 3 SIMULTANEOUS MIN-MAX ALGORITHMS FOR DIFFERENTIAL EQUILIBRIUM

Simultaneous gradient-based min-max algorithms such as GDA and SGA are often used to find local Nash equilibrium when $\mathcal{M}_1$ and $\mathcal{M}_2$ are Euclidean (Daskalakis & Panageas, 2018; Letcher et al., 2019). Similar to GDA, we extend the SGA algorithm to Riemannian manifold with two-time scale update (Heusel et al., 2017) using either deterministic or stochastic gradients. Section 3.1 reviews a classical result of fixed point theorem. Based on this theorem, we then focus on a deterministic analysis of the local convergence of these algorithms to DSE and DNE.

### 3.1 LOCAL CONVERGENCE OF DETERMINISTIC SIMULTANEOUS ALGORITHMS

We use the classical Ostrowski theorem to analyze the local convergence of simultaneous deterministic algorithms on manifold. Each algorithm is defined by an update rule which does not change over iteration. This theorem provides a sufficient condition for the linear convergence rate of an algorithm to a fixed point, based on the spectral radius of the update rule's Jacobian matrix at the fixed point.

A deterministic simultaneous algorithm is defined by two vector fields $(x, y) \mapsto \xi_1(x, y) \in T_x \mathcal{M}_1$, $(x, y) \mapsto \xi_2(x, y) \in T_y \mathcal{M}_2$ on $\mathcal{M}_1$ and $\mathcal{M}_2$, and a suitable choice of manifold retractions (see Absil et al. (2008, Section 4.1)). Initialized at a point $(x(0), y(0))$ on $\mathcal{M}_1 \times \mathcal{M}_2$, the algorithm generates a sequence $(x(t+1), y(t+1)) = \mathbf{T}(x(t), y(t))$, through an update rule $\mathbf{T} : \mathcal{M}_1 \times \mathcal{M}_2 \to \mathcal{M}_1 \times \mathcal{M}_2$ of the following form,
$$\mathbf{T}(x, y) = (\mathcal{R}_{1,x}(\xi_1(x, y)), \mathcal{R}_{2,y}(\xi_2(x, y))),$$
where $\mathcal{R}_{1,x} : T_x \mathcal{M}_1 \to \mathcal{M}_1$ (resp. $\mathcal{R}_{2,y} : T_y \mathcal{M}_2 \to \mathcal{M}_2$) denotes the restriction of a retraction $\mathcal{R}_1$ at $x \in \mathcal{M}_1$ (resp. retraction of $\mathcal{R}_2$ at $y \in \mathcal{M}_2$) For example, on the $\mathcal{M}_1 = \mathbb{R}^{d_1}$ and $\mathcal{M}_2 = S^{d_2}$ in Section 2.2, we will take $\mathcal{R}_{1,x}(\delta) = x + \delta$ for $\delta \in \mathbb{R}^{d_1}$, and $\mathcal{R}_{2,y}(\eta)$ to be the projection of the vector $y + \eta$ in $\mathbb{R}^{d_2+1}$ to the sphere $\mathcal{M}_2$.

We say that $(x^*, y^*) \in \mathcal{M}_1 \times \mathcal{M}_2$ is a fixed point of $\mathbf{T}$ if it is a critical point of the vector fields $\xi_1$ and $\xi_2$. We next define what it means for an update rule $\mathbf{T}$ to be locally convergent to $(x^*, y^*)$. It implies that it is also a point of attraction of $\mathbf{T}$ (Ortega & Rheinboldt, 1970, Definition 10.1.1).

**Definition 3.1** (Locally convergent with (linear) rate $\rho \in (0, 1)$). *Let $(x^*, y^*)$ be a fixed point of $\mathbf{T}$. For any $\epsilon \in (0, 1 - \rho)$, there exists a local stable region $S_\delta \subset \mathcal{M}_1 \times \mathcal{M}_2$, which contains a geodesically convex open set (and homeomorphic to a ball) containing $(x^*, y^*)$, such that started from $(x(0), y(0)) \in S_\delta$, we have $(x(t), y(t)) \in S_\delta, \forall t \geq 1$. Furthermore, let $d(t)$ be the Riemannian distance between $(x(t), y(t))$ and $(x^*, y^*)$, then there exists a constant $C$ s.t. $d(t) \leq C(\rho + \epsilon)^{t+1} d(0), \forall t \geq 0$.*

We next give a sufficient condition of $\mathbf{T}$ to achieve the local linear convergence. It is based on the spectral radius of the following linear transformation on $T_{x^*} \mathcal{M}_1 \times T_{y^*} \mathcal{M}_2$,

$$\mathbf{DT}^* = \mathbf{T}'(x^*, y^*) = \begin{pmatrix} I + \nabla_x \xi_1(x^*, y^*) & D_y \xi_1(x^*, y^*) \\ D_x \xi_2(x^*, y^*) & I + \nabla_y \xi_2(x^*, y^*) \end{pmatrix}. \tag{9}$$

Here $D_x$ and $\nabla_x$ (resp. $D_y$ and $\nabla_y$) denote the differential operator and the Riemannian connection on $\mathcal{M}_1$ (resp. on $\mathcal{M}_2$). Note that $\mathbf{DT}^*$ equals to the tangent map of $\mathbf{T}$ at $(x^*, y^*)$, no matter how one chooses the retraction $\mathcal{R}_1$ and $\mathcal{R}_2$ (c.f. Appendix G.1). When $\mathcal{M}_1$ and $\mathcal{M}_2$ are Euclidean, it is the Jacobian matrix of $\mathbf{T}$ at $(x^*, y^*)$.

**Theorem 3.1** (Ostrowski Theorem on manifold). *Let $(x^*, y^*)$ be a fixed point of $\mathbf{T}$. Assume that $\xi_1$ and $\xi_2$ are continuous on $\mathcal{M}_1 \times \mathcal{M}_2$, and they are differentiable at $(x^*, y^*)$ such that $\rho(\mathbf{DT}^*) < 1$, then $\mathbf{T}$ is locally convergent to $(x^*, y^*)$ with rate $\rho(\mathbf{DT}^*)$.*

This result is proved in Ortega & Rheinboldt (1970, Section 10.1.3) when $\mathcal{M}_1$ and $\mathcal{M}_2$ are Euclidean. The proof idea can be readily extended to the manifold case. In the statement of Theorem 3.1, we add an assumption on the continuity to $\xi_1$ and $\xi_2$ to ensure a non-empty local stable region $S_\delta$. To make the article self-contained, we provide a proof of this theorem in Appendix G. This proof does not give an explicit way to construct the local stable region $S_\delta$. Therefore it is unclear what a good initialization entails. To obtain a precise size of $S_\delta$, extra assumptions on $f$ would be needed.

Theorem 3.1 has a local nature as the convergence rate $\rho(\mathbf{DT}^*)$ does not depend on any global manifold property. One can also use the stronger operator norm assumption $\|\mathbf{DT}^*\| < 1$ in Boumal (2023, Theorem 4.19) to obtain a similar local convergence result.

### 3.2 SIMULTANEOUS GRADIENT-DESCENT-ASCENT ALGORITHM ($\tau$-GDA)

The $\tau$-GDA algorithm uses the Riemannian gradients $\text{grad}_x f(x, y)$ and $\text{grad}_y f(x, y)$ to update $x$ and $y$ simultaneously. The local convergence of deterministic $\tau$-GDA to DSE and DNE is well-studied in Euclidean space (Daskalakis & Panageas, 2018; Jin et al., 2020; Fiez & Ratliff, 2021; Li et al., 2022). This section extends these results to Riemannian manifold. We obtain a sharp lower-bound of $\tau$ for $\tau$-GDA to be locally convergent to DSE. It is based on a refinement of the spectral analysis in Euclidean space (Li et al., 2022).

$\tau$**-GDA algorithm**   In the deterministic setting, the update rule $\mathbf{T}$ of $\tau$-GDA is determined by

$$\xi_1(x, y) = -\gamma \text{grad}_x f(x, y), \quad \xi_2(x, y) = \tau \gamma \text{grad}_y f(x, y). \tag{10}$$

Note that $\gamma > 0$ and we use a ratio $\tau > 0$ to adjust the learning rate (step size) of the Riemannian gradients. The deterministic $\tau$-GDA can be readily extended to stochastic $\tau$-GDA by using an unbiased estimation of the Riemannian gradients (Jordan et al., 2022; Huang & Gao, 2023).

**Local convergence of deterministic $\tau$-GDA**   Based on Theorem 3.1, we are ready to study the local convergence of $\tau$-GDA (defined by (10)) to DSE and DNE. From the definition of Riemannian Hessian and cross-gradient, we rewrite $\mathbf{DT}^* = I + \gamma \mathbf{M}_g$ using the following linear transform

$$\mathbf{M}_g = \begin{pmatrix} -\mathbf{C} & -\mathbf{B} \\ \tau \mathbf{B}^\intercal & -\tau \mathbf{A} \end{pmatrix} = \begin{pmatrix} -\text{Hess}_x f(x^*, y^*) & -\text{grad}_{yx}^2 f(x^*, y^*) \\ \tau \text{grad}_{xy}^2 f(x^*, y^*) & \tau \text{Hess}_y f(x^*, y^*) \end{pmatrix}. \tag{11}$$

For a Hurwitz-stable linear transform $\mathbf{M}$ whose eigenvalues all have strictly negative real part, we write $\gamma^\bullet(\mathbf{M}) = -2 \max_k \frac{\text{Re}(\lambda_k(\mathbf{M}))}{|\lambda_k(\mathbf{M})|^2}$. It computes an upper bound of $\gamma$ such that $\rho(I + \gamma \mathbf{M}) < 1$.

Let $L_g = \max(\|\mathbf{A}\|, \|\mathbf{B}\|, \|\mathbf{C}\|)$ and $\mu_g = \min(L_g, \lambda_{min}(\mathbf{C} + \mathbf{B}\mathbf{A}^{-1}\mathbf{B}^\intercal))$.

**Theorem 3.2.** *Assume $(x^*, y^*)$ is a DSE of $f \in C^2$. If $\tau > \frac{\|\mathbf{C}\|}{\lambda_{min}(\mathbf{A})}$ and $\gamma \in (0, \gamma^\bullet(\mathbf{M}_g))$, $\tau$-GDA is locally convergent to $(x^*, y^*)$ with rate $\rho(I + \gamma \mathbf{M}_g)$. Furthermore, if $\tau \geq \frac{2L_g}{\lambda_{min}(\mathbf{A})}$ and $\gamma = \frac{1}{4\tau L_g}$, the rate is at most $1 - \frac{\mu_g}{16\tau L_g}$.*

The proof is given in Appendix H. This result is an extension of the Euclidean space result in Li et al. (2022, Theorem 4.2) (without assuming $\mathbf{M}_g$ being diagonalizable). When $\mathcal{M}_1$ and $\mathcal{M}_2$ are Euclidean, the range $\{\tau \in \mathbb{R}_+ | \tau > \|\partial_{xx}^2 f(x^*, y^*)\| / \lambda_{min}(-\partial_{yy}^2 f(x^*, y^*))\}$ in Theorem 3.2 is sharp as one can construct a counter-example (Li et al., 2022, Theorem 4.1) to show that if $\tau$ is outside this range, the spectral radius of the Jacobian matrix (9) is strictly larger than one for any $\gamma > 0$ (see also a discussion in Li et al. (2022, Remark 2)).

Theorem 3.2 can be readily applied to analyze the local convergence of $\tau$-GDA to DNE. However, we can obtain a broader range of $\tau$. The next result generalizes local convergence properties of $\tau$-GDA to DNE from Euclidean space to Riemannian manifold.

**Theorem 3.3** (Jin et al. (2020); Zhang et al. (2022)). *Assume $(x^*, y^*)$ is a DNE of $f \in C^2$ and $\bar{\mu}_g = \min(\lambda_{min}(\mathbf{A}), \lambda_{min}(\mathbf{C}))$. If $\tau > 0$ and $\gamma \in (0, \gamma^\bullet(\mathbf{M}_g))$, $\tau$-GDA is locally convergent to $(x^*, y^*)$ with rate $\rho(I + \gamma \mathbf{M}_g)$. Furthermore, if $\tau = 1$ and $\gamma = \frac{\bar{\mu}_g}{2L_g^2}$, the rate is at most $1 - \frac{\bar{\mu}_g^2}{4L_g^2}$.*

The proof is given in Appendix I. The common reason that we can obtain such extensions in Theorem 3.2 and 3.3 is that the spectral analysis of the matrix $\mathbf{M}_g$ is reduced to a similar matrix $M_g$ in a local coordinate (see (42)) and the spectral properties of each intrinsic term $(\mathbf{A}, \mathbf{B}, \mathbf{C})$ in $\mathbf{M}_g$ are the same as those in $M_g$. We can therefore use existing Euclidean results to derive a sufficient condition to control the spectral radius of $M_g$, and then identify an equivalent condition in terms of $\mathbf{M}_g$.

### 3.3   SYMPLECTIC GRADIENT-ADJUSTMENT METHOD ($\tau$-SGA)

In Euclidean space, SGA modifies the update rule of GDA to avoid strong rotational dynamics near a fixed point (Gemp & Mahadevan, 2018). We apply this idea to $\tau$-GDA and extend it to Riemannian manifold. We then study its local convergence to DSE when $\tau$ is large.

$\tau$**-SGA algorithm**   SGA adjusts a vector field $\xi$ using an orthogonal vector field which is constructed from the anti-symmetric part of the Jacobian matrix of $\xi$. In $\tau$-GDA, $\xi(x, y) = (-\delta(x, y), \tau\eta(x, y))$ where $\delta(x, y) = \text{grad}_x f(x, y)$ and $\eta(x, y) = \text{grad}_y f(x, y)$. Based on this idea, the adjustment of $\tau$-SGA depends on the Riemannian cross-gradients $\tilde{\mathbf{B}}(x, y) = \text{grad}_{yx}^2 f(x, y)$ and $\tilde{\mathbf{B}}^\intercal(x, y) = \text{grad}_{xy}^2 f(x, y)$, which is summarized in the following update rule (with a hyper-parameter $\mu \in \mathbb{R}$),

$$\xi_1(x, y) = -\gamma \left( \delta(x, y) + \mu \frac{(\tau + 1)\tau}{2} \tilde{\mathbf{B}}(x, y)[\eta(x, y)] \right), \tag{12}$$

$$\xi_2(x, y) = \gamma \left( \tau\eta(x, y) - \mu \frac{\tau + 1}{2} \tilde{\mathbf{B}}^\intercal(x, y)[\delta(x, y)] \right). \tag{13}$$

In Appendix J, we provide the derivation of this update rule. The next proposition shows that the adjusted direction $(-\tau\tilde{\mathbf{B}}[\eta], -\tilde{\mathbf{B}}^\intercal[\delta])$ is orthogonal to $\xi = (-\delta, \tau\eta)$.

**Proposition 3.1.** *For any $(x, y) \in \mathcal{M}_1 \times \mathcal{M}_2$, we have*

$$\langle \tau\tilde{\mathbf{B}}(x,y)[\eta(x,y)], \delta(x,y)\rangle_x + \langle \tilde{\mathbf{B}}^\intercal(x,y)[\delta(x,y)], -\tau\eta(x,y)\rangle_y = 0.$$

The proof is given in Appendix J.1. In the original SGA method, the orthogonality is essential to make it compatible to potential and Hamiltonian game dynamics. This proposition implies that such compatibility still makes sense for $\tau$-SGA. In Appendix L, we discuss how to perform deterministic and stochastic gradient adjustment using auto-differentiation when $\mathcal{M}_1$ and $\mathcal{M}_2$ are Euclidean embedded sub-manifolds (Absil et al., 2008, Chapter 3.3).

**Local convergence of deterministic $\tau$-SGA to DSE: asymptotic analysis**  We study the local convergence of $\tau$-SGA with deterministic gradients in an asymptotic regime where $\tau \to \infty$. In this regime, to make the term $\tilde{\mathbf{B}}[\eta]$ comparable to the term $\delta$ in (12), we re-parameterize $\mu = \theta\frac{2}{\tau(\tau+1)} \sim \frac{1}{\tau^2}$. As a consequence, $\mu\frac{\tau+1}{2} \sim \frac{1}{\tau}$ and the update rule in (13) can be approximated by the $\xi_2$ in (10).

The next theorem analyzes the local convergence of Asymptotic $\tau$-SGA, which is an approximation of $\tau$-SGA, whose $\xi_1$ (resp. $\xi_2$) is defined by (12) (resp. (10)). In order to apply Theorem 3.1, it is necessary to verify that the corresponding $\xi_1$ is differentiable at $(x^*, y^*)$. This is indeed true since $\tilde{\mathbf{B}}(x, y)$ is continuous at $(x^*, y^*)$ and $\eta(x^*, y^*) = 0$. The following linear transform plays a key role in the asymptotic analysis since it gives an approximation of the $\mathbf{DT}^*$ in $\tau$-SGA by $I + \gamma\mathbf{M}_s$,

$$\mathbf{M}_s = \begin{pmatrix} -\mathbf{C} & -\mathbf{B} \\ \tau\mathbf{B}^\intercal & -\tau\mathbf{A} \end{pmatrix} + \theta\begin{pmatrix} -\mathbf{BB}^\intercal & \mathbf{BA} \\ 0 & 0 \end{pmatrix}. \tag{14}$$

Let $L_s = \max(\|\mathbf{A}\|, \|\mathbf{B}\|, \|\mathbf{C} + \theta\mathbf{BB}^\intercal\|)$ and $\mu_s = \min(L_s, \lambda_{min}(\mathbf{C} + \mathbf{BA}^{-1}\mathbf{B}^\intercal))$.

**Theorem 3.4.** *Assume $(x^*, y^*)$ is a DSE of $f \in C^2$. If $\tau > \min\left(\frac{\|\mathbf{C}\|}{\lambda_{min}(\mathbf{A})}, \frac{\|\mathbf{C}+\theta\mathbf{BB}^\intercal\|}{\lambda_{min}(\mathbf{A})}\right)$, $\mu = \theta\frac{2}{\tau(\tau+1)}$ with $0 \leq \theta \leq \frac{1}{\lambda_{max}(\mathbf{A})}$, and $\gamma \in (0, \gamma^\bullet(\mathbf{M}_s))$, Asymptotic $\tau$-SGA is locally convergent to $(x^*, y^*)$ with rate $\rho(I + \gamma\mathbf{M}_s)$. Furthermore, if $\tau \geq \frac{2L_s}{\lambda_{min}(\mathbf{A})}$ and $\gamma = \frac{1}{4\tau L_s}$, the rate is at most $1 - \frac{\mu_s}{16\tau L_s}$.*

The proof is given in Appendix K. Contrary to Theorem 3.2, the valid range of $\tau$ depends also on the choice of $\theta$. Indeed, if the spectral radius of $\mathbf{C}$ is larger than that of $\mathbf{C} + \theta\mathbf{BB}^\intercal$ with a suitable choice of $\theta$, a broader range of $\tau$ could be used in $\tau$-SGA compared to $\tau$-GDA. For a DSE $(x^*, y^*)$ which is not DNE (i.e. $\mathbf{C}$ is not p.d.), such a choice of $\theta$ can be possible. Furthermore, we obtain a non-trivial improvement on the convergence rate in theory as it significantly improves the rate of the extra-gradient method in Euclidean space (Li et al., 2022, Theorem 5.4) when $L_s = \mu_s = \|\mathbf{C}+\theta\mathbf{BB}^\intercal\| < L_g = \mu_g = \|\mathbf{C}\|$. In this case, we could choose a smaller $\tau_s = \frac{2L_s}{\lambda_{min}(\mathbf{A})}$ and a larger $\gamma_s = \frac{1}{4\tau_s L_s} = \frac{\lambda_{min}(\mathbf{A})}{8L_s^2}$ in Asymptotic $\tau$-SGA to achieve a faster rate $1 - \frac{1}{16\tau_s}$, compared to the rate $1 - \frac{1}{16\tau_g}$ using a larger $\tau_g = \frac{2L_g}{\lambda_{min}(\mathbf{A})}$ and a smaller $\gamma_g = \frac{1}{4\tau_g L_g}$ in $\tau$-GDA.

We next illustrate the convergence results using a numerical example, to show that $\tau$-SGA can indeed converge much faster than $\tau$-GDA to DSE when there are strong rotational forces in its dynamics. In Figure 1, we compare the local convergence rate of $\tau$-GDA and $\tau$-SGA in Example 2 of Section 2.2, where $A = [1; 1; 1] \in \mathbb{R}^{3\times1}$, $b = [1; 1; 0.99] \in \mathbb{R}^3$ and $\kappa = 0.1$. In this case, we find numerically that $x^* = 0.9975$. The corresponding optimal $y^* = (Ax^* - b)/\|Ax^* - b\|$. The initial point of each algorithm is set to be $x = A^+b = 0.9967$, $y = (Ax - b)/\|Ax - b\|$, which is close to the DSE $(x^*, y^*)$. In Figure 1(a), we study the range of $\tau$ for the convergence of $\tau$-GDA with $\gamma = 0.001/\tau$. According to Theorem 3.2 and Proposition 2.2, a valid range of $\tau$ should be larger than $\frac{\max_k |\lambda_k(\mathbf{C})|}{\lambda_{min}(\mathbf{A})} = \kappa\|A^\intercal A\|/\|Ax^* - b\| \approx 36.18$. Figure 1(a) shows that when $\tau = 30$, $\tau$-GDA can slowly diverge. When $\tau = 50$, $\tau$-GDA converges slowly to the value $f(x^*, y^*) = -0.141$. This is a convergence dilemma of $\tau$-GDA: to achieve the local convergence, $\tau$ needs to be large, but the rate can be very slow. In Figure 1(b) and (c), we study $\tau$-SGA with $\theta = 0.15$, using the same learning rate $\tau\gamma$ for $y$ as $\tau$-GDA. Figure 1(b) shows that the gap between Asymptotic $\tau$-SGA and $\tau$-SGA is not so

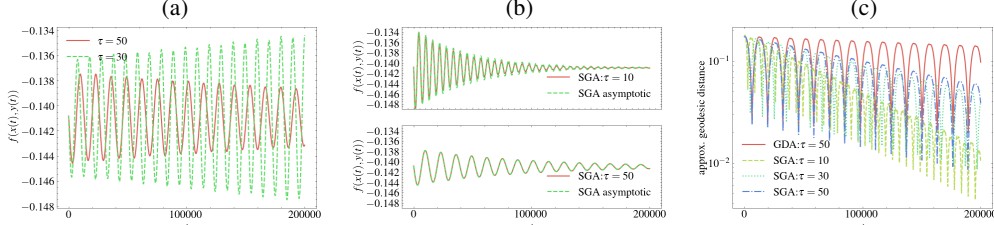

Figure 1: Evolution of $f(x(t), y(t))$ and an approximate geodesic distance between $(x(t), y(t))$ to $(x^*, y^*)$ as a function of the iteration $t$ using deterministic $\tau$-GDA and $\tau$-SGA for Example 2. (a): $\tau$-GDA with $\tau = 30$ vs. $\tau = 50$. (b): $\tau$-SGA vs. Asymptotic $\tau$-SGA with $\tau = 10$ and $\tau = 50$. (c): $\tau$-GDA at $\tau = 50$ vs. $\tau$-SGA at $\tau \in \{10, 30, 50\}$.

big even if $\tau = 10$. Therefore in this specific example, Theorem 3.4 provides a valid picture on the behavior of $\tau$-SGA when $\tau$ is large enough. We find numerically that $\frac{\max_k |\lambda_k(\mathbf{C} + \theta \mathbf{B} \mathbf{B}^{\mathsf{T}})|)|}{\lambda_{min}(\mathbf{A})} \approx 16.7$, which suggests that $\tau$-SGA can be locally convergent with a smaller $\tau$ compared to $\tau$-GDA. Figure 1(c) confirms this analysis and it also shows that a faster linear rate can be achieved by $\tau$-SGA. Surprisingly, the approximate geodesic distance $|x(t) - x^*| + \arccos(y(t)^{\mathsf{T}} y^*)$ (an approximation of the Riemmanian distance $d(t)$) quickly converges towards 0 even if $\tau = 10$.

The local convergence of $\tau$-SGA to DNE in Euclidean space is analyzed in Letcher et al. (2019, Theorem 10) when $\tau = 1$. Using the same proof idea as Theorem 3.3, one can extend this result to Riemmanian manifold. However, when $\tau \neq 1$, a DNE is not necessarily a stable fixed point of the vector field $\xi = (-\delta, \tau \eta)$, c.f Letcher et al. (2019, Definition 4). It is thus unclear if there is a non-trivial range of $\mu$ such that for any $\tau > 0$, $\tau$-SGA is locally convergent to DNE.

## 4   APPLICATION TO WASSERSTEIN GAN

The local convergence analysis in Section 3 shows that a larger $\tau$ is sometimes needed to ensure the local convergence of $\tau$-GDA to DSE compared to $\tau$-SGA. However, in practice, it is numerically hard to validate this theory in GAN. In this section, we study the training of orthogonal Wasserstein GAN using stochastic $\tau$-GDA and $\tau$-SGA. We construct two non-standard examples with the following goals: (1) Illustrate a similar local convergence dilemma near DSE faced by $\tau$-GDA as $\tau$ varies on both synthetic and real datasets. (2) Show that the correction term in $\tau$-SGA plays a key role to improve $\tau$-GDA when $\tau$ is small, leading to a local convergence. (3) Propose simple and clean benchmarks such that theoretical study of smooth Riemannian non-convex non-concave games from GAN could be further developed.

### 4.1   SETUP OF ORTHOGONAL WASSERSTEIN GAN

In Wasserstein GAN, we are interested in the following min-max problem
$$f(x, y) = \mathbb{E}(D_y(\phi_{data})) - \mathbb{E}(D_y(\phi_x))$$
such that $f \in C^2$. The idea of GAN is to approximate the probability distribution of $\phi_{data} \in \mathbb{R}^d$ using a generator $G_x$ parameterized by $x \in \mathcal{M}_1$. The parameter $x$ is optimized so that the distribution of $\phi_x$ approximates that of $\phi_{data}$ through the feedback of a discriminator $D_y : \mathbb{R}^d \to \mathbb{R}$ parameterized by $y \in \mathcal{M}_2$. The expectations in $f$ are taken with respect to the random variables $\phi_{data}$ and $\phi_x$. To build orthogonal Wasserstein GAN (Müller et al., 2019), we assume that $\mathcal{M}_1$ is Euclidean and $\mathcal{M}_2$ is Riemannian. The manifold constraint on $y$ restricts the discriminator family $\{D_y\}_{y \in \mathcal{M}_2}$ to be a subset of 1-Lipschitz continuous functions, up to a constant scaling.

We next study two examples of $\phi_{data}$ built upon unconditional generator $G_x$. It transforms a random noise $Z \in \mathbb{R}^p$ to $\phi_x \in \mathbb{R}^d$ from a lower dimension $p < d$. Further details about $\phi_{data}$, $G_x$ and $D_y$ are given in Appendix M.

**Gaussian distribution by linear generator**   We want to model $\phi_{data} \sim \mathcal{N}(0, \Sigma)$ on dimension $d = 5$ using a PCA-like model with dimension $p = 4$. The covariance matrix $\Sigma$ is diagonal

with a small eigenvalue, i.e. $(1, 2^2, 3^2, 4^2, 0.01)$. We choose the generator $G_x(Z) = A_x Z$, where $A_x \in \mathbb{R}^{d \times p}$ and $Z$ is isotropic normal. For the discriminator, we use the Stiefel manifold $St(k, d)$ of $k$ orthogonal vectors on $\mathbb{R}^d$ to construct $D_y(\phi) = \langle v_y, \sigma(W_y \phi) \rangle$, where $W_y \in St(k, d)$, $v_y \in St(1, k)$ and $\sigma$ is a smooth non-linearity. We set $k = d = 5$. As $A_x$ is the only generator parameter, $x := (A_x) \in \mathcal{M}_1 = \mathbb{R}^{d \times p}$. The discriminator parameter $y := (W_y, v_y) \in \mathcal{M}_2 = St(k, d) \times St(1, k)$.

**Image modeling by DCGAN generator**   We aim to model images from the MNIST and Fashion-MNIST datasets ($d = 28 \times 28$) using a DCGAN generator ($p = 128$). The space $\mathcal{M}_1$ is Euclidean with dimension $d_1 = 1556673$. To simplify the CNN discriminators in WGAN-GP, we consider a hybrid Scattering CNN discriminator build upon the wavelet scattering transform to capture discriminative information in natural images (Bruna & Mallat, 2013; Oyallon et al., 2017). It has only one trainable layer and therefore it is more amenable to theoretical study,

$$D_y(\phi) = \langle v_y, \sigma(w_y \star P(\phi) + b_y) \rangle.$$

The scattering transform $P : \mathbb{R}^d \to \mathbb{R}^{I \times n \times n}$ is 1-Lipschitz-continuous and it has no trainable parameter. The scattering features $P(\phi) \in \mathbb{R}^{I \times n \times n}$ are computed from an image $\phi$ of size $\sqrt{d} \times \sqrt{d}$. We then apply an orthogonal convolutional layer to $P(\phi)$ (Cisse et al., 2017). It has the kernel orthogonality (Achour et al., 2022), by reshaping $w_y$ (size $J \times I \times k \times k$) into a matrix $W_y \in \mathbb{R}^{J \times (Ik^2)}$ such that $W_y \in St(J, Ik^2)$. The bias parameter $b_y \in \mathbb{R}^J$ and the output of the convolutional layer is reshaped into a vector in $\mathbb{R}^{JN^2}$. As in the Gaussian case, we further assume $v_y \in St(1, JN^2)$. In summary, $y := (W_y, b_y, v_y) \in \mathcal{M}_2 = St(J, Ik^2) \times \mathbb{R}^J \times St(1, JN^2)$.

**Initialization of algorithms**   To study the local convergence of stochastic $\tau$-GDA and $\tau$-SGA, we first obtain a reasonable solution of $(x, y)$ in terms of model quality using (alternating) $\tau$-GDA (see Appendix M.6). We then evaluate $\tau$-GDA and $\tau$-SGA initialized from this solution.

### 4.2   RESULTS ON GAUSSIAN DISTRIBUTION BY LINEAR GENERATOR

We study the local convergence of stochastic $\tau$-GDA across various $\tau \in \{1, 10, 100\}$. In Figure 2(a), we show how the function value $f(x(t), y(t))$ (estimated from 1000 training samples) changes over iteration. We observe that when $\tau = 1$, $f(x(t), y(t))$ has a huge oscillation, but when $\tau = 10$ and 100, it has a much smaller oscillation amplitude. To investigate the underlying reason, we compute in Figure 2(b), the evolution of an "angle" quantity every one thousand iterations. The angle at $(x, y)$ is defined as $\frac{\langle v_y, \delta(x,y) \rangle}{\|\delta(x,y)\|}$, where $\delta(x, y) = \mathbb{E}(\sigma(W_y \phi_{data})) - \mathbb{E}(\sigma(W_y \phi_x))$ is estimated from the same batch of samples during the training. When the angle is close to 1, it indicates that $v_y$ is solved to be optimal. We observe that when $\tau = 1$, the angle oscillates between positive and negative values, suggesting that $v_{y(t)}$ is detached from $\delta(x(t), y(t))$. Therefore the minimization of $f$ over $x$ does not get a good feedback to improve the model. This is confirmed in Figure 2(c), where we compute the EMD distance (Rubner et al., 2000) every one thousand iterations, between the empirical measure of $\phi_{data}$ and $\phi_{x(t)}$ (using 2000 validation samples). We observe that the EMD distance increases over $t$ when $\tau = 1$. On the contrary, it stays close to one when $\tau = 10$ and 100. When $\tau = 100$, we find that the covariance error $\|\Sigma - A_{x(t)} A_{x(t)}^\intercal\|$ stays around 0.2 over all iterations. This error is between the smallest eigenvalue of $\Sigma$ (which is 0.01, the PCA error) and the second smallest eigenvalue of $\Sigma$. Therefore to use a large enough $\tau$ is crucial to reduce model error, as it can ensure a positive angle to measure a meaningful distance between $\phi_x$ and $\phi_{data}$. This phenomenon is consistent with the local convergence of $\tau$-GDA to DSE in Figure 1(a).

Table 2: Last iteration measures of stochastic $\tau$-GDA and $\tau$-SGA on the Wasserstein GAN of MNIST. We report the $f$, angle, and FID scores computed at $(x(T), y(T))$. Only significant digits are reported with respect to the standard deviation (shown in $(\pm)$) (see Appendix M.7).

| | $\tau$-GDA | | | | $\tau$-SGA | | | | |
|---|---|---|---|---|---|---|---|---|---|
| $\tau$ | $f$ | angle | FID (train) | FID (val) | $T(10^5)$ | $f$ | angle | FID (train) | FID (val) |
| 5 | 0.13 | 0.5 | 13.38 $(\pm 0.02)$ | 16 | 1 | 0.013 | 0.4 | 6.65 $(\pm 0.04)$ | 8.7 $(\pm 0.1)$ |
| 10 | 0.013 | 0.47 $(\pm 0.02)$ | 7.0 | 9.1 $(\pm 0.1)$ | 3 | 0.012 | 0.3 | 6.2 | 8.3 $(\pm 0.09)$ |
| 20 | 0.0136 | 0.70 $(\pm 0.02)$ | 7.0 | 9.1 $(\pm 0.1)$ | 5 | 0.010 | 0.36 $(\pm 0.02)$ | 5.76 $(\pm 0.036)$ | 7.6 $(\pm 0.08)$ |

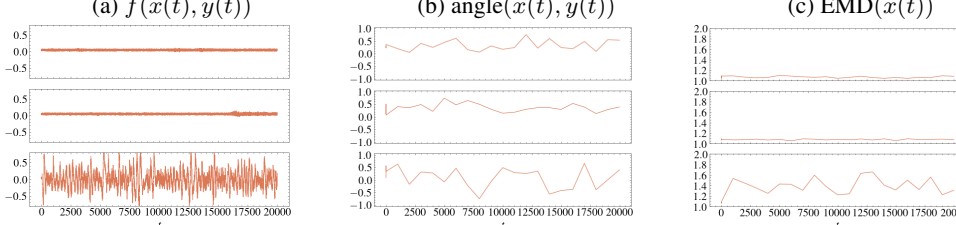

Figure 2: Evolution of $f(x(t), y(t))$, angle$(x(t), y(t))$, and the EMD$(x(t))$ distance as a function of the iteration $t$ using stochastic $\tau$-GDA for the Wasserstein GAN of Gaussian distribution. Top: $\tau = 100, \gamma = 0.0002$. Middle: $\tau = 10, \gamma = 0.002$, Bottom: $\tau = 1, \gamma = 0.02$.

### 4.3 Results on MNIST and Fashion-MNIST by DCGAN generator

We apply the insights from theoretical analysis to improve the convergence of both $\tau$-GDA and $\tau$-SGA, so as to obtain a good generative model $\phi_x$ (measured by the FID scores (Seitzer, 2020)).

In Table 2, we first report the performance of $\tau$-GDA by varying the choice of $\tau$. It is run for $T = 2 \times 10^4$ iterations with $\gamma = 0.1/\tau$. We find that at $\tau = 5$ even though the angle is positive, the value of $f$ and FID scores are much larger than their initial values. By investigating the evolution of $f$ over the iteration $t$, we indeed observe unstable dynamics with much stronger oscillation starting from $t = 5000$. This suggests that $\tau$-GDA does not have local convergence. This instability is improved when $\tau = 10$ or $\tau = 20$. We find that $\tau = 10$ can still result in some instability (see Figure 4 in Appendix N) if $T$ is made about ten times larger. When $\tau = 20$, $\tau$-GDA has a more stable convergence. We next study the performance of $\tau$-SGA by varying the total training iterations $T$, using a fixed $\gamma = 0.02, \tau = 5, \theta = 0.075$. In Table 2, we observe that as $T$ is increased, the value of $f$ and FID scores are decreased. This suggests a converging behavior of $\tau$-SGA, which is not the case in $\tau$-GDA at $\tau = 5$. It shows that the correction term in $\tau$-SGA plays a key role to improve the local convergence of $\tau$-GDA. We also perform a similar study on other datasets, which shows consistent conclusions (see Appendix N.2,N.3 and N.4).

In terms of computational efficiency, we find that $\tau$-GDA and $\tau$-SGA can reach similar FID (test) scores after a similar or smaller amount of training time (see Figure 4 and 7 in Appendix N). But sometimes $\tau$-SGA can be slower (see Figure 6 in Appendix N). In these cases, we find that the computational time of $\tau$-SGA per iteration $t$ is roughly 3-4 times that of $\tau$-GDA. This implies that in terms of the number of iteration $t$, $\tau$-SGA is faster than $\tau$-GDA.

## 5 Conclusion

In this article, we analyze the local convergence of $\tau$-GDA and $\tau$-SGA to two differential equilibria on Riemannian manifold, based on a classical fixed point theorem and spectral analysis. This method allows one to reduce the problem into Euclidean space using a local coordinate chart. We obtain a linear local convergence of $\tau$-GDA to DSE and DNE where the linear rate is upper bounded by the spectral radius of an intrinsic Jacobian matrix. To improve the local convergence, $\tau$-SGA is developed and extended to Riemannian manifold for the first time. Based on the asymptotic analysis where $\tau$ tends to infinity, we find that sometimes an improved rate of $\tau$-SGA to DSE can be reached. The methodology could be served as a basis to analyze other simultaneous algorithms such as the Riemannian Hamiltonian method (Han et al., 2023).

To show the relevance of our results to GAN, we study the behavior of stochastic $\tau$-GDA and $\tau$-SGA on simple benchmarks of orthogonal Wasserstein GANs. Even though our current theory does not apply to analyze these stochastic algorithms, we observe a consistent behavior similar to their deterministic convergence to DSE. This suggests that DSE could be a suitable solution set towards which our chosen initialization algorithms (alternating) $\tau$-GDA converge. This phenomenon is also observed in non-zero sum games such as NS-GAN and WGAN-GP using Adam-based algorithms (Berard et al., 2020).

ACKNOWLEDGMENTS

We thank all the anonymous reviewers for their valuable feedback. Sixin Zhang acknowledges very helpful discussions with Shengyuan Zhao, Yuxin Ge, Jérôme Renault and Yijia Wang. This work was partially supported by a grant from AAP EMERGENCE under the Project HERMES. It was also supported by the computing resources in the OCCIDATA and CALMIP platforms.

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

## A  Notations

Let $d_1$ and $d_2$ denote the dimension of $\mathcal{M}_1$ and $\mathcal{M}_2$. Let $\mathcal{X}(\mathcal{M}_1)$ (resp. $\mathcal{X}(\mathcal{M}_2)$) be the set of continuously differentiable vector fields on $\mathcal{M}_1$ (resp. $\mathcal{M}_2$). Since $f$ is twice continuously differentiable, we have for any $y \in \mathcal{M}_2$ (resp. $x \in \mathcal{M}_1$), $\text{grad}_x f(\cdot, y) \in \mathcal{X}(\mathcal{M}_1)$ (resp. $\text{grad}_y f(x, \cdot) \in \mathcal{X}(\mathcal{M}_2)$). We write $g_1$ and $g_2$ to denote the Riemannian metric on $\mathcal{M}_1$ and $\mathcal{M}_2$.

Let $\partial$ and $\partial^2$ denote the first-order and second-order partial derivatives of a function on Euclidean space. For a real symmetric matrix $A$, $\lambda_{max}(A)$ denotes the maximal eigenvalue of $A$, $\lambda_{min}(A)$ denotes the minimal eigenvalue of $A$, and $\lambda_k(A)$ denotes its $k$-th smallest eigenvalue. For a linear transform $B$, $\rho(B)$ denotes its spectral radius and $\|B\|$ denotes its operator norm.

## B  Definition of DSE

The main idea is to use the local coordinate chart $(O_1 \times O_2, \varphi_1 \times \varphi_2)$ to represent the smooth vector fields $(x, y) \mapsto \text{grad}_x(x, y)$ and $(x, y) \mapsto \text{grad}_y(x, y)$ around the point $(x^*, y^*)$, so as to show the equivalence between (4)-(6) and (1)-(3).

For each $x \in O_1$ (resp. $y \in O_2$), let $\{E_{1,i}(x)\}_{i \le d_1}$ (resp. $\{E_{2,j}(y)\}_{j \le d_2}$) be the canonical basis of the tangent space $T_x\mathcal{M}_1$ (resp. $T_y\mathcal{M}_2$), defined by the tangent map $D\varphi_1^{-1}(\varphi_1(x))[e_{1,i}]$ of the canonical basis $\{e_{1,i}\}_{i \le d_1}$ on $\mathbb{R}^{d_1}$ (resp. $D\varphi_2^{-1}(\varphi_2(y))[e_{2,j}]$ of $\{e_{2,j}\}_{j \le d_2}$ on $\mathbb{R}^{d_2}$).

Let the local coordinate of $(x, y) \in O_1 \times O_2$ be $(u_1, u_2) = (\varphi_1(x), \varphi_2(y)) \in \bar{O}_1 \times \bar{O}_2 = \varphi_1(O_1) \times \varphi_2(O_2) \subset \mathbb{R}^{d_1} \times \mathbb{R}^{d_2}$. We can represent

$$\xi_1(x, y) = \text{grad}_x f(x, y) = \sum_{i \le d_1} \xi_{1,i}(x, y) E_{1,i}(x), \tag{15}$$

$$\xi_2(x, y) = \text{grad}_y f(x, y) = \sum_{j \le d_2} \xi_{2,j}(x, y) E_{2,j}(y). \tag{16}$$

From (15) and (16), the local coordinates of the Riemannian gradients on $\bar{O}_1 \times \bar{O}_2$ are

$$\bar{\xi}_1(u_1, u_2) = \sum_{i \le d_1} \xi_{1,i}(\varphi_1^{-1}(u_1), \varphi_2^{-1}(u_2)) e_{1,i}, \tag{17}$$

$$\bar{\xi}_2(u_1, u_2) = \sum_{j \le d_2} \xi_{2,j}(\varphi_1^{-1}(u_1), \varphi_2^{-1}(u_2)) e_{2,j}. \tag{18}$$

**Equivalence between** (4) **and** (1)    Let the local Riemannian metric matrix at $x = \varphi_1^{-1}(u_1) \in O_1$ and $y = \varphi_2^{-1}(u_2) \in O_2$ be

$$\bar{g}_1(u_1) = (g_1(E_{1,i}(x), E_{1,i'}(x)))_{i,i' \le d_1}, \quad \bar{g}_2(u_2) = (g_2(E_{2,j}(y), E_{2,j'}(y)))_{j,j' \le d_2}. \tag{19}$$

From (Absil et al., 2008, chap. 3.6), we have for $(u_1, u_2) \in \bar{O}_1 \times \bar{O}_2$,

$$\bar{\xi}_1(u_1, u_2) = \bar{g}_1(u_1)^{-1} \cdot \partial_{u_1} \bar{f}(u_1, u_2), \tag{20}$$

$$\bar{\xi}_2(u_1, u_2) = \bar{g}_2(u_2)^{-1} \cdot \partial_{u_2} \bar{f}(u_1, u_2). \tag{21}$$

The equivalence between the DSE condition (4) and (1) follows from (20) and (21), as $\bar{\xi}_1(u_1^*, u_2^*) = 0$ i.f.f. (if and only if) $\text{grad}_x f(x^*, y^*) = 0$ (similarly for the relationship between $\bar{\xi}_2$ and $\text{grad}_y f$).

**Equivalence between** (5) **and** (2)    Let $\eta^* \in T_{y^*}\mathcal{M}_2$, with its local coordinate $\bar{\eta}^* = D\varphi_2(y^*)[\eta^*]$. To show that the positive definiteness of $-\text{Hess}_y f(x^*, y^*)$ is equivalent to the positive definiteness of $-\partial_{u_2 u_2}^2 \bar{f}(u_1^*, u_2^*)$, it is sufficient to verify that

$$\langle \bar{\eta}^*, \partial_{u_2 u_2}^2 \bar{f}(u_1^*, u_2^*) \bar{\eta}^* \rangle = \langle \eta^*, \text{Hess}_y f(x^*, y^*)[\eta^*] \rangle_{y^*}, \tag{22}$$

where $\langle \cdot, \cdot \rangle$ denotes the inner product on Euclidean space, and $\langle \cdot, \cdot \rangle_{y^*}$ denotes the Riemannian inner product on $T_{y^*}M$.

Recall that the Riemannian Hessian on $\mathcal{M}_2$ is defined by the Riemannian connection $\nabla_y : \mathcal{X}(\mathcal{M}_2) \times \mathcal{X}(\mathcal{M}_2) \to \mathcal{X}(\mathcal{M}_2)$ with $\mathrm{Hess}_y f(x, y) = \nabla_y \xi_2(x, y)$. By the R-*linearity* and Leibniz law of the connection, we have

$$\nabla_y \xi_2(x^*, y^*)[\eta^*] = \nabla_y(\sum_j \xi_{2,j}(x^*, y^*) E_{2,j}(y^*))[\eta^*]$$

$$= \sum_j \nabla_y(\xi_{2,j}(x^*, y^*) E_{2,j}(y^*))[\eta^*]$$

$$= \sum_j (D_y \xi_{2,j}(x^*, y^*)[\eta^*]) E_{2,j}(y^*) + \xi_{2,j}(x^*, y^*) \nabla_y(E_{2,j}(y^*))[\eta^*] \quad (23)$$

$$= \sum_j \left( \frac{\partial \bar{\xi}_{2,j}}{\partial u_2}(u_1^*, u_2^*) \bar{\eta}^* \right) E_{2,j}(y^*). \quad (24)$$

We have obtained (24) from (23) because any DSE $(x^*, y^*)$ is a critical point of $f$, therefore $\xi_{2,j}(x^*, y^*) = 0$ for each $j \leq d_2$. Moreover, the tangent map $D_y \xi_{2,j}(x^*, y^*)[\eta^*] = \frac{\partial \bar{\xi}_{2,j}}{\partial u_2}(u_1^*, u_2^*) \bar{\eta}^*$ by definition (Absil et al., 2008, chap. 3.5.4).

Therefore the local coordinate of the tangent vector $\nabla_y \xi_2(x^*, y^*)[\eta^*] \in T_{y^*}\mathcal{M}_2$ is $\frac{\partial \bar{\xi}_2}{\partial u_2}(u_1^*, u_2^*) \bar{\eta}^* \in \mathbb{R}^{d_2}$. It follows from (21),(24), and the Riemannian inner product (Absil et al., 2008, chap. 3.6) that

$$\langle \eta^*, \mathrm{Hess}_y f(x^*, y^*)[\eta^*] \rangle_{y^*} = \left\langle \bar{\eta}^*, \bar{g}_2(u_2^*) \cdot \frac{\partial \bar{\xi}_2}{\partial u_2}(u_1^*, u_2^*) \bar{\eta}^* \right\rangle \quad (25)$$

$$= \left\langle \bar{\eta}^*, \partial^2_{u_2 u_2} \bar{f}(u_1^*, u_2^*) \bar{\eta}^* \right\rangle. $$

Therefore (22) holds. This verifies the equivalence between (5) and (2).

**Equivalence between** (6) **and** (3)  Assume $(x^*, y^*)$ is a DSE of $f$, from the equivalence between (4),(5) and (1),(2), we deduce that $(x^*, y^*)$ (resp. $(u_1^*, u_2^*)$) is a critical point of $f$ (resp. $\bar{f}$). Moreover, $\mathrm{Hess}_y f(x^*, y^*)$ and $\partial^2_{u_2 u_2} \bar{f}(u_1^*, u_2^*)$ are negative definite.

To simplify the notation, let

$$A = -\partial^2_{u_2 u_2} \bar{f}(u_1^*, u_2^*), \quad B = \partial^2_{u_2 u_1} \bar{f}(u_1^*, u_2^*), \quad C = \partial^2_{u_1 u_1} \bar{f}(u_1^*, u_2^*). \quad (26)$$

The condition (6) is equivalent to the positive definiteness of the matrix $C + BA^{-1}B^\mathsf{T}$.

Let $\delta^* \in T_{x^*}\mathcal{M}_1$, with its local coordinate $\bar{\delta}^* = D\varphi_1(x^*)[\delta^*]$. As in (22), it suffices to verify that

$$\langle \bar{\delta}^*, (C + BA^{-1}B^\mathsf{T})\bar{\delta}^* \rangle = \langle \delta^*, [\mathrm{Hess}_x f - \mathrm{grad}^2_{yx} f \cdot (\mathrm{Hess}_y f)^{-1} \cdot \mathrm{grad}^2_{xy} f](x^*, y^*)[\delta^*] \rangle_{x^*}.$$

Based on the proof of (22), we also have $\langle \bar{\delta}^*, C\bar{\delta}^* \rangle = \langle \delta^*, \mathrm{Hess}_x f(x^*, y^*)[\delta^*] \rangle_{x^*}$. It remains to verify that

$$\langle \bar{\delta}^*, BA^{-1}B^\mathsf{T}\bar{\delta}^* \rangle = \langle \delta^*, [-\mathrm{grad}^2_{yx} f \cdot (\mathrm{Hess}_y f)^{-1} \cdot \mathrm{grad}^2_{xy} f](x^*, y^*)[\delta^*] \rangle_{x^*}. \quad (27)$$

Let $\eta^* = [\mathrm{Hess}_y f^{-1} \cdot \mathrm{grad}^2_{xy} f](x^*, y^*)[\delta^*]$, we compute the local coordinate $\bar{\eta}^*$ of $\eta^*$ by converting the following equation

$$\mathrm{Hess}_y f(x^*, y^*)[\eta^*] = \mathrm{grad}^2_{xy} f(x^*, y^*)[\delta^*], \quad (28)$$

into local coordinate. From (24), we have $\mathrm{Hess}_y f(x^*, y^*)[\eta^*] = \sum_j \left( \frac{\partial \bar{\xi}_{2,j}}{\partial u_2}(u_1^*, u_2^*) \bar{\eta}^* \right) E_{2,j}(y^*)$.

By (16) and (18), we obtain

$$\mathrm{grad}^2_{xy} f(x^*, y^*)[\delta^*] = D_x \mathrm{grad}_y f(x^*, y^*)[\delta^*] = \sum_j D_x \xi_{2,j}(x^*, y^*)[\delta^*] E_{2,j}(y^*)$$

$$= \sum_j \left( \frac{\partial \bar{\xi}_{2,j}}{\partial u_1}(u_1^*, u_2^*) \bar{\delta}^* \right) E_{2,j}(y^*). \quad (29)$$

It follows that (28) is equivalent to

$$\frac{\partial \bar{\xi}_2}{\partial u_2}(u_1^*, u_2^*)\bar{\eta}^* = \frac{\partial \bar{\xi}_2}{\partial u_1}(u_1^*, u_2^*)\bar{\delta}^*. \tag{30}$$

We can compute the right hand side of (27) in the local coordinate as in (29) and (25),

$$\begin{aligned}
\langle \delta^*, -\mathrm{grad}_{yx}^2 f(x^*, y^*)[\eta^*]\rangle_{x^*} &= -\langle \delta^*, D_y \mathrm{grad}_x f(x^*, y^*)[\eta^*]\rangle_{x^*} \\
&= -\left\langle \delta^*, \sum_i \left(\frac{\partial \bar{\xi}_{1,i}}{\partial u_2}(u_1^*, u_2^*)\bar{\eta}^*\right) E_{1,i}(x^*)\right\rangle_{x^*} \\
&= -\left\langle \bar{\delta}^*, \bar{g}_1(u_1^*) \cdot \frac{\partial \bar{\xi}_1}{\partial u_2}(u_1^*, u_2^*)\bar{\eta}^*\right\rangle.
\end{aligned}$$

The left hand side of (27) can computed based on (20) and (21), which result in

$$A = -\bar{g}_2(u_2^*) \cdot \frac{\partial \bar{\xi}_2}{\partial u_2}(u_1^*, u_2^*), \tag{31}$$

$$B = \bar{g}_1(u_1^*) \cdot \frac{\partial \bar{\xi}_1}{\partial u_2}(u_1^*, u_2^*), \tag{32}$$

$$C = \bar{g}_1(u_1^*) \cdot \frac{\partial \bar{\xi}_1}{\partial u_1}(u_1^*, u_1^*). \tag{33}$$

Furthermore, the symmetry of the Hessian matrix of $\bar{f}$ implies that

$$B^{\mathsf{T}} = \partial_{u_1 u_2}^2 \bar{f}(u_1^*, u_2^*) = \bar{g}_2(u_2^*) \cdot \frac{\partial \bar{\xi}_2}{\partial u_1}(u_1^*, u_2^*). \tag{33}$$

As a consequence, (30) implies that $\bar{\eta}^* = -A^{-1}B^{\mathsf{T}}\bar{\delta}^*$, and

$$\langle \bar{\delta}^*, BA^{-1}B^{\mathsf{T}}\bar{\delta}^*\rangle = -\left\langle \bar{\delta}^*, \bar{g}_1(u_1^*) \cdot \frac{\partial \bar{\xi}_1}{\partial u_2}(u_1^*, u_2^*)\bar{\eta}^*\right\rangle.$$

Therefore (27) holds.

## C  IMPLICIT FUNCTION THEOREM ON RIEMANNIAN MANIFOLD

We shall state an implicit function theorem which allows one to understand why $(x^*, y^*)$ is a local minimax point in the manifold case. This theorem implies the existence of a solution $(x, h(x))$ sufficiently close to $(x^*, y^*)$ s.t. $\mathrm{grad}_y f(x, h(x)) = 0$. Moreover, due to the continuity of $\mathrm{Hess}_y f$ on $\mathcal{M}_1 \times \mathcal{M}_2$ and the continuity of $h$ near $x^*$, $-\mathrm{Hess}_y f(x, h(x))$ is positive definite, provided that $x$ is close enough to $x^*$. As a consequence, $h(x)$ is the unique strict local maximum of $y \mapsto f(x, y)$ in this neighbor of DSE.

Recall that the set of continuously differentiable vector fields on $\mathcal{M}_2$ is denoted by $\mathcal{X}(\mathcal{M}_2)$. The Riemannian connection on $\mathcal{M}_2$ is denoted by $\nabla_y$.

Consider a parameterized vector field $\xi$ defined at each $x \in \mathcal{M}_1$ with $\xi(x, \cdot) \in \mathcal{X}(\mathcal{M}_2)$. By definition, for each $(x, y) \in \mathcal{M}_1 \times \mathcal{M}_2$, $\xi(x, y) \in T_y\mathcal{M}_2$, and $\nabla_y \xi(x, y)$ is a linear map from $T_y\mathcal{M}_2$ to $T_y\mathcal{M}_2$.

**Theorem C.1.** *Assume $\xi$ is continuously differentiable on an open set $E \subset \mathcal{M}_1 \times \mathcal{M}_2$. Let $(x^*, y^*) \in E$ be a solution of*

$$\xi(x, y) = 0, \quad (x, y) \in E.$$

*If $\nabla_y \xi(x^*, y^*)$ is invertible on $T_{y^*}\mathcal{M}_2$, there exists an open set $U_1 \times U_2 \subset E$ and an open set $W_1 \subset U_1$ such that*

$$\forall x \in W_1, \exists! \, y \in U_2 \quad s.t. \quad \xi(x, y) = 0.$$

*Let the unique $y = h(x)$, i.e. $h : W_1 \to U_2$, such that*

$$y^* = h(x^*), \quad \xi(x, h(x)) = 0, \quad \forall x \in W_1,$$

then $h$ is continuously differentiable on $W_1$. Let $\delta \in T_{x^*}\mathcal{M}_1$, and denote the tangent map of $h$ at $x^*$ by $D_x h(x^*)$, we have

$$D_x h(x^*)[\delta] = -\nabla_y \xi(x^*, y^*)^{-1} \cdot D_x \xi(x^*, y^*)[\delta].$$

We can prove this result by using a local coordinate chart around the point $(x^*, y^*)$ to represent a smooth vector field, and then adopt the proof technique of the implicit function theorem in Euclidean space (Rudin, 1976, Theorem 9.28).

## D   PROOF OF PROPOSITION 2.1

We verify the intrinsic definition of DSE for $(x^*, y^*)$. For the first-order condition in (1), we compute based on Boumal (2023, Proposition 3.61),

$$\partial_x f(x, y) = A^\mathsf{T} y, \quad \mathrm{grad}_y f(x, y) = (I - yy^\mathsf{T})(Ax - b). \tag{34}$$

From the identity involving the pseudo-inverse $A^\mathsf{T} A A^+ = A^\mathsf{T}$, we deduce that $A^\mathsf{T} y^* = 0$. As $b \notin \mathrm{Range}(A)$, we have $Ax^* - b \neq 0$ and therefore $y^*$ is parallel to the (non-zero) vector $Ax^* - b$. From above, the first-order condition holds, i.e. $\partial_x f(x^*, y^*) = 0, \mathrm{grad}_y f(x^*, y^*) = 0$.

From (34), we first verify the second-order condition (2). From Boumal (2023, Corollary 5.16), we compute for $\eta \in T_y \mathcal{M}_2$,

$$\begin{aligned}
\mathrm{Hess}_y f(x, y)[\eta] &= (I - yy^\mathsf{T})(\langle \partial_y \mathrm{grad}_y f(x, y), \eta \rangle) \\
&= (I - yy^\mathsf{T})(-(y\eta^\mathsf{T} + \eta y^\mathsf{T})(Ax - b)) \\
&= \langle y, Ax - b \rangle (-\eta).
\end{aligned}$$

We verify that $-\mathrm{Hess}_y f(x^*, y^*)$ is d.p, because for non-zero $\eta^* \in T_{y^*}\mathcal{M}_2$, we have $\|\eta^*\|^2 > 0$. Moreover, $Ax^* - b \neq 0$, therefore

$$\langle -\mathrm{Hess}_y f(x^*, y^*)[\eta^*], \eta^* \rangle = \langle y^*, Ax^* - b \rangle \|\eta^*\|^2 = \|Ax^* - b\| \|\eta^*\|^2 > 0.$$

We now check the second-order condition (3). It is clear that $\partial^2_{xx} f(x, y) = 0$. We next show $\delta^* \mapsto \mathrm{grad}^2_{xy} f(x^*, y^*)[\delta^*]$ is an injection from $T_{x^*}\mathcal{M}_1$ to $T_{y^*}\mathcal{M}_2$. We compute from (34),

$$\mathrm{grad}^2_{xy} f(x^*, y^*)[\delta^*] = \sum_{i \leq d_1} \partial_{x_i} \mathrm{grad}_y f(x^*, y^*) \delta^*_i = (I - y^* y^{*\mathsf{T}}) A \delta^*.$$

If $\delta^* \neq 0$, then $A\delta^* \neq 0$ because $\mathrm{Ker}(A) = \{0\}$. But $A\delta^*$ is not parallel to $y^*$, since $y^*$ is along the direction $Ax^* - b$ which is not in the range of $A$. This proves that $\mathrm{grad}^2_{xy} f(x^*, y^*)[\delta^*] \neq 0$ if $\delta^* \neq 0$.

The above injection property implies that for $\delta^* \neq 0$, $\eta^* = \mathrm{grad}^2_{xy} f(x^*, y^*)[\delta^*] \neq 0$. As $-\mathrm{Hess}_y f(x^*, y^*)$ is d.p, we use the symmetry of the Riemannian cross-gradients (see Proposition 3.1) to obtain: If $\delta^* \neq 0$, then

$$\langle \delta^*, [\mathrm{Hess}_x f - \mathrm{grad}^2_{yx} f \cdot (\mathrm{Hess}_y f)^{-1} \cdot \mathrm{grad}^2_{xy} f](x^*, y^*)[\delta^*] \rangle = \langle \eta^*, -\mathrm{Hess}_y f(x^*, y^*)^{-1}[\eta^*] \rangle > 0.$$

Therefore (3) holds.

## E   PROOF OF PROPOSITION 2.2

As in the proof of Proposition 2.1 in Appendix D, we have

$$\partial_x f(x, y) = A^\mathsf{T} y - \kappa A^\mathsf{T} Ax, \quad \mathrm{grad}_y f(x, y) = (I - yy^\mathsf{T})(Ax - b). \tag{35}$$

To show the existence of DSE, we shall construct a solution of the form $(x, y^*_x)$, where $\partial_x f(x, y^*_x) = 0$. We consider $y^*_x = (Ax - b)/\|Ax - b\|$ so that $\mathrm{grad}_y f(x, y^*_x) = 0$. We verify that $\mathrm{Ker}(A) = \{0\}$ implies that $A^\mathsf{T} A$ is p.d. It follows that $A^+ = (A^\mathsf{T} A)^{-1} A^\mathsf{T}$, and that the condition $\partial_x f(x, y^*_x) = 0$ is equivalent to

$$\frac{A^\mathsf{T}(Ax - b)}{\|Ax - b\|} = \kappa A^\mathsf{T} Ax \quad \Leftrightarrow \quad (1 - \kappa \|Ax - b\|)x = A^+ b.$$

For $(x^*, y^*) = (x^*, y^*_{x^*})$ to be a DSE, we assume $x^* = cA^+b$ and we want to find a $c \in \mathbb{R}$ such that
$$F(\kappa, c) = c(1 - \kappa\|cAA^+b - b\|) - 1 = 0.$$
From Proposition 2.1, we have $F(0, 1) = 0$. We next apply the implicit function theorem (c.f. Theorem C.1) to show the existence of $c$ close to 1 if $\kappa$ is close to 0. It suffices to verify that $\frac{\partial F}{\partial c}(0, 1) = 1$ based on
$$\frac{\partial F}{\partial c}(\kappa, c) = \left(1 - \kappa\|cAA^+b - b\|\right) + c\left(-\kappa\frac{\langle AA^+b, cAA^+b - b\rangle}{\|cAA^+b - b\|}\right).$$
This suggests that there is an implicit function $h$ defined on an open neighbor $W_1$ of $\kappa = 0$ ($0 \in W_1$) such that $F(\kappa, h(\kappa)) = 0$. Moreover, $h(0) = 1$. We next check that if $c = h(\kappa)$ for some range $0 < \kappa < \kappa_0$, then $(x^*, y^*)$ defined in the statement is a DSE of $f$. Since (35) holds at $(x^*, y^*)$, it suffices to verify the second-order condition of DSE.

We verify the second-order condition (2) by using the same proof as Proposition 2.1. To check the second-order condition (3), we first compute from (35)
$$\eta^* = \mathrm{grad}^2_{xy} f(x^*, y^*)[\delta^*] = \sum_{i \leq d_1} \partial_{x_i} \mathrm{grad}_y f(x^*, y^*)\delta^*_i = (I - y^*y^{*\mathsf{T}})A\delta^*.$$
By following the proof of Proposition 2.1, we then compute
$$
\begin{aligned}
S(\kappa) &= \langle \delta^*, [\mathrm{Hess}_x f - \mathrm{grad}^2_{yx} f \cdot (\mathrm{Hess}_y f)^{-1} \cdot \mathrm{grad}^2_{xy} f](x^*, y^*)[\delta^*] \rangle \\
&= -\kappa\|A\delta^*\|^2 + \langle \eta^*, -\mathrm{Hess}_y f(x^*, y^*)^{-1}[\eta^*] \rangle \\
&= -\kappa\|A\delta^*\|^2 + \frac{1}{\|Ax^* - b\|}\|\eta^*\|^2.
\end{aligned}
$$
It is clear that $S(\kappa)$ is a continuous function of $\kappa$ because $x^* = h(\kappa)A^+b$ and $y^* = (Ax^* - b)/\|Ax^* - b\|$ are continuous with respect to $\kappa$. Moreover, $S(0) > 0$ from the proof of Proposition 2.1 (due to the injection property of $\delta^* \mapsto \mathrm{grad}^2_{xy} f(x^*, y^*)[\delta^*]$). Therefore there exists $\kappa_0 > 0$ so that $S(\kappa) > 0$ for $0 < \kappa < \kappa_0$ (i.e. (3) holds).

## F    PROOF OF PROPOSITION 2.3

We verify the intrinsic definition of DNE for $(x^*, y^*)$. For the first-order condition (7), we compute
$$\partial_x f(x, y) = A^\mathsf{T}(Ax + y - b), \quad \mathrm{grad}_y f(x, y) = (I - yy^\mathsf{T})(Ax + y - b). \tag{36}$$
From the identity involving the pseudo-inverse $A^\mathsf{T}AA^+ = A^\mathsf{T}$, we deduce that $A^\mathsf{T}(Ax^* + y^* - b) = 0$. As $b \notin \mathrm{Range}(A)$, we have $Ax^* - b \neq 0$ and therefore $y^*$ is parallel to the vector $Ax^* - b$. From above, the first-order condition in (1) holds.

From (36), we next verify the second-order condition (8). It is clear that $\partial^2_{xx} f(x, y) = A^\mathsf{T}A$ is p.d. From Boumal (2023, Corollary 5.16), we compute for $\eta \in T_y \mathcal{M}_2$,
$$
\begin{aligned}
\mathrm{Hess}_y f(x, y)[\eta] &= (I - yy^\mathsf{T})(\langle \partial_y \mathrm{grad}_y f(x, y), \eta \rangle) \\
&= (I - yy^\mathsf{T})(\eta - (y\eta^\mathsf{T} + \eta y^\mathsf{T})(Ax + y - b)) \\
&= \eta(1 - \langle y, Ax + y - b \rangle) \\
&= \langle y, Ax - b \rangle(-\eta).
\end{aligned}
$$
We verify that $-\mathrm{Hess}_y f(x^*, y^*)$ is d.p, because for non-zero $\eta^* \in T_{y^*}\mathcal{M}_2$, we have $\|\eta^*\|^2 > 0$. Moreover, $Ax^* - b \neq 0$, therefore
$$\langle -\mathrm{Hess}_y f(x^*, y^*)[\eta^*], \eta^* \rangle = \langle y^*, Ax^* - b \rangle\|\eta^*\|^2 = \|Ax^* - b\|\|\eta^*\|^2 > 0.$$
Therefore (8) holds.

## G    PROOF OF THEOREM 3.1

The proof is illustrated in Figure 3, which contains three main steps: (1) identify a set $\bar{S}$ where the local update rule $\bar{\mathbf{T}}$ is well-defined. (2) adapt the proof technique of the classical Ostrowski Theorem to identity a local stable region $\bar{S}_\delta$ and pull it back to the manifold (i.e. $S_\delta$). (3) relate a Euclidean distance in the local coordinate to the Riemannian distance on $S_\delta$ to establish the linear convergence.

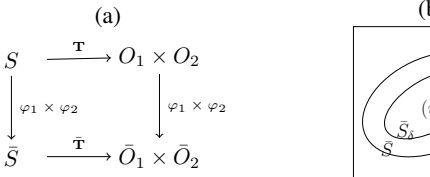

Figure 3: Local convergence of a deterministic simultaneous algorithm. (a): The update rule $\mathbf{T}$ on the manifold $\mathcal{M}_1 \times \mathcal{M}_2$ induces a local update rule $\bar{\mathbf{T}}$ in the local coordinate system, defined by a chart $(O_1 \times O_2, \varphi_1 \times \varphi_2)$ around $(x^*, y^*)$. (b): The induced update rule $\bar{\mathbf{T}}$ is defined in the local coordinate on the set $\bar{S}$ around $(u_1^*, u_2^*) = \varphi_1 \times \varphi_2 \circ (x^*, y^*)$. It contains a local stable region $\bar{S}_\delta$.

**Preliminary**   Let $\varphi = \varphi_1 \times \varphi_2$. Let us consider the set $S = \mathbf{T}^{-1}(O_1 \times O_2) \cap (O_1 \times O_2)$ and its local coordinate domain $\bar{S} = (\varphi_1 \times \varphi_2)(S) \subset \bar{O}_1 \times \bar{O}_2$. By definition, $(x, y) \in S \subset O_1 \times O_2$ and $\mathbf{T}(x, y) \in O_1 \times O_2$. Therefore the induced dynamics $\bar{\mathbf{T}}$ is well-defined on $\bar{S}$, i.e. $\forall (u_1, u_2) \in \bar{S}$,

$$\bar{\mathbf{T}}(u_1, u_2) = \varphi(\mathbf{T}(\varphi_1^{-1}(u_1), \varphi_2^{-1}(u_2))).$$

Note that $S$ and $\bar{S}$ are non-empty open sets since $\mathbf{T}$ is continuous (by the continuity of the vector fields $\xi_1, \xi_2$ and the retractions $\mathcal{R}_1$ and $\mathcal{R}_2$) and $(x^*, y^*)$ is a fixed point of $\mathbf{T}$.

**Local stable region**   We aim to identify a local stable region $\bar{S}_\delta \subset \bar{S}$ around $(u_1^*, u_2^*)$ (the local coordinate of $(x^*, y^*)$), so that if $(u_1, u_2) \in \bar{S}_\delta$, then $\bar{\mathbf{T}}(u_1, u_2) \in \bar{S}_\delta \subset \bar{O}_1 \times \bar{O}_2$. Let $S_\delta = \varphi^{-1}(\bar{S}_\delta)$, and assume $(x(0), y(0)) \in S_\delta$, then by recursion $\bar{\mathbf{T}}(\varphi_1(x(t)), \varphi_2(y(t)))$ is always well defined since the sequence $(x(t), y(t)) \in S_\delta \subset S, \forall t \geq 1$.

To construct such a region, the key is to verify that the spectral radius $\rho$ of the Jacobian matrix $\bar{\mathbf{T}}'(u_1^*, u_2^*)$ is strictly smaller than one. In Appendix G.1, we verify that the eigenvalues of $\mathbf{T}'(x^*, y^*)$ are the same as $\bar{\mathbf{T}}'(u_1^*, u_2^*)$. Therefore according to our assumption, $\rho = \rho(\mathbf{T}'(x^*, y^*) < 1$. From the proof of Ostrowski Theorem (Ortega & Rheinboldt, 1970, Section 10.1.3), for an arbitrary $\epsilon > 0$, there exists a norm $\|\cdot\|_\epsilon$ on $\mathbb{R}^{d_1} \times \mathbb{R}^{d_2}$, such that

$$\|\bar{\mathbf{T}}'(u_1^*, u_2^*)\|_\epsilon \leq \rho + \epsilon/2.$$

Furthermore, due to the differentiability of $\bar{\mathbf{T}}$ at the fixed point, there exists $\delta > 0$, and $\bar{S}_\delta = \{(u_1, u_2) | \|(u_1, u_2) - (u_1^*, u_2^*)\|_\epsilon < \delta\} \subset \bar{S}$, such that

$$\|\bar{\mathbf{T}}(u_1, u_2) - (u_1^*, u_2^*)\|_\epsilon \leq (\|\bar{\mathbf{T}}'(u_1^*, u_2^*)\|_\epsilon + \epsilon/2)\|(u_1, u_2) - (u_1^*, u_2^*)\|_\epsilon$$

$$\leq (\rho + \epsilon)\|(u_1, u_2) - (u_1^*, u_2^*)\|_\epsilon, \quad \forall (u_1, u_2) \in \bar{S}_\delta. \qquad (37)$$

Let's choose $\epsilon$ so that $\rho + \epsilon < 1$. It follows that the open set $S_\delta = \varphi^{-1}(\bar{S}_\delta)$ is a local stable region. Furthermore, we can identify an open geodesically convex subset of $S_\delta$, on which Riemannian distance equals to geodesic distance. This set is written as $S_\delta'$. It contains $(x^*, y^*)$ and is homeomorphic to a ball, i.e. without any hole.

**Locally convergent with linear rate $\rho$**   From (37), if $(u_1(0), u_2(0)) \in \bar{S}_\delta$, the sequence $(u_1(t), u_2(t))$ stays in $\bar{S}_\delta$ and converges to $(u_1^*, u_2^*)$ as $t \to \infty$. As $\varphi = \varphi_1 \times \varphi_2$ is a continuous bijection (homeomorphism) from $O_1 \times O_2$ to $\bar{O}_1 \times \bar{O}_2$, we have equivalently that if $(x(0), y(0)) \in S_\delta = \varphi^{-1}(\bar{S}_\delta)$, the sequence $(x(t), y(t))$ will stay in $S_\delta$ and converges to $(x^*, y^*)$.

The linear convergence rate in the local coordinate system in (37) can be used to control the Riemannian distance $d(t)$ between $\mathbf{p}(t) = (x(t), y(t)) \in S_\delta$ and $\mathbf{p}^* = (x^*, y^*) \in S_\delta'$. Since $\mathbf{p}(t)$ converges to $\mathbf{p}^*$ as $t \to \infty$, it suffices to analyze large enough $t$ for the convergence rate. Without loss of generality, we assume next that $\mathbf{p}(t) \in S_\delta'$ for any $t \geq 0$.

Let $\Gamma_\delta(t)$ be the set of piece-wise smooth curves restricted on $S_\delta$ with an initial point $\mathbf{p}(t)$ (at time 0) and a last point $\mathbf{p}^*$ (at time 1). Each curve $\gamma \in \Gamma_\delta(t)$ is composed of a finite number of smooth curves, indexed by $k$. The $k$-th smooth curve $\gamma_k$ is defined on some time interval $[s_k, s_{k+1}] \subset [0, 1]$. Then we have

$$d(t) = \inf_{\gamma = (\gamma_k)_k \in \Gamma_\delta(t)} \sum_k \int_{s_k}^{s_{k+1}} \|\gamma_k'(s)\|_{\gamma_k(s)} ds,$$

where $\| \cdot \|_{\mathbf{p}}$ is the Riemmanian metric at $\mathbf{p} \in \mathcal{M}$. As $S'_\delta$ is geodesically convex, the Riemannian distance $d(t)$ can be attained at certain geodesic curve $\gamma \in \Gamma_\delta(t)$.

As the closure of the set $S_\delta$ is compact and $\bar{S}_\delta$ is convex, there are two positive constants $0 < A_{\epsilon,\delta} \leq B_{\epsilon,\delta}$ such that

$$A_{\epsilon,\delta}\|\varphi(\mathbf{p}(t)) - \varphi(\mathbf{p}^*)\|_\epsilon \leq d(t) \leq B_{\epsilon,\delta}\|\varphi(\mathbf{p}(t)) - \varphi(\mathbf{p}^*)\|_\epsilon. \tag{38}$$

A relevant proof on how to obtain $A_{\epsilon,\delta}$ and $B_{\epsilon,\delta}$ is given in Hu (1969)[Chapter 3, Lemma 3.3].

From (37), we have

$$\|\varphi(\mathbf{p}(t)) - \varphi(\mathbf{p}^*)\|_\epsilon \leq (\rho + \epsilon)^{t+1}\|\varphi(\mathbf{p}(0)) - \varphi(\mathbf{p}^*)\|_\epsilon. \tag{39}$$

Combining (38) and (39), we obtain the constant $C = B_{\epsilon,\delta}/A_{\epsilon,\delta}$ such that

$$d(t) \leq C(\rho + \epsilon)^{t+1}d(0).$$

We remark that in general, this constant $C$ can also depend on the initial point $\mathbf{p}(0)$.

## G.1 SPECTRAL RADIUS OF JACOBIAN MATRIX IN THE LOCAL COORIDATE

We next compute the matrix $\bar{\mathbf{T}}'(u_1^*, u_2^*)$, and then relate it to $\mathbf{T}'(x^*, y^*)$. This computation also tells us that the tangent map of $\mathbf{T}$ at $(x^*, y^*)$ equals to $\mathbf{T}'(x^*, y^*)$.

First of all, we specify the induced dynamics $\bar{\mathbf{T}}$ of $\mathbf{T}$ by using the local coordinate representation of the retractions $\mathcal{R}_1$ and $\mathcal{R}_2$ near the fixed point $(x^*, y^*)$. Let $(O_1 \times O_2, \varphi_1 \times \varphi_2)$ be the local chart. Since a retraction (Absil et al., 2008, Section 4.1) is a smooth function from the tangent bundle of a manifold to the manifold itself, we can identify an open subset $B_1$ of the tangent bundle of $\mathcal{M}_1$ (similarly on $B_2$ for $\mathcal{M}_2$) such that $x \in O_1$ and $\mathcal{R}_{1,x}(\xi_1) \in O_1$ if $(x, \xi_1) \in B_1$ This set $B_1$ can then be mapped to an open set $\bar{B}_1 \subset \mathbb{R}^{d_1} \times \mathbb{R}^{d_1}$, using the local chart of the tangent bundle (Absil et al., 2008, Section 3.5.3). On this subset $\bar{B}_1$, we can define the local representation of $\mathcal{R}_1$ as follows

$$\bar{\mathcal{R}}_1 : \bar{B}_1 \to \bar{O}_1, \quad \bar{\mathcal{R}}_1(u_1, \bar{\xi}_1) = \varphi_1 \circ \mathcal{R}_{1,\varphi_1^{-1}(u_1)}(D\varphi_1^{-1}(u_1)[\bar{\xi}_1]),$$

Similarly for $\mathcal{R}_2$, we can define $\bar{\mathcal{R}}_2 : \bar{B}_2 \to \bar{O}_2$ on an open set $\bar{B}_2 \subset \mathbb{R}^{d_2} \times \mathbb{R}^{d_2}$. From the above definition of $\bar{\mathcal{R}}_1$ and $\bar{\mathcal{R}}_2$, as well as the local coordinate of $\xi_1(x, y)$ and $\xi_2(x, y)$ (as in (17),(18)), we obtain the induced dynamics

$$\bar{\mathbf{T}}(u_1, u_2) = \begin{pmatrix} \bar{\mathcal{R}}_1(u_1, \bar{\xi}_1(u_1, u_2)) \\ \bar{\mathcal{R}}_2(u_2, \bar{\xi}_2(u_1, u_2)) \end{pmatrix}, \tag{40}$$

which is well defined on the non-empty open set $\{(u_1, u_2) \in \bar{O}_1 \times \bar{O}_2 | (u_1, \bar{\xi}_1(u_1, u_2)) \in \bar{B}_1, (u_2, \bar{\xi}_2(u_1, u_2)) \in \bar{B}_2\}$.

From (40), we can compute the Jacobian matrix $\bar{\mathbf{T}}'(u_1^*, u_2^*)$ using the chain rule. As $\bar{\xi}_1(u_1^*, u_2^*) = 0$ and $\bar{\xi}_2(u_1^*, u_2^*) = 0$, we have

$$\bar{\mathbf{T}}'(u_1^*, u_2^*) = \begin{pmatrix} I_{d_1} + \frac{\partial \bar{\xi}_1}{\partial u_1}(u_1^*, u_2^*) & \frac{\partial \bar{\xi}_1}{\partial u_2}(u_1^*, u_2^*) \\ \frac{\partial \bar{\xi}_2}{\partial u_1}(u_1^*, u_2^*) & I_{d_2} + \frac{\partial \bar{\xi}_2}{\partial u_2}(u_1^*, u_2^*) \end{pmatrix}, \tag{41}$$

since the retraction by definition satisfies $\frac{\partial \bar{\mathcal{R}}_1}{\partial u_1}(u_1^*, 0) = I_{d_1}, \frac{\partial \bar{\mathcal{R}}_1}{\partial \bar{\xi}_1}(u_1^*, 0) = I_{d_1}$ (similarly for $\bar{\mathcal{R}}_2$).

We next verify that eigenvalues of the matrix in (41) are the same as $\mathbf{T}'(x^*, y^*)$. Let $\delta^* = \sum_i \bar{\delta}_i^* E_{1,i}(x^*) \in T_{x^*}\mathcal{M}_1$, $\eta^* = \sum_j \bar{\eta}_j^* E_{2,j}(y^*) \in T_{y^*}\mathcal{M}_2$, then following the same argument

as in (24) and (29), we have

$$\nabla_x \xi_1(x^*, y^*)[\delta^*] = \sum_{i=1}^{d_1} \left( \frac{\partial \bar{\xi}_{1,i}}{\partial u_1}(u_1^*, u_2^*) \bar{\delta}^* \right) E_{1,i}(x^*),$$

$$D_y \xi_1(x^*, y^*)[\eta^*] = \sum_{i=1}^{d_1} \left( \frac{\partial \bar{\xi}_{1,i}}{\partial u_2}(u_1^*, u_2^*) \bar{\eta}^* \right) E_{1,i}(x^*),$$

$$D_x \xi_2(x^*, y^*)[\delta^*] = \sum_{j=1}^{d_2} \left( \frac{\partial \bar{\xi}_{2,j}}{\partial u_1}(u_1^*, u_2^*) \bar{\delta}^* \right) E_{2,j}(y^*),$$

$$\nabla_y \xi_2(x^*, y^*)[\eta^*] = \sum_{j=1}^{d_2} \left( \frac{\partial \bar{\xi}_{2,j}}{\partial u_2}(u_1^*, u_2^*) \bar{\eta}^* \right) E_{2,j}(y^*).$$

From the definition of eigenvalue and eigenvector pairs, these four equations indicate that their eigenvalues are indeed the same. They also indicate that the tangent map of $\mathbf{T}$ at $(x^*, y^*)$ equals to (9), which means that the tangent map of $\mathbf{T}$ at the fixed point does not depend on the choice of retraction.

## H    PROOF OF THEOREM 3.2

We first rewrite $\mathbf{T}'(x^*, y^*) = I + \gamma \mathbf{M}_g$ in a local coordinate chart with $\bar{\mathbf{T}}'(u_1^*, u_2^*) = I_{d_1+d_2} + \gamma M_g'$.

From the definition of $\mathbf{M}_g$ in (11) and the connection between $\mathbf{T}'$ and $\bar{\mathbf{T}}'$ in (41),(31)-(33), we have

$$M_g' = \begin{pmatrix} -\bar{g}_1(u_1^*)^{-1} \cdot C & -\bar{g}_1(u_1^*)^{-1} \cdot B \\ \tau \bar{g}_2(u_2^*)^{-1} \cdot B^{\intercal} & -\tau \bar{g}_2(u_2^*)^{-1} \cdot A \end{pmatrix},$$

where the matrices $A,B,C$ are defined in (26). Furthermore, $\mathbf{M}_g$ and $M_g'$ have the same eigenvalues.

We next show that the real part of each eigenvalue of $M_g'$ is strictly smaller than zero. From this, $\gamma^\bullet(\mathbf{M}_g) = \gamma^\bullet(M_g') > 0$. We first check that if $0 < \gamma < \gamma^\bullet(M_g')$, the spectral radius of $\bar{\mathbf{T}}'(u_1^*, u_2^*)$ is strictly smaller than one. In general, if $\lambda = \lambda_0 + i\lambda_1$ is a complex eigenvalue of a matrix $M$ with $\lambda_0 < 0$, then $1 + \gamma\lambda$ is an eigenvalue of the matrix $I + \gamma M$. To ensure $|1 + \gamma\lambda| < 1$, it is sufficient that $0 < \gamma < \gamma^\bullet(M)$. This is because $|1 + \gamma\lambda|^2 = 1 + 2\gamma\lambda_0 + \gamma^2\lambda_0^2 + \gamma^2\lambda_1^2 < 1$ holds if $0 < \gamma < -2\lambda_0/(\lambda_0^2 + \lambda_1^2)$.

To analyze $M_g'$, we could not apply Li et al. (2022, Lemma 5.2) directly since the local metric $\bar{g}_1(u_1^*)$ and $\bar{g}_2(u_2^*)$ are not identity matrices. We consider a matrix which is similar to $M_g'$ so that they have the same eigenvalues,

$$M_g = \begin{pmatrix} -\bar{\mathbf{C}} & -\bar{\mathbf{B}} \\ \tau \bar{\mathbf{B}}^{\intercal} & -\tau \bar{\mathbf{A}} \end{pmatrix}, \tag{42}$$

where

- $\bar{\mathbf{C}} = \bar{g}_1(u_1^*)^{-1/2} \cdot C \cdot \bar{g}_1(u_1^*)^{-1/2}$
- $\bar{\mathbf{B}} = \bar{g}_1(u_1^*)^{-1/2} \cdot B \cdot \bar{g}_2(u_2^*)^{-1/2}$
- $\bar{\mathbf{A}} = \bar{g}_2(u_2^*)^{-1/2} \cdot A \cdot \bar{g}_2(u_2^*)^{-1/2}$

Under the assumptions of Theorem 3.2, we check that the following conditions hold:

- $\bar{\mathbf{A}}$ is p.d because $\mathbf{A}$ is p.d.
- $\bar{\mathbf{C}} + \bar{\mathbf{B}}\bar{\mathbf{A}}^{-1}\bar{\mathbf{B}}^{\intercal}$ is p.d because $\mathbf{C} + \mathbf{B}\mathbf{A}^{-1}\mathbf{B}^{\intercal}$ is p.d.
- $\tau > \|\bar{\mathbf{C}}\|/\lambda_{min}(\bar{\mathbf{A}})$.

Indeed, we can relate these conditions to the following intrinsic quantities on the manifold $\mathcal{M}_1 \times \mathcal{M}_2$, by following the proof in Appendix B:

- Let $\mathbf{A} = -\text{Hess}_y f(x^*, y^*)$, then $\mathbf{A}$ and $\bar{\mathbf{A}}$ have the same eigenvalues.

- Let $\mathbf{B} = \text{grad}^2_{yx} f(x^*, y^*)$, $\mathbf{B}^\intercal = \text{grad}^2_{xy} f(x^*, y^*)$, then $\mathbf{C} + \mathbf{B}\mathbf{A}^{-1}\mathbf{B}^\intercal$ and $\bar{\mathbf{C}} + \bar{\mathbf{B}}\bar{\mathbf{A}}^{-1}\bar{\mathbf{B}}^\intercal$ have the same eigenvalues.

- Let $\mathbf{C} = \text{Hess}_x f(x^*, y^*)$, then $\mathbf{C}$ and $\bar{\mathbf{C}}$ have the same eigenvalues. As $\mathbf{C}$ and $\bar{\mathbf{C}}$ are symmetric, $\|\mathbf{C}\| = \max_k |\lambda_k(\mathbf{C})| = \|\bar{\mathbf{C}}\|$. Therefore, the condition $\tau > \|\mathbf{C}\|/\lambda_{min}(\mathbf{A})$ is equivalent to $\tau > \|\bar{\mathbf{C}}\|/\lambda_{min}(\bar{\mathbf{A}})$.

## H.1 Spectral analysis of $M_g$

To analyze the eigenvalues of $M_g$, we use the next proposition which is adapted from Li et al. (2022, Lemma 5.2) with a refined range of $\tau$. For two real symmetric matrices $A$ and $B$, we write $A \geq B$ if $A - B$ is semi-p.d. We write $A > B$ if $A - B$ is p.d.

Let us first introduce a working assumption which will be needed several times.

**Assumption H.1.** *Let $A \in \mathbb{R}^{m \times m}$, $B \in \mathbb{R}^{n \times m}$, $C \in \mathbb{R}^{n \times n}$ such that $A$ and $C + BA^{-1}B^\intercal$ are positive definite.*

**Proposition H.1.** *Under Assumption H.1 and $\tau > \|C\|/\lambda_{min}(A)$, any eigenvalue $\lambda = \lambda_0 + i\lambda_1$ (with $\lambda_0 \in \mathbb{R}, \lambda_1 \in \mathbb{R}$) of the following matrix,*

$$M = \begin{pmatrix} -C & -B \\ \tau B^\intercal & -\tau A \end{pmatrix}$$

*satisfies $\lambda_0 < 0$.*

*Proof.* We show that $\lambda_0 \geq 0$ will lead to a contradiction, by following the proof of Li et al. (2022, Lemma 5.2). Assume that $\lambda$ is an eigenvalue of $M$, i.e. $\det(\lambda I - M) = 0$. To compute $\det(\lambda I - M)$, note that $\lambda I + \tau A$ is invertible since $\lambda_0 I + \tau A$ is positive definite ($\tau A$ is positive definite and $\lambda_0 \geq 0$). By the Schur complement, we have

$$\det(\lambda I - M) = \det \begin{pmatrix} C + \lambda I & B \\ -\tau B^\intercal & \tau A + \lambda I \end{pmatrix}$$
$$= \det(\lambda I + \tau A)\det(H(\lambda))$$

where $H(\lambda) = \lambda I + C + B(\lambda/\tau I + A)^{-1}B^\intercal$. As $\lambda I + \tau A$ is invertible, $\det(\lambda I - M) = 0$ implies that $\det(H(\lambda)) = 0$.

Let the spectral decomposition of $A$ be $U\Lambda_A U^\intercal$, with orthogonal $U \in \mathbb{R}^{m \times m}$ and diagonal $\Lambda_A = \text{diag}(\lambda_1(A), \cdots, \lambda_m(A)) \in \mathbb{R}^{m \times m}$. Then

$$H(\lambda) = \lambda I + C + \tilde{B}D\tilde{B}^\intercal,$$

with $\tilde{B} = BU$ and $D = \text{diag}(d_1, \cdots, d_m)$ where

$$d_k = \frac{1}{\lambda/\tau + \lambda_k(A)} = \frac{\lambda_0/\tau + \lambda_k(A) - i\lambda_1/\tau}{(\lambda_0/\tau + \lambda_k(A))^2 + (\lambda_1/\tau)^2}, \quad 1 \leq k \leq m.$$

It follows that if $\lambda_0 > \|C\|$, the real-part of $H(\lambda)$ is

$$\text{Re}(H(\lambda)) = \lambda_0 I + C + \tilde{B}\text{Re}(D)\tilde{B}^\intercal \quad \text{p.d.} \tag{43}$$

This is contradictory to the fact that $\det(H(\lambda)) = 0$ (Li et al., 2022, Corollary 10.2).

On the other hand, if $0 \leq \lambda_0 \leq \|C\|$ and $\lambda_1 \neq 0$, we consider for $\beta \in \mathbb{R}$,

$$\text{Re}(H(\lambda)) + \frac{\tau\beta}{\lambda_1}\text{Im}(H(\lambda)) = \lambda_0 I + C + \tilde{B}\text{Re}(D)\tilde{B}^\intercal + \frac{\tau\beta}{\lambda_1}(\lambda_1 I + \tilde{B}\text{Im}(D)\tilde{B}^\intercal)$$
$$= (\lambda_0 + \tau\beta)I + C + \tilde{B}F\tilde{B}^\intercal, \tag{44}$$

where $F = \text{diag}(f_1, \cdots, f_m)$ with

$$f_k = \frac{\lambda_0/\tau + \lambda_k(A) - \beta}{(\lambda_0/\tau + \lambda_k(A))^2 + (\lambda_1/\tau)^2}, \quad 1 \leq k \leq m. \tag{45}$$

Take $\beta = \lambda_{min}(A)$, then $f_k \geq 0$ for each $k \leq m$. The condition $\tau\lambda_{min}(A) > \|C\|$ implies that $(\lambda_0 + \tau\beta)I + C$ is p.d. and together with (45), we have (44) is p.d., so it is contradictory to the fact that $\det(H(\lambda)) = 0$ (Li et al., 2022, Lemma 10.1).

Lastly, if $0 \leq \lambda_0 \leq \|C\|$ and $\lambda_1 = 0$, we have from (43),

$$H(\lambda) = \lambda_0 I + C + \tilde{B}D\tilde{B}^{\mathsf{T}} \tag{46}$$

with $d_k = \frac{1}{\lambda_0/\tau + \lambda_k(A)} = \frac{1}{\lambda_k(A)} - \frac{\lambda_0/\tau}{(\lambda_0/\tau + \lambda_k(A))\lambda_k(A)} \geq \frac{1}{\lambda_k(A)} - \frac{\lambda_0/\tau}{\lambda_{min}(A)\lambda_k(A)}$. It follows that

$$H(\lambda) \geq \lambda_0 I + C + \left(1 - \frac{\lambda_0/\tau}{\lambda_{min}(A)}\right)BA^{-1}B^{\mathsf{T}}$$

$$= \left(1 - \frac{\lambda_0/\tau}{\lambda_{min}(A)}\right)(C + BA^{-1}B^{\mathsf{T}}) + \frac{\lambda_0/\tau}{\lambda_{min}(A)}C + \lambda_0 I.$$

As $\tau > \|C\|/\lambda_{min}(A)$, $0 \leq \lambda_0 \leq \|C\|$ we have that $0 \leq \frac{\lambda_0/\tau}{\lambda_{min}(A)} < 1$ and that $I + \frac{1}{\tau\lambda_{min}(A)}C$ is p.d, i.e. we find that again $H(\lambda)$ is p.d which is contradictory. In conclusion, $\lambda_0 < 0$. $\qquad\square$

## H.2 LOCAL CONVERGENCE RATE OF $\tau$-GDA

We use the next result obtained in Li et al. (2022, Lemma 5.3) to control the local convergence rate of $\tau$-GDA. It controls the spectral radius of the matrix $I + \gamma M_g$ on a specific range of $\tau$ and $\gamma$.

Recall that $L_g = \max(\|\mathbf{A}\|, \|\mathbf{B}\|, \|\mathbf{C}\|)$ and $\mu_g = \min(L_g, \lambda_{min}(\mathbf{C} + \mathbf{B}\mathbf{A}^{-1}\mathbf{B}^{\mathsf{T}}))$.

**Proposition H.2.** *Assume $(x^*, y^*)$ is a DSE of $f \in C^2$. If $\tau \geq \frac{2L_g}{\lambda_{min}(\mathbf{A})}$ and $\gamma = \frac{1}{4\tau L_g}$, we have $\rho(I + \gamma M_g) \leq 1 - \frac{\mu_g}{16\tau L_g}$.*

*Proof.* Let $L = \max(\|\bar{\mathbf{A}}\|, \|\bar{\mathbf{B}}\|, \|\bar{\mathbf{C}}\|)$ and $\mu_x = \min(L, \lambda_{min}(\bar{\mathbf{C}} + \bar{\mathbf{B}}\bar{\mathbf{A}}^{-1}\bar{\mathbf{B}}^{\mathsf{T}}))$. We verify that $\lambda_{min}(\mathbf{A}) = \lambda_{min}(\bar{\mathbf{A}})$, $L = L_g$ and $\mu_g = \mu_x$. From the proof of Li et al. (2022, Lemma 5.3), we obtain directly the upper bound of the spectral radius of $I + \gamma M_g$, related to $L$ and $\mu_x$. $\qquad\square$

## I PROOF OF THEOREM 3.3

The proof is based on Jin et al. (2020, Proposition 26) and Zhang et al. (2022, Theorem 4).

To show the valid range of $\tau$, we consider a matrix $\tilde{M}_g$ which is similar to $M_g$ (defined in (42)),

$$\tilde{M}_g = \begin{pmatrix} 0 & \sqrt{\frac{1}{\tau}}I \\ I & 0 \end{pmatrix} \begin{pmatrix} -\bar{\mathbf{C}} & -\bar{\mathbf{B}} \\ \tau\bar{\mathbf{B}}^{\mathsf{T}} & -\tau\bar{\mathbf{A}} \end{pmatrix} \begin{pmatrix} 0 & \sqrt{\frac{1}{\tau}}I \\ I & 0 \end{pmatrix}^{-1}$$

$$= \begin{pmatrix} \sqrt{\tau}\bar{\mathbf{B}}^{\mathsf{T}} & -\sqrt{\tau}\bar{\mathbf{A}} \\ -\bar{\mathbf{C}} & -\bar{\mathbf{B}} \end{pmatrix} \begin{pmatrix} 0 & I \\ \sqrt{\tau}I & 0 \end{pmatrix} = \begin{pmatrix} -\tau\bar{\mathbf{A}} & \sqrt{\tau}\bar{\mathbf{B}}^{\mathsf{T}} \\ -\sqrt{\tau}\bar{\mathbf{B}} & -\bar{\mathbf{C}} \end{pmatrix}.$$

We verify that for any $\tau > 0$, the eigenvalues of $\tilde{M}_g$ are strictly negative, therefore one can compute $\gamma^{\bullet}(\mathbf{M}_g) = \gamma^{\bullet}(M_g) = \gamma^{\bullet}(\tilde{M}_g)$ to set the convergence range for $\gamma$.

As in Daskalakis & Panageas (2018, Lemma 2.7), the Ky Fan inequality implies that the real part of each eigenvalue $\lambda$ of $\tilde{M}_g$ is upper bounded by the maximal eigenvalue of $(\tilde{M}_g + \tilde{M}_g^{\mathsf{T}})/2$. We verify that indeed $\lambda_{max}((\tilde{M}_g + \tilde{M}_g^{\mathsf{T}})/2) < 0$ because $\bar{\mathbf{A}}$ and $\bar{\mathbf{C}}$ are p.d., and $\tau > 0$. Therefore, the real part of $\lambda$ is strictly smaller than $0$.

To analyze the convergence rate of $\tau$-GDA at $\tau = 1$. It suffies to analyze the spectral radius of $M_g$. We next apply the proof of Zhang et al. (2022, Theorem 4) to the matrix $M_g$. It implies that if $\gamma = \mu/(2L^2)$ with $\mu = \min(\lambda_{min}(\bar{\mathbf{A}}), \lambda_{min}(\bar{\mathbf{C}}))$ and $L = \max(\|\bar{\mathbf{A}}\|, \|\bar{\mathbf{B}}\|, \|\bar{\mathbf{C}}\|)$, then $\rho(I + \gamma M_g) < 1 - \mu^2/(4L^2)$. The eigenvalues of $\bar{\mathbf{A}}$, $\bar{\mathbf{B}}\bar{\mathbf{B}}^{\mathsf{T}}$ and $\bar{\mathbf{C}}$ are the same as $\mathbf{A}$, $\mathbf{B}\mathbf{B}^{\mathsf{T}}$ and $\mathbf{C}$, thus $\mu = \bar{\mu}_g$ and $L = L_g$. Therefore we obtain the spectral radius upper bound $1 - \bar{\mu}_g^2/(4L_g^2)$.

## J    DERIVATION OF DETERMINISTIC $\tau$-SGA ALGORITHM

The $\tau$-SGA algorithm modifies the vector field $\xi(x, y) = (-\delta(x, y), \tau\eta(x, y))$ of $\tau$-GDA using the anti-symmetric part of the Jacobian matrix of $\xi(x, y)$. As $\delta(x, y) = \text{grad}_x f(x, y)$ and $\eta(x, y) = \text{grad}_y f(x, y)$, the Jacobian matrix is the following linear transform on $T_x\mathcal{M}_1 \times T_y\mathcal{M}_2$ (which is a natural extension from the Euclidean case),

$$J(x, y) = \begin{pmatrix} -\tilde{\mathbf{C}}(x, y) & -\tilde{\mathbf{B}}(x, y) \\ \tau\tilde{\mathbf{B}}^{\mathsf{T}}(x, y) & -\tau\tilde{\mathbf{A}}(x, y) \end{pmatrix} = \begin{pmatrix} -\text{Hess}_x(x, y) & -\text{grad}^2_{yx}f(x, y) \\ \tau\text{grad}^2_{xy}f(x, y) & \tau\text{Hess}_y(x, y) \end{pmatrix}. \tag{47}$$

Here we introduce the notation $\tilde{\mathbf{A}}, \tilde{\mathbf{B}}, \tilde{\mathbf{C}}$ in (47) to simplify the equation. The $\tau$-SGA update rule is obtained from [1]

$$\xi(x, y) + \mu\left(\frac{J(x, y) - J^{\mathsf{T}}(x, y)}{2}\right)\xi(x, y)$$

$$= \xi(x, y) + \mu\frac{\tau+1}{2}\begin{pmatrix} 0 & -\tilde{\mathbf{B}}(x, y) \\ \tilde{\mathbf{B}}^{\mathsf{T}}(x, y) & 0 \end{pmatrix}\xi(x, y)$$

$$= \left[\begin{pmatrix} -\delta \\ \tau\eta \end{pmatrix} + \mu\frac{\tau+1}{2}\begin{pmatrix} -\tau\tilde{\mathbf{B}}[\eta] \\ -\tilde{\mathbf{B}}^{\mathsf{T}}[\delta] \end{pmatrix}\right](x, y).$$

### J.1    PROOF OF PROPOSITION 3.1

We aim to show that the correction term, which is proportional to $(\tau\tilde{\mathbf{B}}[\eta], \tilde{\mathbf{B}}^{\mathsf{T}}[\delta])$, is orthogonal to the $\tau$-GDA direction $(-\delta, \tau\eta)$ at each $(x, y)$, under the Riemannian metric on the tangent space $T_x\mathcal{M}_1 \times T_y\mathcal{M}_2$.

We follow the proof in Appendix B, by using a local coordinate chart $(O_1 \times O_2, \varphi_1 \times \varphi_2)$ around the point $(x, y)$ rather than $(x^*, y^*)$. As in (17) and (18), this coordinate chart maps $\delta(x, y)$ and $\eta(x, y)$ to their local coordinates $\bar{\delta}(u_1, u_2)$ and $\bar{\eta}(u_1, u_2)$. It also induces a canonical basis $\{E_{1,i}(x)\}_{i\leq d_1}$ on $T_x\mathcal{M}_1$ and $\{E_{2,j}(y)\}_{j\leq d_2}$ on $T_y\mathcal{M}_2$ . By the definition of cross-gradients,

$$\tilde{\mathbf{B}}(x, y)[\eta(x, y)] = D_y\text{grad}_x f(x, y)[\eta(x, y)]$$
$$= D_y\delta(x, y)[\eta(x, y)]$$
$$= \sum_i\left(\frac{\partial\bar{\delta}_i}{\partial u_2}(u_1, u_2)\bar{\eta}(u_1, u_2)\right)E_{1,i}(x).$$

Using the Riemannian metric $\bar{g}_1$ represented in the local coordinate as in (19), it turns out that

$$\langle\tilde{\mathbf{B}}(x, y)[\eta(x, y)], \delta(x, y)\rangle_x = \bar{\delta}(u_1, u_2)^{\mathsf{T}}\bar{g}_1(u_1)\left(\frac{\partial\bar{\delta}}{\partial u_2}(u_1, u_2)\bar{\eta}(u_1, u_2)\right). \tag{48}$$

Similarly we have

$$\langle\tilde{\mathbf{B}}^{\mathsf{T}}(x, y)[\delta(x, y)], \eta(x, y)\rangle_y = \bar{\eta}(u_1, u_2)^{\mathsf{T}}\bar{g}_2(u_2)\left(\frac{\partial\bar{\eta}}{\partial u_1}(u_1, u_2)\bar{\delta}(u_1, u_2)\right). \tag{49}$$

We conclude that (48) equals to (49) because as in (32),(33)

$$\bar{g}_1(u_1)\frac{\partial\bar{\delta}}{\partial u_2}(u_1, u_2) = \left(\bar{g}_2(u_2)\frac{\partial\bar{\eta}}{\partial u_1}(u_1, u_2)\right)^{\mathsf{T}}.$$

## K    PROOF OF THEOREM 3.4

Since $f$ is twice continuously differentiable, we apply Theorem 3.1 to analyze the induced dynamics $\bar{\mathbf{T}}$ (where $\mathbf{T}$ is the update rule of Asymptotic $\tau$-SGA). Following (40), we obtain

$$\bar{\mathbf{T}}(u_1, u_2) = \begin{pmatrix} \bar{\mathcal{R}}_1\left(u_1, -\gamma\left[\bar{\delta} + \mu\frac{(\tau+1)\tau}{2}\frac{\partial\bar{\delta}}{\partial u_2}\bar{\eta}\right](u_1, u_2)\right) \\ \bar{\mathcal{R}}_2\left(u_2, \gamma\tau\bar{\eta}(u_1, u_2)\right) \end{pmatrix}. \tag{50}$$

---

[1]Note that in the original SGA rule (Letcher et al., 2019, Proposition 5) the transpose of $\frac{J(x,y)-J^{\mathsf{T}}(x,y)}{2}$ is considered since their definition of $J$ has a sign difference compared to the $J$ in (47).

From (50), we compute the Jacobian matrix $\bar{\mathbf{T}}'(u_1^*, u_2^*) = I + \gamma M_s'$, where

$$M_s' = \begin{pmatrix} -\frac{\partial \bar{\delta}}{\partial u_1}(u_1^*, u_2^*) & -\frac{\partial \bar{\delta}}{\partial u_2}(u_1^*, u_2^*) \\ \tau \frac{\partial \bar{\eta}}{\partial u_1}(u_1^*, u_2^*) & \tau \frac{\partial \bar{\eta}}{\partial u_2}(u_1^*, u_2^*) \end{pmatrix} - \mu \frac{(\tau+1)\tau}{2} \begin{pmatrix} [\frac{\partial \bar{\delta}}{\partial u_2} \frac{\partial \bar{\eta}}{\partial u_1}](u_1^*, u_2^*) & [\frac{\partial \bar{\delta}}{\partial u_2} \frac{\partial \bar{\eta}}{\partial u_2}](u_1^*, u_2^*) \\ 0 & 0 \end{pmatrix}.$$

This can be reduced to the analysis of the eigenvalues of a similar matrix $M_s$ as in (42),

$$M_s = \begin{pmatrix} -\bar{\mathbf{C}} & -\bar{\mathbf{B}} \\ \tau \bar{\mathbf{B}}^{\mathsf{T}} & -\tau \bar{\mathbf{A}} \end{pmatrix} + \mu \frac{(\tau+1)\tau}{2} \begin{pmatrix} -\bar{\mathbf{B}}\bar{\mathbf{B}}^{\mathsf{T}} & \bar{\mathbf{B}}\bar{\mathbf{A}} \\ 0 & 0 \end{pmatrix}.$$

As in the proof of Theorem 3.2, we verify that $M_s$ and the $\mathbf{M}_s$ in (14) have the same eigenvalues. From the assumption of Theorem 3.4, we verify that $\tau > \min(\|\bar{\mathbf{C}}\|, \|\bar{\mathbf{C}} + \theta \bar{\mathbf{B}}\bar{\mathbf{B}}^{\mathsf{T}}\|)/\lambda_{min}(\bar{\mathbf{A}})$, and $0 \le \theta \le 1/\lambda_{max}(\bar{\mathbf{A}})$. We can therefore apply Proposition K.1 (see next) to conclude that the real-part of each eigenvalue of $\mathbf{M}_s$ is strictly negative. Therefore for $0 < \gamma < \gamma^\bullet(\mathbf{M}_s)$, Asymptotic $\tau$-SGA is locally convergent to DSE with rate $\rho(I + \gamma \mathbf{M}_s)$.

Before we proceed to analyze $M_s$, we remark that the non-asymptotic analysis of $\tau$-SGA remains an interesting open question. Indeed, if we want to analyze $\tau$-SGA beyond the asymptotic regime (e.g. $\tau$ is small), one needs to consider this induced dynamics $\bar{\mathcal{R}}_2 \left( u_2, \gamma \left[ \tau \bar{\eta} - \mu \frac{\tau+1}{2} \frac{\partial \bar{\eta}}{\partial u_1} \bar{\delta} \right] (u_1, u_2) \right)$ for the variable $u_2$. The analysis of the spectral radius of the Jacobian matrix $\bar{\mathbf{T}}'(u_1^*, u_2^*)$ is harder as one could not easily adapt the proof of Proposition K.1 to this case.

## K.1 SPECTRAL ANALYSIS OF $M_s$

The following proposition is adapted from Proposition H.1 to analyze the eigenvalues of $M_s$.

**Proposition K.1.** *Under Assumption H.1, $\tau > \frac{\min(\|C\|, \|C+\theta BB^{\mathsf{T}}\|)}{\lambda_{min}(A)}$, and $\mu = \theta \frac{2}{\tau(\tau+1)}$ with $0 \le \theta \le 1/\lambda_{max}(A)$, any eigenvalue $\lambda = \lambda_0 + i\lambda_1$ (where $\lambda_0 \in \mathbb{R}, \lambda_1 \in \mathbb{R}$) of the following matrix*

$$M = \begin{pmatrix} -C & -B \\ \tau B^{\mathsf{T}} & -\tau A \end{pmatrix} + \mu \frac{(\tau+1)\tau}{2} \begin{pmatrix} -BB^{\mathsf{T}} & BA^{\mathsf{T}} \\ 0 & 0 \end{pmatrix}$$

*satisfies $\lambda_0 < 0$.*

*Proof.* We show that $\lambda_0 \ge 0$ will lead to a contradiction. Assume that $\lambda$ is an eigenvalue of $M$, i.e. $\det(\lambda I - M) = 0$. By following the proof of Proposition H.1, we have

$$\det(\lambda I - M) = \det \begin{pmatrix} C + \theta BB^{\mathsf{T}} + \lambda I & B - \theta BA \\ -\tau B^{\mathsf{T}} & \tau A + \lambda I \end{pmatrix}$$

$$= \det(\lambda I + \tau A)\det(H(\lambda)) \tag{51}$$

where $H(\lambda) = \lambda I + C + \theta BB^{\mathsf{T}} + (B - \theta BA)(\lambda/\tau I + A)^{-1}B^{\mathsf{T}}$. As $\lambda I + \tau A$ is invertible, $\det(\lambda I - M) = 0$ implies that $\det(H(\lambda)) = 0$.

Let the spectral decomposition of $A$ be $U\Lambda_A U^{\mathsf{T}}$, with orthogonal $U \in \mathbb{R}^{m \times m}$ and diagonal $\Lambda_A = \mathrm{diag}(\lambda_1(A), \cdots, \lambda_m(A)) \in \mathbb{R}^{m \times m}$. Then

$$H(\lambda) = \lambda I + C + \theta BB^{\mathsf{T}} + \tilde{B}D\tilde{B}^{\mathsf{T}}, \tag{52}$$

with $\tilde{B} = BU$ and $D = \mathrm{diag}(d_1, \cdots, d_m)$ where

$$d_k = \frac{1 - \theta \lambda_k(A)}{\lambda/\tau + \lambda_k(A)} = (1 - \theta \lambda_k(A)) \frac{\lambda_0/\tau + \lambda_k(A) - i\lambda_1/\tau}{(\lambda_0/\tau + \lambda_k(A))^2 + (\lambda_1/\tau)^2}, \quad 1 \le k \le m.$$

It follows that if $\lambda_0 > \min(\|C\|, \|C + \theta BB^{\mathsf{T}}\|)$, the real-part of $H(\lambda)$ is

$$\mathrm{Re}(H(\lambda)) = \lambda_0 I + C + \theta BB^{\mathsf{T}} + \tilde{B}\mathrm{Re}(D)\tilde{B}^{\mathsf{T}} \ge \lambda_0 I + C + \theta BB^{\mathsf{T}} \quad \text{p.d} \tag{53}$$

This is contradictory to the fact that $\det(H(\lambda)) = 0$ (Li et al., 2022, Corollary 10.2).

On the other hand, if $0 \le \lambda_0 \le \min(\|C\|, \|C + \theta BB^\mathsf{T}\|)$ and $\lambda_1 \ne 0$, we consider for $\beta \in \mathbb{R}$,

$$\text{Re}(H(\lambda)) + \frac{\tau\beta}{\lambda_1}\text{Im}(H(\lambda)) = \lambda_0 I + C + \theta BB^\mathsf{T} + \tilde{B}\text{Re}(D)\tilde{B}^\mathsf{T} + \frac{\tau\beta}{\lambda_1}(\lambda_1 I + \tilde{B}\text{Im}(D)\tilde{B}^\mathsf{T})$$

$$= (\lambda_0 + \tau\beta)I + C + \theta BB^\mathsf{T} + \tilde{B}F\tilde{B}^\mathsf{T}, \tag{54}$$

where $F = \text{diag}(f_1, \cdots, f_m)$ with

$$f_k = (1 - \theta\lambda_k(A))\frac{\lambda_0/\tau + \lambda_k(A) - \beta}{(\lambda_0/\tau + \lambda_k(A))^2 + (\lambda_1/\tau)^2}, \quad 1 \le k \le m. \tag{55}$$

Take $\beta = \lambda_{min}(A)$, then $f_k \ge 0$ for each $k \le m$. The condition $\tau\lambda_{min}(A) > \min(\|C\|, \|C + \theta BB^\mathsf{T}\|)$ implies that $(\lambda_0 + \tau\beta)I + C + \theta BB^\mathsf{T}$ is p.d. and together with (55), we get that (54) is p.d., so it is contradictory to the fact that $\det(H(\lambda)) = 0$ (Li et al., 2022, Lemma 10.1).

Lastly, if $0 \le \lambda_0 \le \min(\|C\|, \|C + \theta BB^\mathsf{T}\|)$ and $\lambda_1 = 0$, we have from (53)

$$H(\lambda) = \lambda_0 I + C + \theta BB^\mathsf{T} + \tilde{B}D\tilde{B}^\mathsf{T}$$

with $d_k = \frac{1 - \theta\lambda_k(A)}{\lambda_0/\tau + \lambda_k(A)} = \frac{1 - \theta\lambda_k(A)}{\lambda_k(A)} - \frac{(1-\theta\lambda_k(A))\lambda_0/\tau}{(\lambda_0/\tau + \lambda_k(A))\lambda_k(A)} \ge (1 - \theta\lambda_k(A))(\frac{1}{\lambda_k(A)} - \frac{\lambda_0/\tau}{\lambda_{min}(A)\lambda_k(A)})$. As $\tau > \min(\|C\|, \|C + \theta BB^\mathsf{T}\|)/\lambda_{min}(A)$ and $0 \le \lambda_0 \le \min(\|C\|, \|C + \theta BB^\mathsf{T}\|)$ we have $0 \le \frac{\lambda_0/\tau}{\lambda_{min}(A)} < 1$ and it follows

$$H(\lambda) \ge \lambda_0 I + C + \theta BB^\mathsf{T} + \left(1 - \frac{\lambda_0/\tau}{\lambda_{min}(A)}\right)BA^{-1}B^\mathsf{T} - \theta(1 - \frac{\lambda_0/\tau}{\lambda_{min}(A)})\tilde{B}\tilde{B}^\mathsf{T}$$

$$= (1 - \lambda_0 c_0)(C + BA^{-1}B^\mathsf{T}) + \lambda_0(c_0 C + c_0 \theta BB^\mathsf{T} + I)$$

$$\ge (1 - \lambda_0 c_0)(C + BA^{-1}B^\mathsf{T}) \tag{56}$$

where $c_0 = \frac{1}{\tau\lambda_{min}(A)}$. The last inequity (56) is due to $\lambda_0 \ge 0$ and the condition $1 > c_0\min(\|C\|, \|C + \theta BB^\mathsf{T}\|)$. They imply that $\lambda_0(c_0 C + c_0\theta BB^\mathsf{T} + I) \ge 0$. As $1 - \lambda_0 c_0 \in (0, 1]$, (56) implies that $H(\lambda)$ is p.d which is contradictory. In conclusion, $\lambda_0 < 0$. □

### K.2 LOCAL CONVERGENCE RATE OF ASYMPTOTIC $\tau$-SGA

The local convergence analysis in Proposition H.2 provides an upper bound of the rate $\rho(I + \gamma M_g)$ of $\tau$-GDA. This section extends this result to Asymptotic $\tau$-SGA. We aim to obtain an upper bound of the rate $\rho(I + \gamma M_s)$ which is smaller than that of $\rho(I + \gamma M_g)$.

The following lemma analyzes the eigenvalues of $M_s$ for Asymptotic $\tau$-SGA when $\mu = \theta\frac{2}{\tau(\tau+1)}$. It refines the eigenvalue bounds of Li et al. (2022, Lemma 5.2) on $M_g$ of $\tau$-GDA (when $\mu = 0$).

**Lemma K.1.** *Under Assumption H.1, $\tau > 0$, and $0 \le \theta \le 1/\lambda_{max}(A)$, any eigenvalue $\lambda = \lambda_0 + i\lambda_1$ (where $\lambda_0 \in \mathbb{R}, \lambda_1 \in \mathbb{R}$) of the following matrix*

$$M = \begin{pmatrix} -C & -B \\ \tau B^\mathsf{T} & -\tau A \end{pmatrix} + \theta\begin{pmatrix} -BB^\mathsf{T} & BA^\mathsf{T} \\ 0 & 0 \end{pmatrix}$$

*satisfies*

a. $|\lambda_1| \le \sqrt{\tau}\sqrt{1 - \theta\lambda_{min}(A)}\|B\|$.

b. *If $\lambda_1 \ne 0$, we have $\lambda_0 \le -\lambda_+$ with $\lambda_+ = \frac{1}{2}(\lambda_{min}(A)\tau - \min(\|C\|, \|C + \theta BB^\mathsf{T}\|))$.*

c. *Let $\tau = \frac{\min(\|C\|, \|C+\theta BB^\mathsf{T}\|)+\alpha}{\lambda_{min}(A)}$ with $\alpha > 0$. If $\lambda_1 = 0$, we have $\lambda_0 \le -\lambda'_+$ with $\lambda'_+ = \min(\lambda_{min}(C + BA^{-1}B^\mathsf{T}), \min(\|C\|, \|C + \theta BB^\mathsf{T}\|) + \alpha)$.*

d. *Let $L = \max(\|A\|, \|B\|, \|C + \theta BB^\mathsf{T}\|)$. If $\tau \ge 1$, we have $\lambda_0^2 + \lambda_1^2 \le \|M\|^2 \le 4\tau^2 L^2$.*

*Proof.* Assume that $\lambda$ is an eigenvalue of $M$, i.e. $\det(\lambda I - M) = 0$. If $\lambda I + \tau A$ is invertible, we can compute $\det(\lambda I - M)$ using the Schur complement (51) to obtain

$$\det(\lambda I - M) = \det(\lambda I + \tau A)\det(H(\lambda)).$$

Let the spectral decomposition of $A$ be $U\Lambda_A U^\mathsf{T}$, with orthogonal $U \in \mathbb{R}^{m \times m}$ and diagonal $\Lambda_A$. Then $H(\lambda)$ can be rewritten into (52).

Part a: If $|\lambda_1| > \sqrt{\tau}\sqrt{1 - \theta\lambda_{min}(A)}\|B\|$, then $\lambda I + \tau A$ is invertible since $\lambda_1 I$ is. Therefore $\det(H(\lambda)) = 0$. Consider

$$
\begin{aligned}
\frac{1}{\lambda_1}\text{Im}(H(\lambda)) &= I + \frac{1}{\lambda_1}\tilde{B}\text{Im}(D)\tilde{B}^\mathsf{T} \\
&= I - \frac{1}{\tau}\tilde{B}\text{diag}\left(\frac{1 - \theta\lambda_k(A)}{(\lambda_0/\tau + \lambda_k(A))^2 + (\lambda_1/\tau)^2}\right)_{k \le m}\tilde{B}^\mathsf{T} \\
&\ge I - \frac{1}{\tau}\frac{\tau^2}{\lambda_1^2}(1 - \theta\lambda_{min}(A))\tilde{B}\tilde{B}^\mathsf{T} = I - \frac{\tau}{\lambda_1^2}(1 - \theta\lambda_{min}(A))BB^\mathsf{T}
\end{aligned}
$$

As $|\lambda_1| > \sqrt{\tau}\sqrt{1 - \theta\lambda_{min}(A)}\|B\|$, we have that $I - \frac{\tau}{\lambda_1^2}(1 - \theta\lambda_{min}(A))BB^\mathsf{T}$ is p.d. and therefore $\frac{1}{\lambda_1}\text{Im}(H(\lambda))$ is p.d., which leads to a contradiction.

Part b: Assume $\lambda_0 > -\lambda_+$. As $\lambda_1 \ne 0$, $\lambda I + \tau A$ is invertible and $\det(H(\lambda)) = 0$. Following (54), we consider

$$
\text{Re}(H(\lambda)) + \frac{\tau\beta}{\lambda_1}\text{Im}(H(\lambda)) = (\lambda_0 + \tau\beta)I + C + \theta BB^\mathsf{T} + \tilde{B}F\tilde{B}^\mathsf{T}.
$$

Take $\beta = \frac{1}{2\tau}(\lambda_{min}(A)\tau + \min(\|C\|, \|C + \theta BB^\mathsf{T}\|)) > 0$, then $\tilde{B}F\tilde{B}^\mathsf{T}$ is semi-p.d (positive semidefinite) because each $f_k$ in (55) is larger than zero ($f_k \ge 0$). Indeed, $\forall 1 \le k \le m$,

$$
\begin{aligned}
\lambda_0/\tau + \lambda_k(A) - \beta &> -\lambda_+/\tau + \lambda_k(A) - \beta \\
&= -\frac{\lambda_{min}(A)\tau - \min(\|C\|, \|C + \theta BB^\mathsf{T}\|)}{2\tau} - \beta + \lambda_k(A) \\
&= \lambda_k(A) - \lambda_{min}(A) \ge 0.
\end{aligned}
$$

Furthermore, either $(\lambda_0 + \tau\beta)I + C$ or $(\lambda_0 + \tau\beta)I + C + \theta BB^\mathsf{T}$ is p.d. due to the following:

- If $\|C\| < \|C + \theta BB^\mathsf{T}\|$, we have $\lambda_0 + \tau\beta > \|C\|$ because $-\lambda_+ + \tau\beta = \|C\|$.

- If $\|C\| \ge \|C + \theta BB^\mathsf{T}\|$, we have $\lambda_0 + \tau\beta > \|C + \theta BB^\mathsf{T}\|$ because $-\lambda_+ + \tau\beta = \|C + \theta BB^\mathsf{T}\|$.

As a consequence, $\text{Re}(H(\lambda)) + \frac{\tau\beta}{\lambda_1}\text{Im}(H(\lambda))$ is p.d. which is a contradiction. Therefore $\lambda_0 \le -\lambda_+$.

Part c: From Proposition K.1, we know that $\lambda_0 < 0$. If $\lambda_1 = 0$ and $0 > \lambda_0 > -\lambda'_+$, $\lambda_0 I + \tau A$ remains p.d. since

$$
\lambda_0 I + \tau A \ge (\tau\lambda_{min}(A) + \lambda_0)I > (\min(\|C\|, \|C + \theta BB^\mathsf{T}\|) + \alpha - \lambda'_+)I \quad \text{semi-p.d.}
$$

Therefore $\lambda I + \tau A$ is invertible and $\det(H(\lambda)) = 0$. Following (52), we have

$$
\begin{aligned}
H(\lambda) &= \lambda_0 I + C + \theta BB^\mathsf{T} + \tilde{B}\text{diag}\left(\frac{1 - \theta\lambda_k(A)}{\lambda_0/\tau + \lambda_k(A)}\right)_{k \le m}\tilde{B}^\mathsf{T} \\
&= \lambda_0 I + C + \theta BB^\mathsf{T} + BA^{-1}B^\mathsf{T} - \tilde{B}\text{diag}\left(\frac{\lambda_0/\tau + \theta\lambda_k^2(A)}{(\lambda_0/\tau + \lambda_k(A))\lambda_k(A)}\right)_{k \le m}\tilde{B}^\mathsf{T} \quad\quad (57) \\
&= \lambda_0 I + C + BA^{-1}B^\mathsf{T} - \tilde{B}\text{diag}\left(\frac{(1 - \theta\lambda_k(A))\lambda_0/\tau}{(\lambda_0/\tau + \lambda_k(A))\lambda_k(A)}\right)_{k \le m}\tilde{B}^\mathsf{T} \quad\quad (58) \\
&> C + BA^{-1}B^\mathsf{T} - \lambda'_+ I \\
&\ge C + BA^{-1}B^\mathsf{T} - \lambda_{min}(C + BA^{-1}B^\mathsf{T})I \quad \text{semi-p.d.}
\end{aligned}
$$

As a consequence, $H(\lambda)$ is p.d. which is a contradiction. Therefore $\lambda_0 \le -\lambda'_+$.

Part d: Note that $\|I - \theta A\| \leq 1$. For any $(x, y) \in \mathbb{R}^n \times \mathbb{R}^m$, we compute

$$
\left\| M \begin{pmatrix} x \\ y \end{pmatrix} \right\| = \left\| \begin{matrix} -Cx - By - \theta BB^{\mathsf{T}}x + \theta BA^{\mathsf{T}}y \\ \tau B^{\mathsf{T}}x - \tau Ay \end{matrix} \right\|
$$

$$
= \sqrt{\| -Cx - By - \theta BB^{\mathsf{T}}x + \theta BA^{\mathsf{T}}y \|^2 + \|\tau B^{\mathsf{T}}x - \tau Ay\|^2}
$$

$$
\leq \sqrt{2(\|(C + \theta BB^{\mathsf{T}})x\|^2 + \|(B - \theta BA^{\mathsf{T}})y\|^2) + 2\tau^2(\|B^{\mathsf{T}}x\|^2 + \|Ay\|^2)}
$$

$$
\leq \sqrt{2\|C + \theta BB^{\mathsf{T}}\|^2 \|x\|^2 + 2\|B\|^2 \|I - \theta A\|^2 \|y\|^2 + 2\tau^2 \|B^{\mathsf{T}}\|^2 \|x\|^2 + 2\tau^2 \|A\|^2 \|y\|^2}
$$

$$
\leq \sqrt{2(1 + \tau^2)L^2(\|x\|^2 + \|y\|^2)} \leq 2\tau L \left\| \begin{pmatrix} x \\ y \end{pmatrix} \right\|.
$$

Therefore $\|M\| \leq 2\tau L$ and $\lambda_0^2 + \lambda_1^2 \leq \|M\|^2 \leq 4\tau^2 L^2$. $\qquad\square$

From Lemma K.1, we are ready to obtain an upper bound of the local convergence rate of Asymptotic $\tau$-SGA. Recall that $L_s = \max(\|\mathbf{A}\|, \|\mathbf{B}\|, \|\mathbf{C} + \theta\mathbf{BB}^{\mathsf{T}}\|)$ and $\mu_s = \min(L_s, \lambda_{min}(\mathbf{C} + \mathbf{BA}^{-1}\mathbf{B}^{\mathsf{T}}))$.

**Proposition K.2.** *Assume $(x^*, y^*)$ is a DSE of $f \in C^2$. If $\tau \geq \frac{2L_s}{\lambda_{min}(\mathbf{A})}$ and $\gamma = \frac{1}{4\tau L_s}$, we have $\rho(I + \gamma M_s) \leq 1 - \frac{\mu_s}{16\tau L_s}$.*

*Proof.* Let $L = \max(\|\bar{\mathbf{A}}\|, \|\bar{\mathbf{B}}\|, \|\bar{\mathbf{C}} + \theta\bar{\mathbf{B}}\bar{\mathbf{B}}^{\mathsf{T}}\|)$ and $\mu_x = \min(L, \lambda_{min}(\bar{\mathbf{C}} + \bar{\mathbf{B}}\bar{\mathbf{A}}^{-1}\bar{\mathbf{B}}^{\mathsf{T}}))$. We verify that $\lambda_{min}(\mathbf{A}) = \lambda_{\min}(\bar{\mathbf{A}})$, $L = L_s$ and $\mu_x = \mu_s$. Let $\lambda = \lambda_0 + i\lambda_1$ be an eigenvalue of $M_s$. It follows that we only need to show that $|1 + \gamma\lambda| \leq 1 - \frac{\mu_x}{16\tau L}$.

To apply Lemma K.1, we denote $A = \bar{\mathbf{A}}$, $B = \bar{\mathbf{B}}$ and $C = \bar{\mathbf{C}}$. We rewrite that $\tau\lambda_{min}(A) = \min(\|C\|, \|C + \theta BB^{\mathsf{T}}\|) + \alpha$ with $\alpha \geq 2L - \min(\|C\|, \|C + \theta BB^{\mathsf{T}}\|) > 0$. There are two cases to verify:

- Case $\lambda_1 = 0$: we have $\lambda_0 \leq -\lambda'_+$. Since $\tau\lambda_{min}(A) > L$, we have $\lambda'_+ = \min(\tau\lambda_{min}(A), \lambda_{min}(C + BA^{-1}B^{\mathsf{T}})) \geq \min(L, \lambda_{min}(C + BA^{-1}B^{\mathsf{T}})) = \mu_x$. Thus $\lambda_0 \leq -\mu_x$ and $1 + \gamma\lambda_0 \leq 1 - \frac{1}{4\tau L}\mu_x$. On the other hand, $\tau \geq 2$ and $\lambda_0 \geq -2\tau L$, thus $1 + \gamma\lambda_0 \geq 1 - \frac{1}{4\tau L}2\tau L = 1/2$. Therefore $|1 + \gamma\lambda_0| \leq 1 - \frac{1}{4\tau L}\mu_x \leq 1 - \frac{\mu_x}{16\tau L}$.

- Case $\lambda_1 \neq 0$: we know that $-2\tau L \leq \lambda_0 \leq -\lambda_+ = -\frac{\alpha}{2}$ and $|\lambda_1| \leq \sqrt{\tau}L$. Note that $\alpha = \tau\lambda_{min}(A) - \min(\|C\|, \|C + \theta BB^{\mathsf{T}}\|) \geq \tau\lambda_{min}(A) - L$ and by assumption $L \leq \frac{\tau\lambda_{min}(A)}{2}$. Thus $\alpha \geq \frac{\tau\lambda_{min}(A)}{2}$. It follows that

$$
|1 + \gamma\lambda|^2 = (1 + \gamma\lambda_0)^2 + \gamma^2\lambda_1^2
$$

$$
\leq \left(1 - \gamma\frac{\alpha}{2}\right)^2 + \gamma^2\tau L^2
$$

$$
= \left(1 - \frac{\alpha}{8\tau L}\right)^2 + \frac{1}{16\tau}
$$

$$
\leq \left(1 - \frac{\lambda_{min}(A)}{16L}\right)^2 + \frac{\lambda_{min}(A)}{32L} \leq 1 - \frac{\lambda_{min}(A)}{16L}.
$$

Therefore $|1 + \gamma\lambda| \leq \sqrt{1 - \frac{\lambda_{min}(A)}{16L}} \leq 1 - \frac{\lambda_{min}(A)}{32L} \leq 1 - \frac{L}{16\tau L} \leq 1 - \frac{\mu_x}{16\tau L}$.

$\qquad\square$

## L  COMPUTATIONAL EFFICIENCY OF $\tau$-SGA

We first discuss the computation of $\tilde{\mathbf{B}}[\eta]$ and $\tilde{\mathbf{B}}^{\mathsf{T}}[\delta]$ when $\mathcal{M}_1$ (resp. $\mathcal{M}_2$) is an embedded sub-manifold of $\mathbb{R}^{d'_1}$ (resp. $\mathbb{R}^{d'_2}$). We then propose a linear-time computational procedure using auto-differentiation when $\mathcal{M}_1$ is Euclidean, which is applicable to orthogonal Wasserstein GANs. When $\mathcal{M}_2$ is also Euclidean, this procedure is equivalent to the one proposed in Balduzzi et al. (2018).

To compute $\tilde{\mathbf{B}}[\eta]$ at $(x, y)$, we first fix $\eta = \eta(x, y) \in T_y \mathcal{M}_2$. From the property of embedded sub-manifolds, we use $\leftrightarrow$ to identify a tangent vector $\delta \in T_x \mathcal{M}_1$ with a vector $\delta' = (\delta_i')_{i \le d_1'} \in \mathbb{R}^{d_1'}$ (resp. $\eta \in T_y \mathcal{M}_2$ with $\eta' = (\eta_i')_{i \le d_2'} \in \mathbb{R}^{d_2'}$).

This allows one to compute the cross-gradients in the embedded space, by

$$\tilde{\mathbf{B}}[\eta](x, y) = D_y \mathrm{grad}_x f(x, y)[\eta] \quad \leftrightarrow \quad D_y \delta'(x, y)[\eta'] = (\langle (\mathrm{grad}_y \delta_i'(x, y))', \eta' \rangle_y)_{i \le d_1'} \in \mathbb{R}^{d_1'}.$$

Note that $\langle (\mathrm{grad}_y \delta_i'(x, y))', \eta' \rangle_y$ is computed on $\mathbb{R}^{d_2'}$ with a metric induced from $T_y \mathcal{M}_2$. Assume that its computational time is $O(d_2')$ for each $i$. Then it takes $O(d_2' d_1')$ to compute $\tilde{\mathbf{B}}[\eta](x, y)$ in the embedded space. We can obtain a similar cost for $\tilde{\mathbf{B}}^\intercal[\delta](x, y)$.

**Euclidean $\mathcal{M}_1$ case** When $\mathcal{M}_1 = \mathbb{R}^{d_1}$, the computational complexity of $\tilde{\mathbf{B}}[\eta](x, y)$ can be significantly reduced: $\forall i \le d_1$,

$$D_y \partial_{x_i} f(x, y)[\eta] = \partial_{x_i} \langle \mathrm{grad}_y f(x, y), \eta \rangle_y \quad \leftrightarrow \quad \partial_{x_i} \langle \eta'(x, y), \eta' \rangle_y. \tag{59}$$

Importantly, one does not need to recompute $\langle \eta'(x, y), \eta' \rangle_y$ for each $i$. Therefore, the whole cost of $\tilde{\mathbf{B}}[\eta](x, y)$ is $O(d_1 + d_2')$. Note that in (59), the term $\eta'$ is "detached".

For $\tilde{\mathbf{B}}^\intercal[\delta](x, y)$, we "detach" $\delta = \delta(x, y) \in T_x \mathcal{M}_1$ and compute

$$\tilde{\mathbf{B}}^\intercal[\delta](x, y) = \sum_{i \le d_1} \partial_{x_i} \eta(x, y) \delta_i \quad \leftrightarrow \quad \partial_{\eta''} \left( \sum_{i \le d_1} \partial_{x_i} \langle \eta'(x, y), \eta'' \rangle \delta_i \right)_{|_{\eta'' = 0}}.$$

This implies that we can first compute $\partial_{x_i} \langle \eta'(x, y), \eta'' \rangle$ for each $i$ as in (59). We then compute its sum with $\delta_i$ which takes $O(d_1)$. Finally an extra auto-differentiation is taken with respect to $\eta''$ which costs $O(d_2')$. The whole cost of $\tilde{\mathbf{B}}^\intercal[\delta](x, y)$ is therefore $O(d_1 + d_2')$. In this case, we use the Euclidean metric on $\mathbb{R}^{d_2'}$ to evaluate $\langle \eta'(x, y), \eta'' \rangle$ rather than the induced Riemannian metric (59).

## L.1 Extension to stochastic $\tau$-SGA

We construct stochastic $\tau$-SGA through an unbiased estimation of the terms in the update rule of deterministic $\tau$-SGA. To achieve this, we compute $\partial_{x_i} \langle \eta'(x, y), \eta' \rangle_y$ in (59) using two mini-batches independently sampled from a training set, one to estimate $\eta'(x, y)$, the other to estimate $\eta'$. Similarly, we use these mini-batches to estimate $\eta'(x, y)$ and $\delta$ in $\tilde{\mathbf{B}}^\intercal[\delta](x, y)$.

## M Details of numerical experiments

In the stochastic-gradient setting, the expectation of $D_y(\phi_{data})$ (resp. $D_y(\phi_x)$) is estimated at each iteration of an algorithm, using a batch of samples of data in the training set (resp. a batch of samples of $Z$). In this setting, there are an infinite number of training samples from the GAN generator $\phi_x$.

### M.1 Choice of image datasets

**MNIST dataset** We consider all the 10 classes. There are 50000 training samples, 10000 validation samples and 10000 test samples.

**MNIST (digit 0) dataset** Among all the digit 0 images in the training (and validation) set of MNIST, we take 4932 as training samples, 991 as validation samples. There are 980 test samples.

**Fashion-MNIST dataset** We consider all the 10 classes. There are 50000 training samples, 10000 validation samples and 10000 test samples.

**Fashion-MNIST (T-shirt) dataset** Among all the T-shirt images in the training set of Fashion-MNIST, we take 4977 as training samples, 1023 as validation samples. There are 1000 test samples.

## M.2    Choice of smooth non-linearity

We aim to build GAN models whose value function $f$ is twice continuously differentiable, based on smooth non-linearities studied in Biswas et al. (2022). For the discriminator of Gaussian distribution, $\rho$ is a smooth approximation of the absolute value non-linearity of the form

$$\sigma(a) = \sqrt{a^2 + \epsilon^2}, \quad \epsilon = 10^{-6}.$$

This function is twice continuously differentiable since $\sigma''(a) = \frac{\epsilon^2}{2(a^2 + \epsilon^2)^{3/2}}$. Similarly, for the discriminator in the image modeling, we use a smooth ReLu non-linearity

$$\sigma(a) = (a + \sqrt{a^2 + \epsilon^2})/2.$$

## M.3    DCGAN generator

We use a smooth DCGAN generator to model the images from the MNIST and Fashion-MNIST datasets, adapted from WGAN-GP (Gulrajani et al., 2017).[2] We consider this generator because there is no batch normalization module needed (this module is typically used for other datasets such as CIFAR-10). We make a slight modification of the default DCGAN so that the function $x \mapsto G_x(Z)$ is twice continuously differentiable at any $x \in \mathcal{M}_1$ for a fixed $Z$. For this, each ReLu non-linearity in $Z \mapsto G_x(Z)$ is replaced by the smooth ReLu non-linearity in Appendix M.2.

## M.4    Scattering CNN discriminator for image modeling

We construct a smooth Lipschitz-continuous discriminator with one trainable layer

$$D_y(\phi) = \langle v_y, \sigma(w_y \star P(\phi) + b_y) \rangle,$$

where $\sigma$ is the smooth ReLu defined in Appendix M.2. For a fixed $\phi$, this makes the function $y \mapsto D_y(\phi)$ twice continuously differentiable at any $y \in \mathcal{M}_2$. However, the function $\phi \mapsto D_y(\phi)$ is not everywhere twice continuously differentiable due to the modulus non-linearity in the scattering transform. We therefore replace this modulus non-linearity $z \mapsto |z| = |z_{re} + iz_{im}|$ by $z \mapsto \sqrt{z_{re}^2 + z_{im}^2 + \epsilon^2}$ for each complex number input $z = z_{re} + iz_{im}$. This makes the function $\phi \mapsto D_y(\phi)$ twice continuously differentiable. As a consequence, the function $(x, y) \mapsto f(x, y)$ is also twice continuously differentiable, which is induced from the smoothness of $(x, y) \mapsto D_y(G_x(Z))$ and $y \mapsto D_y(\phi_{data})$.

**Scattering transform**    The input $\phi$ with dimension $d = 784$ is represented as an image of size $28 \times 28$. It is pre-processed by the wavelet scattering transform $P(\phi)$ to extract stable edge-like information using Morlet wavelet at different orientations and scales. We use the second-order scattering transform with four wavelet orientations (between $[0, \pi)$) and two wavelet scales. It first computes the convolution of $\phi$ with each wavelet filter, then a smooth modulus non-linearity is applied to each feature map. This computation is repeated one more time on each obtained feature map and then a low-pass filter is applied to each of the channels. The obtained scattering features $P(\phi)$ is an image of size $9 \times 9$ with 25 channels ($I = 25, n = 9$).

**Orthogonal CNN layer**    The orthogonal CNN layer is parameterized by the kernel $w_y$ and bias $b_y$. The kernel $w_y$ has $5 \times 5$ spatial size ($k = 5$). With a suitable padding and stride (two by two), we obtain an output image of size $5 \times 5$ ($N = 5$) with $J = 256$ channels. Therefore the embedding space dimension of $v_y$ is $JN^2 = 6400$.

## M.5    Stiefel manifold geometry

For the discriminators of the Gaussian distribution and the image modeling, part of the parameters in $y$ belong to Stiefel manifolds. To choose a Riemannian metric on a Stiefel manifold (which is non-Euclidean), we use the one in Manton (2002, equation 20). We also use the SVD projection in Manton (2002, Proposition 12) as the retraction $\mathcal{R}_2$ on each Stiefel manifold.

---

[2] https://github.com/caogang/wgan-gp

## M.6 Initialization for local convergence

### M.6.1 Simultaneous $\tau$-GDA initialization for Gaussian distribution

Starting from a random initialization of $(x, y)$, we apply the stochastic $\tau$-GDA method to build a pre-trained model. It is pre-trained with batch size 1000, learning rate $\gamma = 0.0002$ and $\tau = 100$ for $T = 50000$ iterations.

### M.6.2 Alternating $\tau$-GDA initialization for MNIST and Fashion-MNIST

The stochastic alternating $\tau$-GDA (Zhang et al., 2022) is often used in the training of WGAN-GP (Gulrajani et al., 2017), and it can be extended to Riemannian manifold as the simultaneous $\tau$-GDA. Starting from a random initialization of $(x, y)$, we apply alternating $\tau$-GDA to build a pre-trained model since we observe that the simultaneous $\tau$-GDA method is unstable when $\tau$ is small; while its convergence can be slow when $\tau$ is big.

During the alternating $\tau$-GDA pre-training, each iteration amounts to perform $\tau = 5$ gradient updates of $y$ (with learning rate 0.1) and one gradient update of $x$ (with learning rate 0.1).

**MNIST**   It is pre-trained with batch size 128 for a total number of $2 \times 10^5$ iterations. We obtain a pre-trained model with FID (train) = 7.3 and FID (val) = 9.4.

**MNIST (digit 0)**   It is pre-trained with batch size 128 for a total number of $10^4$ iterations. We obtain a pre-trained model with FID (train) = 12 and FID (val) = 18.

**Fashion-MNIST**   It is pre-trained with batch size 128 for a total number of $2 \times 10^5$ iterations. We obtain a pre-trained model with FID (train) = 17.7 and FID (val) = 20.

**Fashion-MNIST (T-shirt)**   It is pre-trained with batch size 128 for $5 \times 10^4$ iterations. We obtain a pre-trained model with FID (train) = 26 and FID (val) = 40.

## M.7 Statistical estimation of the evaluation quantities

**Estimate $f$ and angle**   After training, we estimate the $f$ and angle values by re-sampling $\phi_x$ ten times (similar to the estimation of FID). To compute the angle for the scattering CNN discriminator, we use instead $\delta(x, y) = \mathbb{E}(\sigma(w_y \star P(\phi_{data}) + b_y)) - \mathbb{E}(\sigma(w_y \star P(\phi_x) + b_y))$.

**Estimate FID scores**   To compute FID (train) (resp. FID (val)), we generate the same number of fake samples as the training samples (resp. the validation samples), and report an average value by re-sampling the fake samples 10 times (with its standard deviation if needed). Similarly, we compute FID (test) using the same amount of fake samples as test samples, but without re-sampling.

## N   Extra numerical experiments

We perform extra numerical simulations on the MNIST and Fashion-MNIST datasets by using the same Wasserstein GAN architecture detailed in Section M.3 and M.4. The training on MNIST and Fashion-MNIST datasets are performed with a relatively large batch size 512 (as we consider the full dataset, this results in a smaller stochastic gradient variance), while the training on MNIST (digit 0) and Fashion-MNIST (T-shirt) datasets are performed with batch size 128.

### N.1 Extra results on MNIST

We compare the computational time between $\tau$-SGA and $\tau$-SGA in Figure 4. We see that at $\tau = 10$, $\tau$-GDA converges faster than $\tau$-SGA at the initial stage, but then in a sunden the dynamics becomes unstable, resulting in a much larger FID score. At $\tau = 20$, $\tau$-GDA is convergent and its computational speed is similar to $\tau$-SGA (at $\tau = 5$).

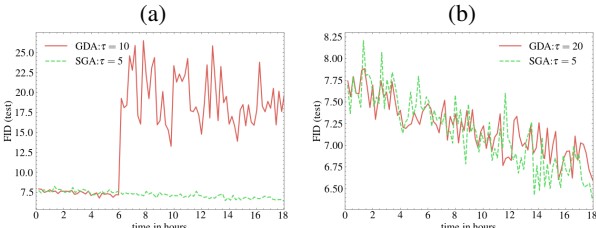

Figure 4: Evolution of the FID (test) score of stochastic $\tau$-GDA and $\tau$-SGA as a function of a wall-clock time on the Wasserstein GAN of MNIST. a): $\tau$-GDA at $\tau = 10$ vs $\tau$-SGA at $\tau = 5$. b): $\tau$-GDA at $\tau = 20$ vs $\tau$-SGA at $\tau = 5$.

Table 3: Last iteration measures of stochastic $\tau$-GDA and $\tau$-SGA on the Wasserstein GAN of MNIST (digit 0). We report the $f$, angle, and FID scores computed at $(x(T), y(T))$. $\tau$-GDA is trained for $T = 2 \times 10^4$ iterations with $\gamma = 0.05/\tau$. $\tau$-SGA is trained longer with $\gamma = 0.01$, $\tau = 5$, $\theta = 0.075$.

| | $\tau$-GDA | | | | $\tau$-SGA | | | | |
|---|---|---|---|---|---|---|---|---|---|
| $\tau$ | $f$ | angle | FID (train) | FID (val) | $T(10^5)$ | $f$ | angle | FID (train) | FID (val) |
| 5 | -0.09 | -0.26 | 21 | 29 $_{(\pm 0.9)}$ | 1 | 0.02 | 0.16 | 12 | 21 $_{(\pm 0.9)}$ |
| 10 | 0.09 | 0.11 | 34 | 36 $_{(\pm 0.6)}$ | 2 | 0.018 | 0.2 | 9.3 $_{(\pm 0.2)}$ | 15 $_{(\pm 0.5)}$ |
| 30 | 0.036 | 0.4 | 13 | 20 $_{(\pm 0.95)}$ | 3 | 0.016 | 0.2 | 8.2 $_{(\pm 0.2)}$ | 14 $_{(\pm 0.3)}$ |

## N.2 Extra results on MNIST (digit 0)

The results of $\tau$-GDA and $\tau$-SGA are reported in Table 3. For $\tau$-GDA, we vary $\tau \in \{5, 10, 30\}$. We observe that when $\tau = 30$, $\tau$-GDA has more stable dynamics than $\tau = 5$ and $\tau = 10$ because the angle stays around a positive constant. At $\tau = 5$, we observe a negative angle at $t = T$ in $\tau$-GDA because it oscillates around zero over $t$. On the other hand, a larger $\tau$ tends to slowdown the reduction of $f$ as in Figure 1(a). Furthermore, the FID scores at $T = 3 \times 10^5$ are only slightly improved compared to those at $T = 2 \times 10^4$. Facing such a dilemma, we evaluate the performance of $\tau$-SGA using $\tau = 5$ such that $\tau\gamma$ remains the same. This is the case where $\tau$-GDA is not convergent due to oscillating angles. We find that both the angle and the FID scores are significantly improved with a suitable choice of $\theta$ and $T$ in $\tau$-SGA.

Regarding the choice of $\theta$ in $\tau$-SGA, we have used this dataset to tune this parameter and then chosen a reasonable value to be used for the other datasets. Intuitively, when $\theta$ is too small, $\tau$-SGA can be as unstable as $\tau$-GDA. When $\theta$ is too big, it may amplify the stochastic gradient noise in the correction term of $\tau$-SGA. Therefore to choose a suitable $\theta$ is a delicate question. In Figure 5, we observe nevertheless that in a wide range of $\theta$, the performance of $\tau$-SGA in terms of the FID (test) score is similar. It also suggests that $\tau$-SGA could have a global convergence property when $\theta$ is small.

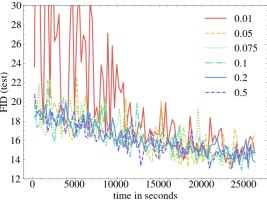

Figure 5: Evolution of the FID (test) score of stochastic $\tau$-SGA as a function of a wall-clock time on the Wasserstein GAN of MNIST (digit 0). We vary $\theta \in [0.01, 0.5]$ with a fixed $\tau = 5$.

## N.3 Extra results on Fashion-MNIST

We perform extra numerical simulations on the Fashion-MNIST dataset. In Table 4, we study the performance of $\tau$-GDA by varing the choice of $\tau$. It is run for $T = 2 \times 10^4$ iterations with $\gamma = 0.1/\tau$.

At $\tau = 1$ and $\tau = 5$, we observe a similar instability in $\tau$-GDA as the MNIST case at $\tau = 5$ and $\tau = 10$ (in Table 2). At $\tau = 10$, $\tau$-GDA has a stable dynamics and a good performance in terms of FID scores.

We next study the performance of $\tau$-SGA by varing the training iterations $T$, using the same $\gamma = 0.02$, $\tau = 5$, $\theta = 0.075$. Table 4 reaches a similar conclusion as Table 2, confirming the importance of the correction term in $\tau$-SGA to improve the local convergence of $\tau$-GDA.

Lastly, we compare the computational time between $\tau$-GDA and $\tau$-SGA in Figure 6. Different to the MNIST case in Figure 4, we find that $\tau$-GDA has a faster computational speed at $\tau = 10$, compared to $\tau$-SGA. This suggests that the speedup of the $\tau$-SGA in terms of the number of iterations is less significant in this case since $\tau$-GDA works well with a relatively small $\tau$. On the other hand, a larger $\tau = 20$ in $\tau$-GDA makes it slower.

Table 4: Last iteration measures of stochastic $\tau$-GDA and $\tau$-SGA on the Wasserstein GAN of Fashion-MNIST. We report the $f$, angle, and FID scores computed at $(x(T), y(T))$. $\tau$-GDA is trained for $T = 2 \times 10^4$ iterations with $\gamma = 0.1/\tau$. $\tau$-SGA is trained longer with $\gamma = 0.02$, $\tau = 5$, $\theta = 0.075$.

| | $\tau$-GDA | | | | $\tau$-SGA | | | | |
|---|---|---|---|---|---|---|---|---|---|
| $\tau$ | $f$ | angle | FID (train) | FID (val) | $T(10^5)$ | $f$ | angle | FID (train) | FID (val) |
| 1 | 1.60 $(\pm 0.006)$ | 0.92 | 107 | 109 $(\pm 0.3)$ | 1 | 0.019 | 0.33 $(\pm 0.01)$ | 16.98 $(\pm 0.05)$ | 19.2 $(\pm 0.17)$ |
| 5 | 0.02 | 0.48 $(\pm 0.01)$ | 17.6 $(\pm 0.08)$ | 19.8 $(\pm 0.2)$ | 3 | 0.019 | 0.26 $(\pm 0.008)$ | 16 | 18.3 $(\pm 0.1)$ |
| 10 | 0.02 | 0.4 | 17 | 19 | 5 | 0.016 | 0.39 $(\pm 0.03)$ | 14.37 $(\pm 0.05)$ | 16.6 $(\pm 0.1)$ |

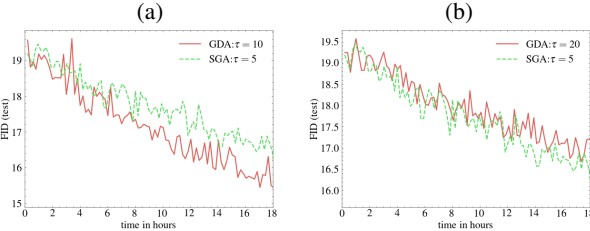

Figure 6: Computational speed of $\tau$-GDA and $\tau$-SGA as a function of a wall-clock time on the Wasserstein GAN of Fashion-MNIST. a): $\tau$-GDA at $\tau = 10$ vs $\tau$-SGA at $\tau = 5$. b): $\tau$-GDA at $\tau = 20$ vs $\tau$-SGA at $\tau = 5$.

### N.4 EXTRA RESULTS ON FASHION-MNIST (T-SHIRT)

From the results in Table 5, we see that the behavior of the $\tau$-GDA and $\tau$-SGA algorithms (in terms of the f and angle measures) are similar to the MNIST (digit 0) case. As in Table 3, we observe that one can improve the local convergence of $\tau$-GDA in the range of small $\tau$ by using $\tau$-SGA. We also find that an improved convergence (in terms of the angle) lead to improved GAN models in terms of the FID scores.

Table 5: Last iteration measures of stochastic $\tau$-GDA and $\tau$-SGA on the Wasserstein GAN of Fashion-MNIST (T-shirt). We report the $f$, angle, and FID scores computed at $(x(T), y(T))$. $\tau$-GDA is trained for $T = 2 \times 10^4$ iterations with $\gamma = 0.1/\tau$. $\tau$-SGA is trained longer with $\gamma = 0.02$, $\tau = 5$, $\theta = 0.075$.

| | $\tau$-GDA | | | | $\tau$-SGA | | | | |
|---|---|---|---|---|---|---|---|---|---|
| $\tau$ | $f$ | angle | FID (train) | FID (val) | $T(10^5)$ | $f$ | angle | FID (train) | FID (val) |
| 5 | -0.52 | -0.31 | 80 $(\pm 0.4)$ | 92 $(\pm 1)$ | 1 | 0.02 | 0.30 $(\pm 0.02)$ | 22.7 $(\pm 0.3)$ | 36.1 $(\pm 0.5)$ |
| 10 | 0.03 $(\pm 0.03)$ | 0.05 $(\pm 0.04)$ | 73 | 86 | 2 | 0.018 | 0.35 $(\pm 0.03)$ | 20.6 $(\pm 0.2)$ | 34.1 $(\pm 0.6)$ |
| 30 | 0.01 | 0.15 $(\pm 0.03)$ | 24.7 $(\pm 0.2)$ | 38.7 $(\pm 0.5)$ | 3 | 0.018 | 0.32 $(\pm 0.03)$ | 19.8 $(\pm 0.2)$ | 33.3 $(\pm 0.3)$ |

## N.5 Extra results on computational time

In Figure 7, we compare the computation time of $\tau$-GDA and $\tau$-SGA. The best performed methods are selected to compare at the last iteration, according to the value of $f$ in Example 2 or the FID (val) score on MNIST (digit 0) and Fashion-MNIST (T-shirt).

In Example 2, we set $\gamma = 0.001/\tau$ for $\tau$-GDA and $\tau$-SGA (both with deterministic gradients). We find that $\tau$-SGA has a significant speedup compared to $\tau$-GDA. This is due to a much faster convergence of $\tau$-SGA at $\tau = 10, \theta = 0.15$ compared to the $\tau$-GDA at $\tau = 50$.

On the two image datasets, the speedup is less significant using $\tau$-SGA compared to $\tau$-GDA (both with stochastic gradients). On MNIST (digit 0), we compare $\tau$-GDA at $\tau = 30$ with $\tau$-SGA at $\tau = 5, \theta = 0.075$ using the same discriminator learning rate $\gamma\tau = 0.05$. We find that $\tau$-SGA is slightly faster and it can reach a lower FID (test) score. On Fashion-MNIST (T-shirt), we compare $\tau$-GDA at $\tau = 30$ with $\tau$-SGA at $\tau = 5, \theta = 0.075$ using $\gamma\tau = 0.1$. We find that the speed of $\tau$-GDA and $\tau$-SGA is similar.

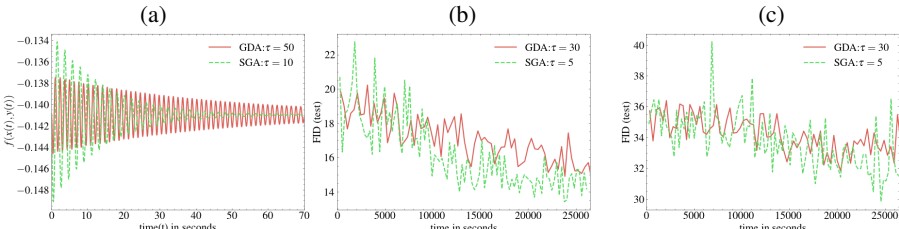

Figure 7: Computational speed of $\tau$-GDA and $\tau$-SGA as a function of a wall-clock time in Example 2 and on MNIST (digit 0) and Fashion-MNIST (T-shirt). The FID (test) score is computed from the fake (GAN model) samples and the test samples of each dataset. a): Example 2. b): MNIST (digit 0). c): Fashion-MNIST (T-shirt).

