# OpenReview forum: "Local convergence of simultaneous min-max algorithms to differential equilibrium on Riemannian manifold"
_ICLR.cc/2025/Conference — ICLR 2025 Poster_

### Official Review · Reviewer_sHDJ · 2024-11-04

**Soundness:** 3
**Presentation:** 3
**Contribution:** 3
**Rating:** 6
**Confidence:** 3

**Summary:**

This paper investigates the local convergence of simultaneous minimax algorithms to differential Stackelberg equilibrium (DSE) and differential Nash equilibrium (DNE) on Riemannian manifolds. The authors study two representative deterministic algorithms, $\tau$-GDA and $\tau$-SGA. For $\tau$-GDA, they derive sufficient conditions for convergence to DSE and DNE. For $\tau$-SGA, they analyze an asymptotic variant and demonstrate a faster convergence rate to DSE under specific conditions. Furthermore, the authors conduct numerical experiments to explore the behavior of stochastic versions of the two algorithms.

**Strengths:**

1. **Significance**: This paper addresses the more general nonconvex-nonconcave setting, extending previous results to Riemannian manifolds.
2. **Comprehensiveness**: It provides a range of results by examining two types of equilibrium points, two algorithms, and extensive numerical results.

**Weaknesses:**

1. **Limited Comparative Analysis and Interpretation**: The relationship and comparative insights between the theoretical results could be strengthened. Here are some examples.
* (i) The paper examines two equilibrium points, DSE and DNE, where DNE is a subclass of DSE with stricter definitions. Convergence to DNE would be expected to require fewer restrictions or achieve a faster rate. Theorems 3.2 and 3.3 support the former expectation, as the range of $\tau$ in Theorem 3.3 is broader than in Theorem 3.2, yet no convergence rate is provided for Theorem 3.3.
* (ii) The authors seem to aim to show the superiority of $\tau$-SGA over  $\tau$-GDA. However, by comparing Theorems 3.2 and 3.4, the range of $\tau$ ensuring convergence for asymptotic $\tau$-SGA exceeds that of $\tau$-GDA only if $\| C + \theta BB^\top \| < \| C \|$. Further discussion on practical ways to ensure this condition or examples where it holds would be helpful. Additionally, there is no theoretical counterpart to Theorem 3.3 for $\tau$-SGA. While $\tau$-SGA is proposed to avoid rotational dynamics, this advantage is demonstrated only in numerical results (e.g., Figure 1), not theoretically, leaving theoretical support insufficient.

2. **Unclear Analysis for (Asymptotic) $\tau$-SGA**:
* (i) In line 327, the asymptotic analysis requires $\theta$ to be of constant order. Why, then, are parameters $\mu$ and $\tau+1$ introduced in Eqs. 12 and 13, and what is their significance?
* (ii) If the authors aim to show the superiority of $\tau$-SGA, Theorem 3.4 alone is insufficient, and Figure 1's simple example is not enough to establish the similarity between $\tau$-SGA and its asymptotic variant. The claim "Theorem 3.4 is valid for $\tau$-SGA" in line 363 lacks rigor. If deriving a theoretical result for $\tau$-SGA is challenging, additional numerical experiments for the asymptotic $\tau$-SGA$ should at least be provided.

**Questions:**

1. In line 277, what is the order of $\gamma^{\cdot}(M)$? Could the authors provide an intuitive explanation of this function?
2. The computation of $\tau$-SGA involves a matrix-vector product. Are the benefits sufficient to justify the additional computational cost?

---

> ### Author Response · Authors · 2024-11-25
>
> > Limited Comparative Analysis and Interpretation: The relationship and comparative insights between the theoretical results could be strengthened. Here are some examples.
> (i) The paper examines two equilibrium points, DSE and DNE, where DNE is a subclass of DSE with stricter definitions. Convergence to DNE would be expected to require fewer restrictions or achieve a faster rate. Theorems 3.2 and 3.3 support the former expectation, as the range of $\tau$ in Theorem 3.3 is broader than in Theorem 3.2, yet no convergence rate is provided for Theorem 3.3.
>
> Thank you for the question. We revised Theorem 3.3 to include
> the convergence rate of $\tau$-GDA to DNE when $\tau=1$, by extending a state-of-the-art result, Theorem 4 in Zhang et al. 2022,
>
> Zhang, Guodong, et al. "Near-optimal local convergence of alternating gradient descent-ascent for minimax optimization." International Conference on Artificial Intelligence and Statistics. PMLR, 2022.
>
> This makes our results on $\tau$-GDA indeed more complete. But we are not sure whether this rate of DNE would be faster than DSE, as the choice of $\tau$ in Theorem 3.2 and 3.3 is different and the constants $\mu$ in the rates are different ($\mu_g \neq \bar{\mu}_g$). We believe that this is not a limitation of our article.
>
> > (ii) The authors seem to aim to show the superiority of $\tau$-SGA over $\tau$-GDA. However, by comparing Theorems 3.2 and 3.4, the range of $\tau$ ensuring convergence for asymptotic $\tau$-SGA exceeds that of $\tau$-GDA only if $| C + \theta BB^\top | < | C |$. Further discussion on practical ways to ensure this condition or examples where it holds would be helpful.
>
> Thanks for the suggestion. In the revised version, we add a further discussion on when $| C + \theta BB^\top | < | C |$ can hold (right after Theorem 3.4):  for a DSE $(x^\ast,y^\ast)$ which is not DNE  (i.e. $C$ is not p.d.), such a choice of $\theta$ can be possible.
>
> However, in practice, we do not know in advance the nature of DSE. Therefore it is hard to know that if a condition can hold in practical examples.
>
> It is possible to construct another toy example, but we believe this is not what the reviewer is asking about.
>
> >  Additionally, there is no theoretical counterpart to Theorem 3.3 for $\tau$-SGA. While $\tau$-SGA is proposed to avoid rotational dynamics, this advantage is demonstrated only in numerical results (e.g., Figure 1), not theoretically, leaving theoretical support insufficient.
>
> We respectfully disagree with this point. In the original SGA article (Letcher et al 2019) this advantage (avoid rotational dynamics) is also only demonstrated empirically,
>
> Letcher, A., Balduzzi, D., Racaniere, S., Martens, J., Foerster, J., Tuyls, K., & Graepel, T. (2019). Differentiable game mechanics. Journal of Machine Learning Research, 20(84), 1-40.
>
> In this article, the main result (Theorem 10) about its local convergence (with $\tau=1$) applies to DNE. But there is no convergence rate given in this theorem. The advantage of this method to avoid rotational dynamics is therefore justified by numerical results. It is likely that a rate upper bound is quite challenging to obtain.
>
> Regarding the theoretical counterpart to Theorem 3.3,
>  in the revised version, we add a comment (at the end of Section 3)
> by mentioning that it is possible to extend Theorem 10 in Letcher et al 2019 to the Riemannian case when $\tau=1$, based on the proof idea of Theorem 3.2 and 3.3. We also add an explanation of this idea right after Theorem 3.3.
>
> > Unclear Analysis for (Asymptotic) $\tau$-SGA:
> (i) In line 327, the asymptotic analysis requires $\theta$ to be of constant order. Why, then, are parameters $\mu$ and $\tau+1$ introduced in Eqs. 12 and 13, and what is their significance?
>
> The reviewer did not fully understand the $\tau$-SGA algorithm. The reason why we first introduce the parameters $\mu$ and $\tau+1$  in Eqs. 12 and 13 is that $\tau$-SGA is derived using the same SGA idea in Euclidean space. We gave a detailed explanation of this in Appendix I. The parameter $\mu$ is free to choose. We then use $\theta$ to rewrite $\mu$ which allows us to perform the analysis of this method when $\tau$ is big. It implies that $\mu$ is of order $1/\tau^2$.

---

> > ### Author Response · Authors · 2024-11-25
> >
> > > (ii) If the authors aim to show the superiority of $\tau$-SGA, Theorem 3.4 alone is insufficient, and Figure 1's simple example is not enough to establish the similarity between $\tau$-SGA and its asymptotic variant. The claim "Theorem 3.4 is valid for $\tau$-SGA" in line 363 lacks rigor. If deriving a theoretical result for $\tau$-SGA is challenging, additional numerical experiments for the asymptotic $\tau$-SGA$ should at least be provided.
> >
> > - First, we excuse the misunderstanding about the interpretation of Theorem 3.4. We now modify the main text in the abstract and introduction (marked in blue), as well as the sentence in line 363 to make the result of Theorem 3.4 precise and clear.
> > Our main point is:
> > we analyze an asymptotic approximation of $\tau$-SGA
> > when the learning rate ratio $\tau$ is big. In some cases, we find that it can achieve a faster convergence rate to differential Stackelberg equilibrium compared to $\tau$-GDA.
> >
> > Therefore we find it unnecessary to add further numerical experiments for the asymptotic $\tau$-SGA.
> >
> > - Secondly, to show the superiority of $\tau$-SGA (as this is challenging in theory), we add further numerical experiments using the full MNIST and Fashion-MNIST datasets (rather than a sub-category). They are non-trivial examples as they are high-dimensional. These results are given in the revised version in Section 4 and Appendix M.
> >
> >
> > > In line 277, what is the order of $\gamma^{\cdot}(M)$? Could the authors provide an intuitive explanation of this function?
> >
> > We do not fully understand the question, which order are you referring to?
> >
> > Thanks again for your question, we now correct a typo in this function by adding the square to $|\lambda|$.
> >
> > To explain this function, we mention in the revised version (right after its definition), that it gives an upper bound of $\gamma$ to control the spectral radius of $I+\gamma M$. But we find it too long to explain intuitively in the main text. In the revision, we added a precise proof about this (see Appendix G, marked in blue).
> >
> > > The computation of $\tau$-SGA involves a matrix-vector product. Are the benefits sufficient to justify the additional computational cost?
> >
> > Thanks for the question. We add a paragraph in Section 4 regarding the computational time. It is not always that $\tau$-SGA is faster in time, but sometimes it can achieve a similar performance as $\tau$-GDA using the same amount of time, suggesting that in terms of the iteration $t$, $\tau$-SGA is 3-4 times faster. Therefore a faster implementation of this method can be useful.
> >
> > We hope that these answer your questions.

---

> > > ### Comment · Reviewer_sHDJ · 2024-11-27
> > >
> > > I thank the authors for their detailed response. I decide to raise my score to 6.

---

### Official Review · Reviewer_ftu6 · 2024-11-05

**Soundness:** 2
**Presentation:** 2
**Contribution:** 1
**Rating:** 1
**Confidence:** 1

**Summary:**

This paper studies min-max algorithms to solve zero-sum differential games on Riemannian
manifold

**Strengths:**

No

**Weaknesses:**

This paper does not have any interesting parts. Firstly, the Riemannian optimization is not practical. Furthermore, could you show some practical utilization of your algorithms?

**Questions:**

I think that this manuscript contributes nothing in science expect some hard understanding of mathematical symbols.

---

> ### Author Response · Authors · 2024-11-25
>
> > This paper does not have any interesting parts. Firstly, the Riemannian optimization is not practical. Furthermore, could you show some practical utilization of your algorithms?
>
> We respectfully disagree that the Riemannian optimization is not practical. Indeed, there are software packages developed already in the literature, therefore one can easily test its usefulness in practice, e.g. the manopt package (https://www.manopt.org/) is implemented in Matlab, Python as well as Julia. Our code will also be released.
>
> To show the usefulness of the method in practice, we now add two examples of GAN, using the full MNIST and Fashion-MNIST datasets. In the MNIST case, the algorithms on Riemannian manifold can achieve similar performance as WGAN-GP in terms of the FID (val) score (with a score below 10, see Table 2).
>
> > I think that this manuscript contributes nothing in science expect some hard understanding of mathematical symbols.
>
> We respectfully disagree with the reviewer regarding this point. Compared to some existing works on the Riemannian min-max problem, our work uses less complicated mathematics. Our theoretical results are supported by numerical results, therefore it is scientific. Our contributions are both conceptual and practical, contributing to the understanding of min-max algorithms

---

### Official Review · Reviewer_Bh31 · 2024-11-07

**Soundness:** 3
**Presentation:** 3
**Contribution:** 3
**Rating:** 6
**Confidence:** 2

**Summary:**

Using the concepts of DSE and DNE on Riemannian manifolds, the author derives conditions on the range of τ and learning rate of x to ensure linear convergence of τ-GDA to DSE and DNE.

It introduces a novel algorithm τ-SGA to enhance the convergence of τ-GDA, allowing a broader range of τ for convergence when τ is large, and achieving faster local convergence than τ-GDA.

Applying these insights, the author improves the training of orthogonal Wasserstein GANs. Numerical results show that enhanced convergence of stochastic τ-GDA and τ-SGA can improve the learned generator on simple benchmarks.

**Strengths:**

This article provides sufficient conditions for the local convergence of τ-GDA and τ-SGA to differential equilibrium on a Riemannian manifold, where the author proves a linear convergence rate of τ-GDA to DSE and DNE, depending on the spectral radius of the Jacobian at equilibrium. THe author introduces τ-SGA on Riemannian manifolds to further improve this rate.

**Weaknesses:**

- Some key terms could benefit from further explanation or intuitive insights. For example, what are the underlying intuitions behind the linear transformations in (9) and (10)?

- It appears that several theorems are extensions of those from the Euclidean case. What are the main differences or key insights that allowed these results to be successfully extended to the Riemannian manifold?

- Given the author's claim of linear convergence rates for the proposed methods, are there any experimental results to support this?

**Questions:**

Question for Theorem 3.1. (Ostrowski Theorem on manifold)

- The author indicates that the theorem is analogous to the theorem in the Euclidean case. Is the Euclidean version of the theorem also based on the spectral radius of a linear transformation similar to (9)?

- What is the size of the local convergence radius around ( 𝑥*, 𝑦*)?

- How can one ensure that the initialization lies within this local neighborhood? Does the initialization used in experiments satisfy this requirement?

- What is the reasoning behind assuming a small spectral radius for 𝐷𝑇* ?

---

> ### Author Response · Authors · 2024-11-25
>
> > Some key terms could benefit from further explanation or intuitive insights. For example, what are the underlying intuitions behind the linear transformations in (9) and (10)?
>
> Thanks for the comment. In the revised version, we explain more clearly (marked in blue)
> the role of $DT^\ast$ and its relation to $M_g$ and $M_s$ in Section 3.
>
> > It appears that several theorems are extensions of those from the Euclidean case. What are the main differences or key insights that allowed these results to be successfully extended to the Riemannian manifold?
>
> You are right. We now give a high-level idea of the proof regarding the extensions right after the revised Theorem 3.3. It is based on the analysis of the Jacobian matrix in a local coordinate system. This matrix is similar to the original one and therefore we can obtain a condition using intrinsic quantifies of the manifold.
>
> > Given the author's claim of linear convergence rates for the proposed methods, are there any experimental results to support this?
>
> Thanks for the question. In the revised version, we replace the plot in
> Fig 1c (Example 2) illustrates the linear convergence rate.
> It is based on computing an approximate geodesic distance, since
> we do not have the formula for the Riemannian distance on the product manifold $M_1 \times M_2$. Fig 1c indicates that the rate is linear since the envelope of the distance under the log scale of the y-axis is a linear function of $t$.
>
> > Question for Theorem 3.1. (Ostrowski Theorem on manifold). The author indicates that the theorem is analogous to the theorem in the Euclidean case. Is the Euclidean version of the theorem also based on the spectral radius of a linear transformation similar to (9)?
>
> Yes, in the Euclidean case, the retraction $R_1$ is of the form $R_{1,x}(\delta) = x + \delta$ (similarly for $R_2$). DT* is therefore reduced to be the Jacobian matrix of the vector field $T(x,y) = (\xi_1 (x,y), \xi_2 (x,y) ) $, in the block form
> $( ( I +  \partial \xi_1 / \partial x, \partial \xi_1 / \partial y ) ,  ( \partial \xi_2 / \partial x,  I + \partial \xi_2 / \partial y ) )$. Ostrowski Theorem then states that (Ortega 1970, Section 10.1.3): If
> $T$ is differentiable at $(x^\ast,y^\ast)$, and the Jacobian matrix at this point has a spectral radius strictly smaller than one, so we have the local linear convergence property for $T$ as in Theorem 3.1.
>
>
> > What is the size of the local convergence radius around ( 𝑥*, 𝑦*)? How can one ensure that the initialization lies within this local neighborhood? Does the initialization used in experiments satisfy this requirement?
>
> This is a very good question, but we currently do not know the local neighborhood size in theory. We now add a comment on this point right after Theorem 3.1 (Ostrowski Theorem on manifold): This proof does not give an explicit way to construct the local stable region $S_\delta$. Therefore
> it is unclear what a good initialization entails. To obtain a precise size of $S_\delta$, extra assumptions on $f$ would be needed.
>
> > What is the reasoning behind assuming a small spectral radius for 𝐷𝑇* ?
>
> Thanks for the question. In general, if the spectral radius of $DT^\ast$ is small, then one obtains a faster upper bound of the convergence rate according to our Definition 1.
>
> The spectral radius of the operator gives also a lower bound on the speed of convergence of the iterates of the method on affine vector
> fields, according to Theorem 2 of Azizian et al 2020,
>
> Azizian, W., Mitliagkas, I., Lacoste-Julien, S., & Gidel, G. (2020, June). A tight and unified analysis of gradient-based methods for a whole spectrum of differentiable games. In International conference on artificial intelligence and statistics (pp. 2863-2873). PMLR.
>
> Therefore in some cases, a smaller spectral radius for 𝐷𝑇* ensures a faster convergence rate.
>
> We hope that these answer your questions.

---

> > ### Comment · Reviewer_Bh31 · 2024-11-26
> >
> > I thank the authors for the responses. I shall keep my score unchanged.

---

### Official Review · Reviewer_v3Po · 2024-11-08

**Soundness:** 4
**Presentation:** 4
**Contribution:** 3
**Rating:** 6
**Confidence:** 3

**Summary:**

This paper analyzes the local convergence of the deterministic τ-GDA and τ-SGA algorithms to differentiable equilibria in min-max games on Riemannian manifolds. Using the Ostrowski theorem, sufficient conditions on the algorithm hyperparameters (in particular, on the learning ratio τ) are given for linear-rate local convergence to Differentiable Stackelberg Equilibria (DSE) for both τ-GDA and Asymptotic τ-SGA (an approximation to τ-SGA that is amenable to easier analysis). Local convergence to Differentiable Nash Equilibria (DNE) is also given for τ-GDA.

This paper marks the first time that τ-SGA has been written for Riemannian manifolds, and the authors put it forth as an algorithm that can solve the problem of τ-GDA’s slow convergence rate in certain settings. Both theory and example are given to show that Asymptotic τ-SGA can be locally convergent at a smaller τ compared to τ-GDA, and with a faster rate; moreover, in Euclidean space the rate guarantee outperforms that of the extra-gradient method, which has been used to overcome the slow convergence rate of τ-GDA but is costly in Riemannian space. Experiments also show that the gap between Asymptotic τ-SGA and the true τ-SGA is small, so the theoretical results should also hold for the more natural τ-SGA.

Finally, the paper applies stochastic τ-GDA and τ-SGA to train orthogonal Wasserstein GANs with a discriminator parametrized in the Stiefel manifold. It is shown that even with good initialization a small τ may cause τ-GDA to oscillate with high amplitude, but that τ-SGA can converge even at those small values of τ.

**Strengths:**

Overall, this paper seems like a significant algorithmic contribution with sound theoretical grounding. The authors extend known, tight local convergence results for τ-GDA from Euclidean space to Riemannian manifolds and introduce τ-SGA to the context of Riemannian manifolds. Both in theory and practice τ-SGA is shown convincingly to be able to achieve (local convergence) in settings where τ-GDA cannot, highlighting its usefulness in this setting of Riemannian min-max problems with minimal assumptions (just f twice continuously differentiable).

The paper is generally well written. The sections on differentiable equilibria, algorithms, and local convergence read especially well, with much background and explanation given to readers. Motivations and the context of prior work and clearly written.

**Weaknesses:**

While it is clearly stated in the introduction that global convergence is difficult to achieve in this setting and local convergence (to differentiable equilibria) may be all we can get theoretically, it is not clear what these limitations mean, or are expected to mean, in practice. The experiments are set up to show local convergence after good initialization, but it is not clear what the good initialization entails or if τ-SGA can be used when you are not near a DSE. Also maybe some comment could be made about the existence and significance of DSE in the settings provided (e.g., the Wasserstein GANs), and about other methods for doing min-max Riemannian optimization in these setting (e.g., over Steifel manifolds).

I also find the writing of the GAN experiments and their setups/parameters/initialization a bit terse and less clearly written than what comes before, with a few things in particular lacking explanation. I ask some questions about this below.

**Questions:**

From the experiments, it seems like τ-SGA is used as a sort of “last mile” algorithm to guarantee fast convergence after good initialization/pretraining (either via ansatz in the analytical examples of Figure 1 or via τ-GDA for the GANs). Is this a reasonable interpretation of the authors’ proposed use of τ-SGA, and is there any reason that τ-GDA—in particular, alternating τ-GDA—is used to pretrain the GANs?

In Theoerems 3.2 and 3.4, after the main statements there are additional (“furthermore…”) statements for when γ is fixed exactly depending on L_g/L_s. Is this reasonable given that L_g depends on the DSE, which is not known in advance? (Maybe this is not so important, but as it is written I am not sure what the significance of these “furthermore” statements is.)

τ-SGA has an additional hyperparameter θ (or μ) to tune compared to τ-GDA, and in the GAN example of Table 2 a “suitable choice” of θ seems to have been made. Does the value of θ matter much, and how is it chosen?

I am not so familiar with the way “retraction” is used in Section 3.1, though it does not affect much as the algorithms are easily understood. Perhaps some quick definition or source could be given?

---

> ### Author Response · Authors · 2024-11-25
>
> > While it is clearly stated in the introduction that global convergence is difficult to achieve... it is not clear what these limitations mean, or are expected to mean, in practice.
>
> Regarding global convergence, we now make our vision clearer in the introduction (see the revised version, marked in blue).
> Indeed, the related works cited in Table 1 all focus on global convergence
> but their assumptions on f do not apply to GANs. This is why our focus is on the local convergence where $f$ has a weaker assumption. We however add a citation (Benaïm, Michel, and Laurent Miclo. "The asymptotic behavior of fraudulent algorithms." arXiv preprint arXiv:2401.12605 (2024).) to the introduction to show that global convergence remains an exciting open question by using novel algorithmic development/analysis.
>
> > It is not clear what the good initialization entails or if τ-SGA can be used when you are not near a DSE.
>
> It is non-trivial to ensure a good initialization for the local convergence. In the revised version, we now make this point cleaner by adding a comment right after Theorem 3.1 (Ostrowski theorem on manifold). The major challenge of this theorem is that the local stable region is non-constructive in our proof, therefore even in theory, we do not know how to decide the initialization.
>
> We nevertheless performed some experiments. We find that in Example 2 if τ-SGA is not so far from the DSE with a random initialization of $(x,y)$, it can still converge to the DSE.
> We also observe in Appendix M. Figure 5, that $\tau$-SGA can first go away from a local solution region, and then converge to another one, suggesting that this algorithm has some global convergence property. This observation is also noted in Appendix M.2.
>
> Your question is complicated to answer in general, as the function $x \to \max_{ y \in M_2} f(x,y)$ can be a non-convex function. Therefore it $x$ is initialized too far away from $x^\ast$ (of a DSE), it can happen that τ-SGA (or τ-GDA) may converge to another DSE (if it exists).
>
> > Also maybe some comment could be made about the existence and significance of DSE in the settings provided (e.g., the Wasserstein GANs), and about other methods for doing min-max Riemannian optimization in these settings (e.g., over Steifel manifolds).
>
> Thanks for the suggestion. We are currently not sure of
> the existence of DSE under the assumption that $f$ is $C^2$.
> In the literature, the existence of a stationary point
> sometimes appears as an assumption when $f$ is $C^2$, see e.g. Assumption 1 in Han el al. 2023.
>
> Han, Andi, et al. "Riemannian Hamiltonian methods for min-max optimization on manifolds." SIAM Journal on Optimization 33.3 (2023): 1797-1827.
>
> However, they do not address the question regarding the existence of a stationary point. They do mention that when $f$ is g-convex-concave, such a point exists. However, this assumption of $f$ typically does not hold in GAN training.
>
> Among all the articles cited in Table 1, we find that only the algorithms proposed in Han el al. 2023 have been used in the training of GANs. The other works focus on other problems such as robust neural network training. In Han el al., only a 2d mixture of Gaussian example is studied, therefore it is not comparable to our setting as we study higher dimensional data distributions.
>
> As we mention in the conclusion, the significance of DSE in the practice of GANs still remains an open problem.
>
> > From the experiments, it seems like τ-SGA is used as a sort of “last mile” algorithm to guarantefast convergence after good initialization/pretraining e... Is this a reasonable interpretation of the authors’ proposed use of τ-SGA, and is there any reason that τ-GDA—in particular, alternating τ-GDA—is used to pretrain the GANs?
>
> Yes, you are right that we use τ-SGA as a “last mile” algorithm in the numerical experiments. This is intentional because the article is about the local convergence of min-max algorithms. In our considered WGANs, we therefore perform a pre-training before running τ-GDA and τ-GDA based on another algorithm.
>
> Regarding the initialization method, we have chosen $\tau$-GDA for the Gaussian generator case, since it is not so long to run this algorithm with a large $\tau=100$ (note that using a small $\tau=1$ is unstable according to Figure 2). Similarly, we could use $\tau$-GDA with $\tau=30$ for the pre-training in the image dataset cases. But we find that alternating $\tau$-GDA can achieve this pre-training much faster since it allows for the use of a smaller $\tau = 5$ ($\tau$-GDA will be unstable). We did not explain these details in Appendix L.6, but we can add them into it if the reviewer find it important. It gives a reasonable solution (in terms of the Wasserstein distance or (val) FID score of the generator). We currently have not studied systematically the behavior of τ-SGA about its global convergence. We believe that this is an exciting topic for future work.

---

> > ### Author Response · Authors · 2024-11-25
> >
> > > I also find the writing of the GAN experiments and their setups/parameters/initialization a bit terse and less clearly written than what comes before, with a few things in particular lacking explanation.
> >
> > Thanks for your comment. We now rewrite this section, to make it clearer and to add some further numerical simulations on the full MNIST and Fashion-MNIST datasets. Some results are deferred to Appendix M due to the lack of space.
> > Please let us know if it is clearer or not.
> >
> > > In Theoerems 3.2 and 3.4, after the main statements there are additional (“furthermore…”) statements for when γ is fixed exactly depending on L_g/L_s. Is this reasonable given that L_g depends on the DSE, which is not known in advance? (Maybe this is not so important, but as it is written I am not sure what the significance of these “furthermore” statements is.)
> >
> > Thanks for the question. We make this point clearer in the revised version
> > by moving our last sentence of the paragraph after "furthermore" (marked in blue), to clarify the significance of this improved rate. It is actually a theoretical contribution, relative to an existing result regarding the convergence rate of the extra-gradient method in Euclidean space. We find that in our considered case, Asymptotic $\tau$-SGA can achieve a faster rate compared to it. Therefore our result is significant in theory.
> >
> > > τ-SGA has an additional hyperparameter θ (or μ) to tune compared to τ-GDA, and in the GAN example of Table 2 a “suitable choice” of θ seems to have been made. Does the value of θ matter much, and how is it chosen?
> >
> > The choice of $\theta$ was tuned on MNIST (0) dataset by running $\tau$-SGA for $T=200k$ iterations. We have considered the range of theta between $[0.01,0.5]$ and we found that $\theta=0.01$ was too small to make it unstable (i.e. with large oscillations of the $f$ and angle around 0 over iterations). With $\theta=0.5$, we find that the dynamics of $f$ over iterations has some (rare) peaks during the training, even though the trend of $f$ still goes down.
> >
> > To answer your question, does the value of $\theta$ matter much, we now add an experiment on theta sensitivity on the MNIST (0) dataset, Fig. 5 in Appendix M.2.
> > It shows that the FID (test) score change over iteration/time is nearly the same over a wide range of $\theta $ (except $\theta=0.01$, too small)
> >
> > We did not perform an exhaustive search for the optimal value of $\theta$, but we found that the one chosen in the article works well across different datasets. This is why we chose this value.
> >
> > > I am not so familiar with the way “retraction” is used in Section 3.1 ... Perhaps some quick definition or source could be given?
> >
> > Thanks for the suggestion. In the revised version, we add a citation that explains the concept of “retraction”. We can add the definition, together with some examples, into the appendix if needed, but this is too long to write in the main text.
> >
> > We hope that these answer your question.

---

### Official Review · Reviewer_Lg9L · 2024-11-08

**Soundness:** 3
**Presentation:** 2
**Contribution:** 3
**Rating:** 6
**Confidence:** 3

**Summary:**

This paper extends the analysis of the local convergence of two-time-scale $\tau$-GDA and $\tau$-SGA methods to the differential Stakelberg equilibrium (DSE) of Euclidean min-max problem to the Riemannian min-max problem. This paper first generalizes the notion of DSE in the Euclidean space to the Riemannian manifold (in Section 2.1), using a local coordinate chart. Then, the authors provide a sufficient condition (in Theorem 3.1) for the fixed point method to have a local linear convergence. Based on Theorem 3.1, this paper shows a specific sufficient condition for the $\tau$-GDA to locally linearly converge to DSE in Theorem 3.2. In addition, the authors generalize $\tau$-SGA for Riemannian min-max problem, which is originally developed to mitigate undesirable rotational dynamics encountered by $\tau$-GDA in the Euclidean space. In the Euclidean space, extragradient (or optimistic gradient) method is widely used to relieve such rotational behavior, but its Riemannian version, named RCEG, is computationally expensive. This paper claims that their $\tau$-SGA equipped with auto-differentiation may have computational benefit over RCEG. Experiments on a toy example and a more realistic Orthogonal WGAN problem empirically confirm this paper's theoretical findings.

**Strengths:**

Existing results on local convergence to DSE and DNE in nonconvex-nonconcave min-max problems in Euclidean space are crucial. Thus, while its extension to Riemannian manifolds may seem straightforward, it is a significant and non-trivial advancement. Although I was not able to check all the details of the proof, the parts I checked looked correct.

**Weaknesses:**

Missing preliminaries for non-experts on Riemannian manifold: For example, adding the definition of Riemannian gradient/Hessian/cross-gradient, the role of local coordinate chart and the relation between $f$ and $\bar{f}$ would help readers to better understand the context.

**Questions:**

- lines 45-47 could be improved to better explain this paper's focus on differential equilibriums (DNE and DSE). This could have been local Nash and local minimax points. Then why differential equilibrium?
- Table 1 is confusing: What is your focus here? The title should be more specific to make a point. Isn't Nash equilibrium found by Zhang et al. (2023) DNE? This is not clear in the table. Also, your result is local (both in optimality and convergence) while others might be global. I am not sure whether local/global should be discussed in the table, but I am sure that this table needs to be reorganized and rewritten.
- I suggest rewriting Definition 3.1 to make it more explicit that it defines local linear convergence.
- line 276: Were you trying to say that $M$ with eigenvalues with only negative real parts?
- lines 524-525: I don't follow what do you mean here.

---

> ### Author Response · Authors · 2024-11-25
>
> > Missing preliminaries for non-experts on Riemannian manifold: For example, adding the definition of Riemannian gradient/Hessian/cross-gradient, the role of local coordinate chart and the relation between $f$ and $\bar{f}$ would help readers to better understand the context.
>
> We thank for the suggestion but we find that there is no enough space in the main text to include these preliminaries. In the revised version, we add some major references that give the definition of Riemannian gradient/Hessian/cross-gradient, the local coordinate chart, and the relation between $f$ and $\bar{f}$. If the reviewer finds it necessary, we can detail them in the appendix.
>
> > lines 45-47 could be improved to better explain this paper's focus on differential equilibriums (DNE and DSE). This could have been local Nash and local minimax points. Then why differential equilibrium?
>
> Thanks for suggesting this. We now make this point clearer on line 46 (see the revised version, marked in blue) to say that our focus on DSE and DNE is due to the generic property of these equilibria among LSE and LNE on smooth $f$.
>
> > Table 1 is confusing: What is your focus here? The title should be more specific to make a point. Isn't Nash equilibrium found by Zhang et al. (2023) DNE? This is not clear in the table. Also, your result is local (both in optimality and convergence) while others might be global. I am not sure whether local/global should be discussed in the table, but I am sure that this table needs to be reorganized and rewritten.
>
> Thanks for the suggestion. In the revised version, we modified the title of Table 1 to make our focus clearer. We want to say that compared to the other works which are about global convergence, we focus on the case where $f$ can be satisfied relative to the practice of GANs. This is not the case in the existing works.
>
> Even though we study the local convergence problem in the case where $f$ is $C^2$. The $f$ in GANs  that we considered are also $C^2$.
>
> The Nash equilibrium studied by Zhang et al. (2023) is not necessarily DNE (but it can be). According to Theorem 3.1 in this article, the assumption made on $f$ does not involve strong geodesic convexity/concavity, therefore the Nash equilibrium can be non-DNE, such as in a bilinear game in Euclidean space.
>
> > I suggest rewriting Definition 3.1 to make it more explicit that it defines local linear convergence.
>
> We have modified Definition 3.1 to make this point clear. Thanks for suggesting this.
>
> > line 276: Were you trying to say that $M$ with eigenvalues with only negative real parts?
>
> You are right. We have made the correction.
>
> > lines 524-525: I don't follow what do you mean here.
>
> We have modified this last sentence to make it clearer: our study suggests that DSE might be a suitable solution set towards which our considered initialization algorithm (alternating) $\tau$-GDA converges.
>
> We hope that these answer your question.

---

### Meta-Review · Area_Chair_LMud · 2024-12-20

**Metareview:**

This paper extends the analysis of local convergence for deterministic $\tau$-GDA and $\tau$-SGA algorithms to min-max problems on Riemannian manifolds, generalizing concepts like Differentiable Stackelberg Equilibrium (DSE) and Differentiable Nash Equilibrium (DNE) from Euclidean spaces. Using theoretical tools like Ostrowski's theorem, the authors provide sufficient conditions for local linear convergence of $\tau$-GDA and a faster convergence rate for an asymptotic variant of $\tau$-SGA, which addresses the slow convergence and undesirable rotational dynamics of $\tau$-GDA. Unlike the computationally expensive Riemannian extragradient methods, $\tau$-SGA offers practical advantages with minimal performance gaps between its theoretical and natural forms. Empirical validation includes training orthogonal Wasserstein GANs on the Stiefel manifold, demonstrating $\tau$-SGA's robustness and improved convergence even under challenging parameter settings.

This paper seems like a significant algorithmic contribution with sound theoretical grounding. The authors extend known, tight local convergence results for τ-GDA from Euclidean space to Riemannian manifolds and introduce τ-SGA to the context of Riemannian manifolds. Existing results on local convergence to DSE and DNE in nonconvex-nonconcave min-max problems in Euclidean space are crucial. Thus, while its extension to Riemannian manifolds may seem straightforward, it is a significant and non-trivial advancement. Both in theory and practice τ-SGA is shown convincingly to be able to achieve (local convergence) in settings where τ-GDA cannot, highlighting its usefulness in this setting of Riemannian min-max problems with minimal assumptions (just f twice continuously differentiable).

The paper is generally well written. The sections on differentiable equilibria, algorithms, and local convergence read especially well, with much background and explanation given to readers. Motivations and the context of prior work and clearly written.

**Additional Comments On Reviewer Discussion:**

The authors did a good job responding to reviewer concerns. Note that one superficial review rating of 1/10 was ignored.

---

### Decision · Program_Chairs · 2025-01-22

Accept (Poster)